# Quantifying spatiotemporal variability of glacier algal blooms and the impact on surface albedo in southwest Greenland

Shujie Wang[1], Marco Tedesco[1,2], Patrick Alexander[1,2], Min Xu[3], Xavier Fettweis[4]

[1]Lamont-Doherty Earth Observatory, Columbia University, Palisades, NY 10964, USA.
[2]NASA Goddard Institute for Space Studies, New York, NY 10025, USA.
[3]Department of Geography, University of Alabama, Tuscaloosa, AL 35401, USA.
[4]Department of Geography, University of Liège, Liège 4000, BELGIUM

*Correspondence to*: Shujie Wang (wangshujie23@gmail.com)

**Abstract.** Albedo reduction due to light-absorbing impurities can substantially enhance ice sheet surface melt by increasing
surface absorption of solar energy. Glacier algae have been suggested to play a critical role in darkening the ablation zone in southwest Greenland. It was very recently found that the Sentinel-3 Ocean and Land Colour Instrument (OLCI) band ratio $R_{709nm}/R_{673nm}$ can characterize the spatial patterns of glacier algal blooms. However, Sentinel-3 was launched in 2016 and current data are only available over three melting seasons (2016-2019). Here, we demonstrate the capability of the MEdium Resolution Imaging Spectrometer (MERIS) for mapping glacier algae from space and extend the quantification of glacier
algal blooms over southwest Greenland back to the period 2004–2011. Several band ratio indices (MERIS chlorophyll-a indices and the impurity index) were computed and compared with each other. The results indicate that the MERIS two-band ratio index (2BDA) $R_{709nm}/R_{665nm}$ is very effective in capturing the spatial distribution and temporal dynamics of glacier algal growth on bare ice in July and August. We analyzed the interannual (2004–2011) and summer (July–August) trends of algal distribution and found significant seasonal and interannual increases in glacier algae close to the Jakobshavn Isbrae Glacier
and along the middle dark zone between the altitudes of 1200 m and 1400 m. Using broadband albedo data from the Moderate Resolution Imaging Spectroradiometer (MODIS) we quantified the impact of glacier algal growth on bare ice albedo, finding a 0.02~0.04 reduction rate in albedo for each algal population doubling. Our analysis indicates the strong potential for the satellite algal index to be used to reduce bare ice albedo biases in regional climate model simulations.

## 1 Introduction

Snow and ice play a critical role in regulating the global energy balance through high surface albedos (Skiles et al., 2018; Warren, 1982). The presence of light-absorbing impurities, including abiotic materials (such as mineral dust and black carbon; e.g. Flanner et al., 2007; Goelles and Bøggild, 2017; Wientjes et al., 2011) and biogenic materials primarily produced by microbial processes (Chandler et al., 2015; Ryan et al., 2018; Stibal et al., 2017; Williamson et al., 2019), can substantially reduce the surface albedo of snow and ice and thus enhance surface melt. Increased meltwater further decreases

surface albedo, triggering a positive feedback mechanism between meltwater production and albedo decline (Box et al., 2012; Tedesco et al., 2011, 2016).

Snow algae and glacier algae are among the main microbial communities in supraglacial environments, which are distributed in Greenland, Antarctica, Alaska, Svalbard, the Himalayas, Siberia, the Rocky Mountains, or the European Alps (Anesio et al., 2017). Algal growth on glaciers and ice sheets not only plays an important role in local and regional carbon and nutrient

cycles but is also crucial for regulating surface melt processes through the reduction in snow and ice albedo resulting from dark algae pigmentation (Lutz et al., 2014; Remias et al., 2012; Stibal et al., 2017; Yallop et al., 2012). Snow algae (mainly Chlorophyceae) are psychrophiles residing in glacial snow or snowfields and bloom on the snow surface after the onset of melting (Lutz et al., 2016, 2017). The visible colour of snow algae varies from green to yellow to orange and red, and is determined by the pigments (chlorophylls, xanthophylls, and secondary carotenoids, etc.) produced in different life stages

(Anesio et al., 2017). Glacier algae (Zygnematales) are different from snow algae, and grow on the bare ice glacier surface when liquid water, nutrients, and photosynthetically active radiation are sufficient (Lutz et al., 2018; Stibal et al., 2017; Yallop et al., 2012). The earliest documentation about glacier algae dates to 1872. During an expedition to Greenland in 1870, Adolf Erik Nordenskiöld and fellow explorers found 'a brown polycellular alga' on the ice surface and within cryoconite holes (Nordenskiöld, 1872). Several field studies (Lutz et al., 2018; Stibal et al., 2015, 2017; Uetake et al., 2010;

Yallop et al., 2012) have investigated the species composition and cell structures of glacier algal communities. The primary glacier algal species are Ancylonema nordenskiöldii, Mesotaenium berggrenii, and Cylindrocystis brebissonii, which are green microalgae and produce pigments including chlorophyll-a, chlorophyll-b, beta-carotene, lutein, and violaxanthin. Ancylonema nordenskiöldii and Mesotaenium berggrenii also generate a phenolic purpurogallin pigment (purpurogallin carboxylic acid-6-O-b-D-glucopyranoside) which absorbs ultraviolet and visible radiation (Remias et al., 2012; Yallop et al.,

2012). It has been suggested that this purpurogallin pigment accounts for the brownish-grey colour of the algae-laden ice (Remias et al., 2012; Yallop et al., 2012).

Recent studies have revealed a significant impact of glacier algal blooms on bare ice albedo in Greenland (Stibal et al., 2017; Tedstone et al., 2020; Williamson et al., 2018). Along the ablation zone over the southwest Greenland Ice Sheet, a dark ice band appears every summer season (Shimada et al., 2016; Tedstone et al., 2017). It was previously thought that this surface

darkening was primarily caused by outcropping of ancient dust (Wientjes and Oerlemans, 2010). Recently, widespread glacier algal blooms were observed in the field and the dark pigments generated by glacier algae were argued to be a primary control on the presence of the dark band (Ryan et al., 2018; Stibal et al., 2017; Williamson et al., 2018; Williamson et al., 2020). Field sampling and spectral measurements indicate that glacier algae have a greater effect on albedo reduction than other nonalgal impurities (Stibal et al., 2017). However, current field measurements of glacier algal abundance and surface

albedo are limited to a very few sites and melting seasons, and it is logistically difficult to use laboratory techniques to measure glacier algae at a regional scale. The impact of glacier algal development on surface albedo over large spatial and temporal scales has not yet been quantified.

Remote sensing provides a synoptic and efficient way to characterize geospatial phenomena across large spatial scales. To date, using remote sensing methods to quantify snow or glacier algae extent or concentration is limited to a few studies (e.g.

Cook et al., 2020; Ganey et al., 2017; Huovinen et al., 2018; Painter et al., 2001; Takeuchi et al., 2006; Wang et al., 2018). Painter et al. (2001) estimated the algal abundance of the snow alga Chlamydomonas nivalis over a snow-covered region in the Sierra Nevada of California from Airborne Visible/Infrared Imaging Spectrometer (AVIRIS) hyperspectral imagery based on chlorophyll-a absorption features between 630 nm and 700 nm. Despite the high capability of airborne hyperspectral imaging data for detecting chlorophylls, the availability of hyperspectral imaging data is constrained over

space and time. Several studies (e.g. Takeuchi et al. 2006; Ganey et al. 2017; Huovinen et al. 2018) mapped red snow algae based on carotenoid absorption features using satellite red and green bands.

Mapping glacier algae using remote sensing is complicated by a number of factors, including the complex pigmentation of glacier algae, insufficient spectral and spatial resolution of satellite data, and the impact of dusts and underlying ice optics that are not yet well understood. The use of carotenoid features is not applicable to glacier algae, as they do not, to our

knowledge, generate secondary carotenoids like snow algae (Painter et al., 2001; Takeuchi et al., 2006). The brownish-grey colour of glacier algae is attributed to the purpurogallin pigments, but the characteristic absorption peaks of purpurogallin pigments are concentrated in the ultraviolet spectrum at 278 nm, 304 nm, and 389 nm (Remias et al.,2012), which are not detectable by current satellite sensors. At visible wavelengths, the absorption by purpurogallin pigments is quite uniform, making it difficult to differentiate between glacier algae and other dark impurities from satellite data based on purpurogallin

spectral properties.

The spectral signature of chlorophyll-a, the primary photosynthetic pigment generated by glacier algae, however, is well-suited for mapping glacier algae using satellite remote sensing techniques. Chlorophyll-a is widely used as a biomarker to detect or quantify algal blooms from remote sensing data (e.g. Gitelson, 1992; Painter et al., 2001), and it was recently found that the spectral signatures of chlorophyll-a in the red and near-infrared (NIR) region can be utilized for mapping glacier

algae (Wang et al., 2018). The red-NIR spectral signature of chlorophyll-a, i.e. absorption at 665-681 nm and reflectance around 709 nm, is present in field hyperspectral data collected over ice surfaces covered by glacier algae (Cook et al., 2020; Stibal et al., 2017). The concentration of chlorophyll-a is generally used as a proxy for algal biomass or abundance, and based on this a number of algorithms have been developed to quantify the biomass contained in algal blooms occurring in aquatic systems (Beck et al., 2016; Blondeau-Patissier et al., 2014; Matthews, 2011; Xu et al., 2019a, 2019b). Using the two-

band ratio ($R_{709nm}/R_{673nm}$) method, Wang et al. (2018) quantified the spatial distribution of glacier algal blooms in southwest Greenland over the summer seasons in 2016 and 2017 from the Sentinel-3 Ocean and Land Colour Instrument (OLCI) data. Despite the moderate (300 m) spatial resolution, the derived spatial pattern based on the red-NIR chlorophyll-a signature matches well with previous field observations (Stibal et al., 2015; Stibal et al., 2017; Williamson et al., 2018). As for higher spatial resolution remote sensing data, Cook et al. (2020) applied a random forest method to classify unmanned aerial vehicle

(UAV) and the Sentinel-2 Multispectral Instrument (MSI) data for identification of high-algae biomass and low-algae biomass areas. However, these data have limitations in terms of spatial coverage, temporal resolution, and spectral

resolution. To establish a long-term time series quantification of glacier algae distribution and study the seasonal process of glacier algal blooms and the impact on albedo change, the use of chlorophyll-a-sensitive ocean color satellite sensors is promising.

The Sentinel-3 OLCI is equipped with 21 spectral bands including seven narrow chlorophyll-a bands. The advanced band configuration of OLCI makes it a valuable sensor for mapping algal blooms not only in water but also on ice (Wang et al., 2018). OLCI was designed based on the opto-mechanical and imaging design of MEdium Resolution Imaging Spectrometer (MERIS) onboard the European Space Agency (ESA)'s Envisat satellite, operational from March 2002 to April 2012, which collected data in 15 spectral bands between 390 nm and 1040 nm. MERIS features in particular a 709 nm band where high

levels of chlorophyll-a produce a characteristic reflectance peak. MERIS data have been broadly used for atmospheric and oceanic studies, with the primary goal of measuring the concentration of chlorophyll-a and suspended sediments in oceans, coastal waters, and inland lakes (Gower et al., 2008; Palmer et al., 2015). Similar configurations of the chlorophyll-targeted bands in terms of wavelength and bandwidth between MERIS and OLCI (Fig. 1a) point to the potential of using MERIS data to reconstruct the spatial distribution of glacier algae prior to 2012. In this study, we make use of the capability of MERIS

for detecting chlorophyll-a to extend the quantification of glacier algae in southwest Greenland back to the 2004–2011 period, and further quantify the impact of glacier algal blooms on bare ice albedo by combining the time series data of MERIS and MODIS.

## 2 Study area and data

### 2.1 Study area and previous field observations

Our study area is located between 66-71°N and 47-51°W in southwest Greenland. This area features high ablation rates and low surface albedos during summertime (Alexander et al., 2014; Fettweis et al., 2011; Moustafa et al., 2015; Stroeve et al., 2013). With the progression of surface melt over time, a dark ice zone forms rapidly and reaches a maximum area from mid-July to mid-August (Tedstone et al., 2017; Wang et al., 2018). The bare ice and dark ice areal extent is highly correlated with meltwater production and surface runoff simulated by the regional climate model Modèle Atmosphérique Régionale (MAR)

(Wang et al. 2018). The peak time of surface darkening coincides with the occurrence of glacier algal blooms observed in the field. The ice alga Ancylonema nordenskiöldii and Mesotaenium berggrenii are the dominant species found in southwest Greenland during July and August (Lutz et al. 2018; Yallop et al. 2012; Williamson et al. 2018). Considering the growth season and surface habitat of glacier algae, we focus our analysis on bare ice in July and August.

There are a limited number of field studies measuring glacier algal abundance and reflectance spectra over the study area

(Cook et al., 2020; Stibal et al., 2015; Stibal et al., 2017; Williamson et al., 2018), and no field measurements were coincident with the acquisition time of the Envisat MERIS data. Here we utilized the previous field observations in a qualitative way for comparison purposes, and extended the derived empirical function from the Sentinel-3 OLCI data (Wang et al., 2018) to MERIS data for characterizing the temporal variations of algal population with surface albedo change. We

utilized field data first presented by Stibal et al. (2015) and Stibal et al. (2017) to validate patterns of spatial variability in
glacier algae distribution and to compare with satellite data to validate the chlorophyll-a spectral signal. Stibal et al. (2015)
collected shallow surface ice cores and measured algal abundance over 14 sites in Greenland during May-September 2013,
of which the sites DS (69°28.56'N, 49°34.838'W), KAN_M (67°3.964'N, 48°49.356'W), and KAN_L (67°5.798'N,
49°56.303'W) are within our study area. KAN_M and KAN_L are located along the Kangerlussuaq transect (K-transect),
and DS is located near to the Jakobshavn Isbrae Glacier. Stibal et al. (2015) documented the algal abundance averaged over
the sampling season (2013 summer) for each site, finding cell counts of 66±31 cells/ml (KAN_L), 5688±3147 cells/ml
(KAN_M) and 10621±2073 cells/ml (DS), respectively. During the 2014 summer season, Stibal et al. (2017) collected both
algal abundance and hyperspectral reflectance measurements via an Analytical Spectral Devices (ASD) Field Spectrometer
over a site near the automatic weather station S6 (67°04.779'N, 49°24.077'W) on the K-transect. They collected multiple
samples each observation day and published the datasets of glacier algal abundance and reflectance spectra at a 10 nm
spectral resolution (Stibal et al., 2017). Here we used the field hyperspectral data to compare with the satellite spectra to
validate the chlorophyll-a signal.

## 2.2 Satellite data

### 2.2.1 MERIS Level-2 data

We used the full spatial resolution (300 m) MERIS Level-2 data acquired during July and August from 2004 to 2011
(https://earth.esa.int/web/guest/-/meris-full-resolution-full-swath-6015). The MERIS Level-2 data were processed from the
Level-1b data (top-of-atmosphere radiances in 15 spectral bands shown in Fig. 1a). ESA adopted different processing
techniques to generate the Level-2 data over land, water, and clouds. The Level-2 data over land include the normalized
surface reflectance in 13 spectral bands, corrected for the atmospheric effects of gaseous absorption and stratospheric
aerosols (ESA, 2011). The full resolution Level-2 data from May 2002 to April 2012 were released at the MERCI file
archive (https://merisfrs-merci-ds.eo.esa.int/) in February 2015. We identified 146 cloudless MERIS images acquired on 135
days from July to August between 2004 and 2011. Since there were no cloudless images available for the 2002 summer
season and only three images for the 2003 summer over the study area, we excluded these two years from our analysis. For
those images affected by clouds over the study area, we checked the MERIS Level-2 Flag data including the pixel types
classified as water, land, and cloud. However, the Flag data fail to correctly capture all the cloud pixels due to limitations of
the algorithm in differentiating clouds from other bright surfaces like snow and ice (ESA, 2011). In this regard, we manually
removed the cloud pixels (patches) from each MERIS image.

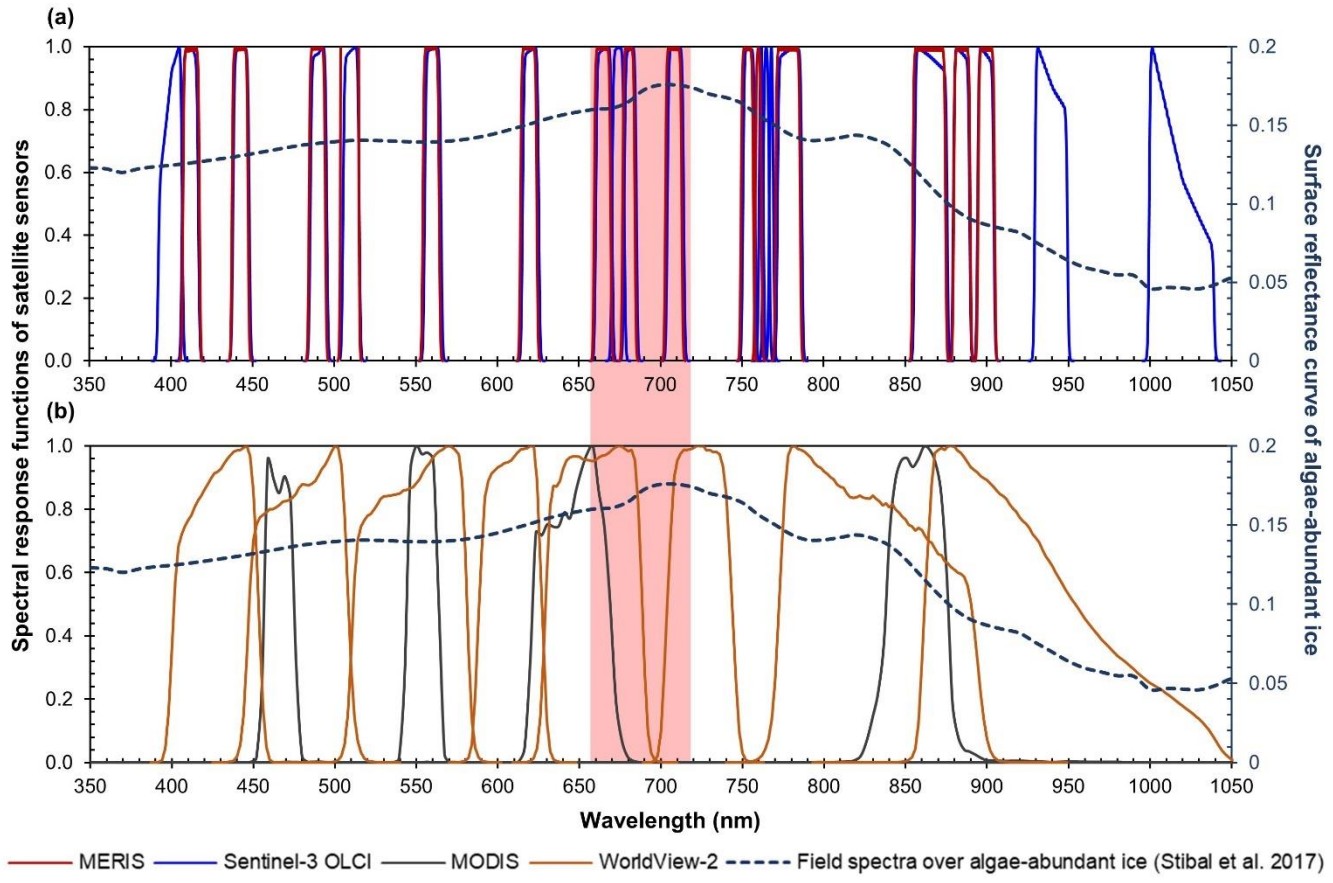

**Figure 1. Spectral response functions of (a) MERIS (red), OLCI (blue), and (b) MODIS (black), and WorldView-2 (orange) over the wavelength range of 350-1050 nm. All the MERIS and OLCI bands are within the 350-1050 nm range, where photosynthetic and photoprotective pigments have spectral responses. Four MODIS bands (over land) and eight WorldView-2 bands are within this spectral range, but with much coarser spectral resolutions. In both sub-plots, the dashed line shows hyperspectral ASD field spectrometer data (right vertical axis) collected over algae-abundant ice by Stibal et al. (2017), containing chlorophyll-a signal at the red-NIR wavelengths (red highlighted region). The plotted field spectrum (sample code: Ab.25.06.14.D1) was measured on 25 June 2014 at 67°04.779'N, 49°24.077'W (near the automatic weather station S6 along the K-transect), with an algal abundance measurement of 121664 cells/ml (Stibal et al., 2017).**

### 2.2.2 MODIS data

We used the MODIS/Terra daily surface reflectance product (MOD09GA Version 6) and daily snow cover product (MOD10A1 Version 6). The MOD09GA data include the atmospherically corrected surface reflectance for the 620-670 nm, 841-876 nm, 459-479 nm, 545-565 nm, 1230-1250 nm, 1628-1652 nm, and 2105-2155 nm MODIS bands (Fig. 1b). The MOD10A1 data include broadband albedo estimated based on the MOD09GA product. We used the version 6 data which are greatly improved in sensor calibration, cloud detection, and aerosol retrieval and correction relative to version 5 (Casey et al., 2017; Lyapustin et al., 2014; Toller et al., 2013). Version 6 data are recommended for assessing temporal variability of surface albedo since they are corrected for sensor degradation issues that impacted earlier versions (Casey et al., 2017). The

spatial resolution of the MODIS datasets is 500 m. We resampled the MODIS data to 300 m using a nearest neighbour resampling method. The cloud masks in the MOD10A1 data were applied to exclude clouds.

### 2.2.3 WorldView-2 imagery

We also used WorldView-2 imagery to validate the spectral signal of glacier algae captured by MERIS data. The WorldView-2 satellite was launched in October 2009, collecting data in nine spectral bands (panchromatic, coast, blue, green, yellow, red, red edge, NIR, and NIR2, Fig. 1b) at a very high spatial resolution (~2 m for the multispectral bands). WorldView satellites have high geolocation accuracy owing to their three-axis stabilized platform equipped with high-precision GPS and attitude sensors (Wang et al., 2016). Although the WorldView-2 spectral bands have wide bandwidths, the red (630-690 nm) and red edge (705-745 nm) bands can capture the chlorophyll-a signal (Fig. 1b), and have been used for mapping algal species in nearshore marine habitats (Reshitnyk et al., 2014). We obtained WorldView-2 imagery acquired in July and August (2009-2011) from the Polar Geospatial Center (PGC, https://www.pgc.umn.edu/). The images were provided as orthorectified top-of-atmospheric radiances in eight multispectral bands. We performed atmospheric corrections to the radiance images and obtained surface reflectance images using the MODerate resolution atmospheric TRANsmission (MODTRAN) based Fast Line-of-sight Atmospheric Analysis of Hypercubes (FLAASH) (Anderson et al., 2002). The sub-Arctic model and rural aerosol model were used for correction of atmospheric effects caused by water vapour and aerosols (Legleiter et al., 2013).

### 2.3 Modèle Atmosphérique Régionale (MAR) outputs

The regional climate model Modèle Atmosphérique Régionale (MAR, Fettweis et al., 2017) combines atmospheric modelling (Gallée and Schayes, 1994) with the Soil Ice Snow Vegetation Atmosphere Transfer Scheme (De Ridder and Gallée, 1998) to simulate surface energy balance and mass balance processes over the Greenland and Antarctic ice sheets. In this study, we examined the relationship between the MAR albedo bias (e.g. Alexander et al., 2014; Moustafa et al., 2015) and glacier algal blooms. The snow albedo in MAR is determined by snowpack temperature, temperature gradient, and liquid water content, and the bare ice albedo is scaled based on the accumulated surface water (Alexander et al., 2014). Since the MAR albedo scheme does not account for impurities, there are significant biases in MAR albedo over the southwest Greenland ablation zone (Alexander et al., 2014). We used the 7.5 km resolution MAR v3.9 daily outputs, forced by the European Centre for Medium-Range Weather Forecasts Interim Reanalysis (ERA-Interim; Dee et al., 2011).

## 3 Methods

### 3.1 Bare ice mapping

We mapped bare ice cover from each MERIS image using a thresholding method applied to surface reflectance data (e.g. Shimada et al., 2016; Tedstone et al., 2017; Wang et al., 2018). To be consistent with previous studies, we used MODIS-

derived bare ice maps as a reference to determine the optimal threshold for the MERIS data. We removed tundra and ocean pixels using the MEaSUREs Greenland Ice Mapping Project classification mask (Howat et al., 2014). We selected 31 MOD09GA images that were coincident with MERIS overpasses and were cloud free over the study area. Following Tedstone et al. (2017), we applied a threshold to the MODIS 841-876 nm reflectance ($R_{841-876\ nm}$), using the criterion $R_{841-876\ nm} < 0.6$ to extract bare ice reference maps from selected MODIS images. For coincident MERIS images, we iteratively applied a threshold value ranging from 0 to 1, increasing by 0.01 at each iteration to the MERIS band 13 (865 nm) and compared the MERIS and MODIS bare ice cover. The optimal threshold was determined based on the F1 score accuracy metric, which is the harmonic average of precision and recall, defined as follows:

$$F1 = 2 * (precision * recall) / (precision + recall) \tag{1}$$

where *precision* is calculated as $N_{TP} / (N_{TP} + N_{FP})$ and *recall* is calculated as $N_{TP} / (N_{TP} + N_{FN})$. $N_{TP}$ is the number of true positives (the number of pixels classified as bare ice by both the MODIS and MERIS data), $N_{FP}$ is the number of false positives (the number of pixels that are only classified as bare ice by the MERIS data), and $N_{FN}$ is the number of false negatives (the number of pixels that are only classified as bare ice by the MODIS data). The average F1 score was calculated for each threshold based on those 31 image pairs. The threshold of 0.53 yielded the highest F1 score (0.957). We excluded supraglacial lakes using the modified normalized difference water index (MNDWI, Yang and Smith, 2013), defined as:

$$MNDWI = (R_{blue} - R_{red}) / (R_{blue} + R_{red}) \tag{2}$$

where $R_{blue}$ is the reflectance at 442 nm (MERIS band 2) and $R_{red}$ is the reflectance at 665 nm (MERIS band 7). Pixels with MNDWI greater than 0.14 (Yang and Smith, 2013) were identified as lake pixels and excluded from analysis. Using the same iterative method described above, we also determined an optimal threshold of 0.47 to extract dark ice pixels (pixels with bare ice containing substantial surface impurities) using the 620 nm MERIS band, following Shimada et al. (2016) and Tedstone et al. (2017). This band has been used to delineate dark ice by applying a threshold based on the assumption that visible wavelengths including the red band are mostly affected by light-absorbing impurities rather than surface water and grain size variations (Shimada et al., 2016; Tedstone et al., 2017; Wang et al., 2018).

### 3.2 Chlorophyll-a indices and impurity index

Chlorophyll-a is the primary photosynthetic pigment generated by glacier algal cells (Williamson et al., 2018; Yallop et al., 2012). Hyperspectral field measurements (Fig. 2d, Cook et al., 2020; Stibal et al., 2017) and the Sentinel-3 OLCI spectra (Wang et al., 2018) both exhibit the typical spectral signatures of chlorophyll-a at the red and NIR wavelengths over algae-abundant ice surfaces, featuring a reflectance peak around 709 nm and absorption features around 665-681 nm. Pure ice has lower reflectance at 709 nm compared to shorter wavelengths (Hall and Martinec, 1985). The magnitude of the reflectance

peak at 709 nm relative to 665-681 nm is highly dependent on the chlorophyll-a content (Binding et al., 2013; Gitelson, 1992). Figure 2d shows the MERIS spectra over a dark ice pixel, compared with WorldView-2 spectra and field hyperspectral measurements by Stibal et al. (2017). The selected MERIS pixel, located near to the Jakobshavn Isbrae

240 Glacier, is close to the site DS where Stibal et al. (2015) measured a high abundance of glacier algae during the 2013 summer season. The MERIS image (Fig. 2a) was acquired on 5 July 2010, and the WorldView-2 image (Fig. 2b and Fig. 2c) was acquired on 9 July 2010. The field hyperspectral curves shown in Fig. 2d were measured over dark ice (R620nm<0.4) with high algal abundance (greater than 10000 cells/ml), featuring chlorophyll-a signatures in the red-NIR region. Despite the differences in absolute values of surface reflectance, the spectral shapes of the MERIS, WorldView-2 and field spectra

245 match quite well, particularly with regard to the presence of chlorophyll-a, validating the ability of MERIS data to capture the glacier algae spectral signal.

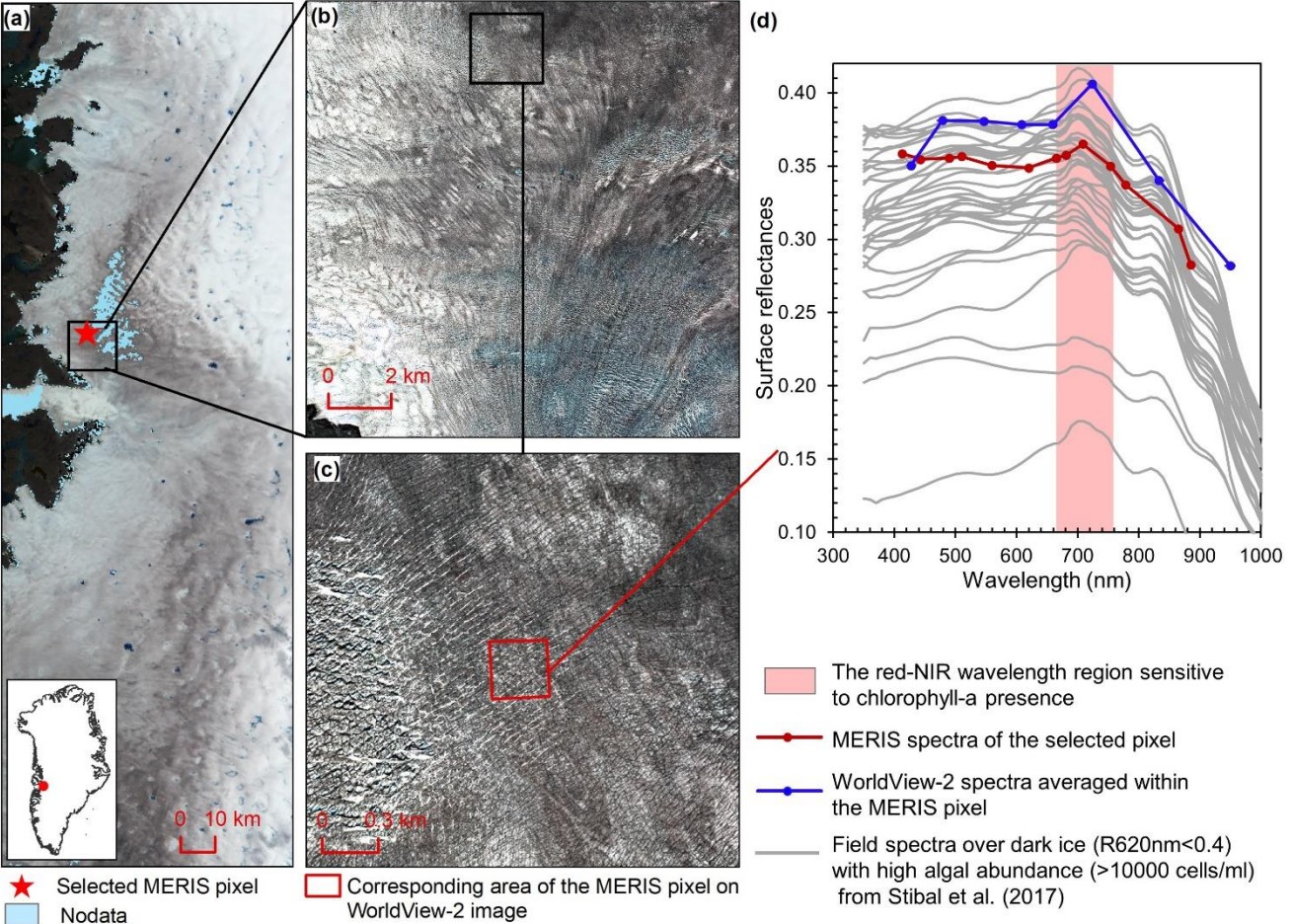

**Figure 2. Comparison between MERIS, WorldView-2, and field spectra over algae-abundant dark ice. (a) MERIS Level-2 image (true colour composite) acquired on 5 July 2010. Pixels with missing data are shown in light blue. (b) WorldView-2 surface**
250 **reflectance image acquired on 9 July 2010 over the square area in (a). (c) Zoomed-in WorldView-2 image, with the area (red**

square) corresponding to the selected MERIS pixel in (a). (d) Reflectance spectra for MERIS and WorldView-2 (2010), and field hyperspectral measurements collected over the algae-abundant dark ice at S6 by Stibal et al. (2017) in 2014.

To map glacier algae using the chlorophyll-a spectral signature, we calculated several MERIS chlorophyll-a indices (Table 1), including the two-band NIR–Red index (2BDA), three-band NIR–Red index (3BDA), normalized difference chlorophyll index (NDCI), and maximum chlorophyll index (MCI) (Moses et al., 2012; Mishra and Mishra, 2012; Binding et al., 2013). The 2BDA and 3BDA methods have been widely applied to estimate chlorophyll-a concentration in aquatic systems using MERIS data (Beck et al., 2016; Moses et al., 2009; Xu et al., 2019a, 2019b), and have proved to be highly accurate for chlorophyll-a retrieval in turbid coastal waters characterized by complex optical properties (Moses et al., 2012). The NDCI (Mishra and Mishra, 2012) was defined based on the concept of the normalized difference vegetation index (NDVI). The MCI measures the height of the 709 nm reflectance peak relative to the baseline obtained by interpolating reflectance between 681 nm and 753 nm (Binding et al., 2013). In addition, we also calculated the impurity index (Dumont et al., 2014) to compare with the chlorophyll-a indices. The impurity index is the ratio between the natural logarithms of the spectral albedos at the green and NIR bands, and was constructed to quantify the impurity content over the Greenland Ice Sheet upon the assumption that the visible wavelengths are much more sensitive to impurity content than the NIR wavelengths. Radiative transfer modelling experiments have shown that the impurity index is less affected by the snow grain size variations than the presence of impurities (Dumont et al., 2014).

Table 1. Equations and MERIS bands used for calculation of different ratio indices.

| Indices | Equation | MERIS bands |
| --- | --- | --- |
| Two-band NIR–Red index (2BDA) | $R_{709nm} / R_{665nm}$ | B7, B9 |
| Three-band NIR–Red index (3BDA) | $(R_{665nm}^{-1} - R_{709nm}^{-1}) * R_{753nm}$ | B7, B9, B10 |
| Normalized Difference Chlorophyll Index (NDCI) | $(R_{709nm} - R_{665nm}) / (R_{709nm} + R_{665nm})$ | B7, B9 |
| Maximum Chlorophyll Index (MCI) | $(R_{709nm} - R_{681nm}) - (R_{753nm} - R_{681nm}) * (709 - 681) / (753 - 681)$ | B8, B9, B10 |
| Impurity index | $\ln(R_{560nm}) / \ln(R_{865nm})$ | B5, B13 |

## 3.3 Sensitivity analysis based on radiative transfer modelling

To evaluate the sensitivity of chlorophyll indices to dust presence, we performed radiative transfer modelling tests using the Snow, Ice, and Aerosol Radiation model (SNICAR; Flanner et al., 2007, 2009). SNICAR is a multi-layer, two-stream radiative transfer model for simulating the spectral albedos of snow over the 300-5000 nm wavelength range (at a 10 nm spectral resolution), accounting for various factors including illumination conditions, snow grain size, snow layer properties, and dust concentrations, etc. The SNICAR online tool (available at snow.engin.umich.edu) allows for simulating the radiative effects of dust in four size bins, in ranges of 0.1-1.0, 1.0-2.5, 2.5-5.0, and 5.0-10.0 μm. Dust optical properties in SNICAR are based on an estimate of global-mean characteristics approximated as a mixture of quartz, limestone,

montmorillonite, illite, and hematite. We simulated the spectral albedos for varying sizes and concentrations of dust under the following conditions: direct incident radiation, a solar zenith angle of 60 degrees, clear sky conditions for Summit, Greenland, a snow grain effective radius of 1500 μm (approximating the ice surface), a snowpack thickness of 100 m, a snowpack density of 400 kg/m$^3$, a range of dust concentrations (0.1, 0.3, 0.5, 0.8, 1, 1.5, 2, 2.5, 3, 5, 8, 10, 30, 50, 80, 100, 300, 500, 800, 1000, 1500, 2000, 2500, and 3000 ppm), and four dust sizes (dust 1: 0.1–1.0μm; dust 2: 1.0–2.5μm; dust 3: 2.5–5.0μm; dust 4: 5.0–10.0 μm). We tested different values of snow grain effective radius (500 μm vs. 1500 μm) and snow density (400 kg/m$^3$ vs. 900 kg/m$^3$). The snow density value has a negligible effect on the simulation results. The simulated spectra using a snow grain effective radius of 1500 μm is the best approximation to the MERIS spectra for clean bare ice.

## 4 Results

### 4.1 Comparison between different ratio indices

Figure 3 shows the MERIS spectra over four distinct sites within our study area to illustrate the spectra associated with different surface types. Each site represents a typical surface type, including clean bare ice, dark ice with a significant chlorophyll-a signal, dark ice with a less significant chlorophyll-a signal, and a supraglacial lake. Figure 3b shows that each surface type is characterized by a distinct spectral curve. The difference between the spectral curves for the two dark ice sites is particularly notable. Figure 3c shows the normalized spectral curves relative to the clean ice spectrum. Both of the dark ice sites have a reflectance at 620 nm of less than 0.47 and are classified as 'dark ice' based on the thresholding method discussed above (Shimada et al., 2016; Tedstone et al., 2017). However, the northern dark ice site has a chlorophyll-a spectral signature between 665 nm and 753 nm that matches the field spectra of algae-abundant ice (Fig. 2d), while at the southern dark ice site, the reflectance peak at 709 nm is much less pronounced. The differences illustrate that pixels classified as "dark ice" can have different spectral properties, and in particular differences associated with reflectance characteristics of chlorophyll-a.

We calculated the 2BDA, 3BDA, NDCI, MCI, and impurity indices over bare ice ($R_{865\ nm}<0.53$) for each MERIS image. Table 2 lists the ratio indices and the reflectance at 620 nm over the four sites shown in Fig. 3a based on a MERIS image acquired on 14 August 2011, to illustrate the differences between indices. The 2BDA, 3BDA, and NDCI chlorophyll-a indices use similar spectral bands and are in general well-correlated; they are highest over the northern dark ice site, and lowest over the supraglacial lake. The MCI chlorophyll-a index, in contrast, reaches a maximum over clean bare ice. The MCI index measures the height of the 709 nm reflectance peak relative to the baseline between 681 nm and 753 nm, and is therefore sensitive to the bare ice spectrum. This index may be less sensitive to the relatively low chlorophyll-a content over ice, and is more suitable for monitoring intense algal blooms with very high chlorophyll-a concentrations in water (Binding et al. 2013). For the impurity index, the clean bare ice has the lowest value, followed by the supraglacial lake, dark ice with the weaker chlorophyll-a signal, and dark ice with the stronger chlorophyll-a signal (Table 2). The supraglacial lake has a higher impurity index relative to clean ice, suggesting that the impurity index may include the darkening effect caused by

meltwater presence. We find that the 2BDA, 3BDA and NDCI indices are most suitable for detection of chlorophyll-a, given their specificity to chlorophyll-a signal bands, the sensitivity of the impurity index to liquid water, and the sensitivity of the MCI index to the bare ice spectrum. Of these three indices, we selected the 2BDA index to characterize the glacier algae distribution owing to its simplicity and effectivity.

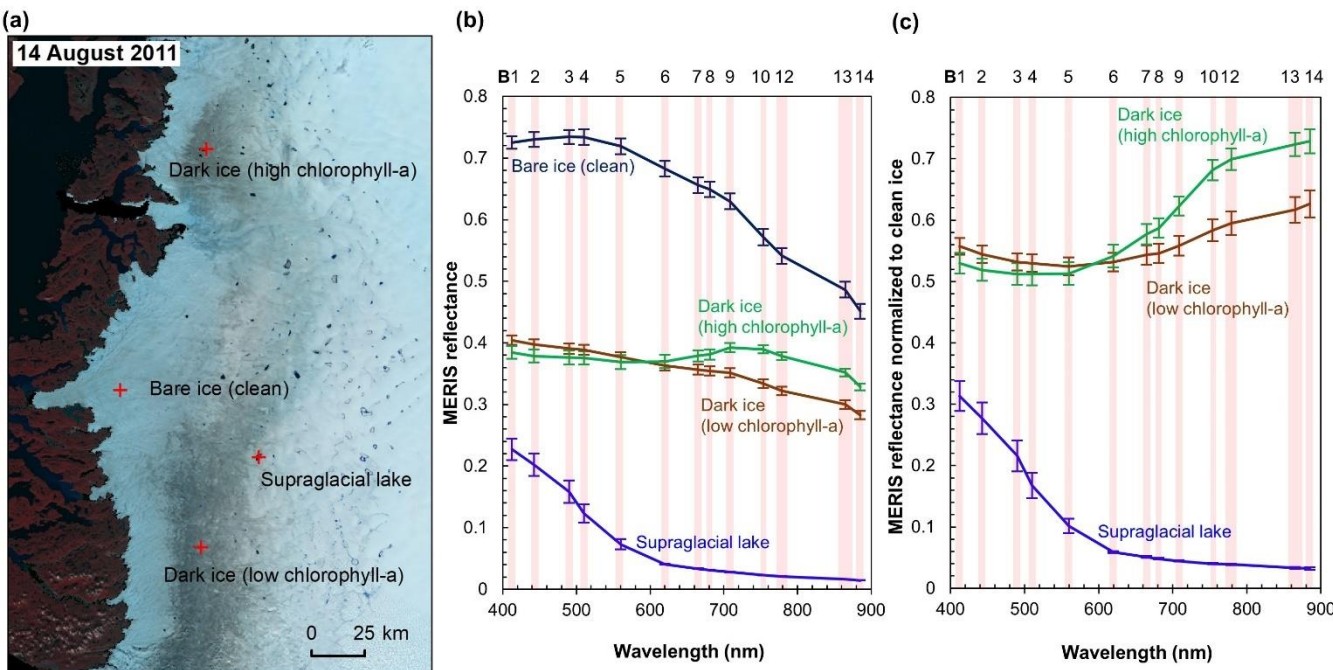

**Figure 3. MERIS spectra of different surface types. (a) MERIS Level-2 image (false colour composite) acquired on 14 August 2011 and locations of the four sample sites. Each site has an area of 1.2 km by 1.2 km, composed of 16 MERIS pixels. (b) MERIS reflectance in 13 spectral bands over the four sites, illustrated by mean and standard deviation values for each band over each site. (c) Normalized reflectance relative to the clean ice spectra.**

**Table 2. Calculated ratio indices and surface reflectance at 620 nm over the four sites.**

| Surface type | 2BDA | 3BDA | NDCI | MCI | Impurity | $R_{620nm}$ |
|---|---|---|---|---|---|---|
| Bare ice (clean) | 0.960 | -0.037 | -0.021 | 0.011 | 0.457 | 0.683 |
| Dark ice (high chlorophyll-a) | 1.035 | 0.035 | 0.017 | 0.008 | 0.955 | 0.369 |
| Dark ice (low chlorophyll-a) | 0.986 | -0.014 | -0.007 | 0.005 | 0.809 | 0.362 |
| Supraglacial lake | 0.839 | -0.131 | -0.087 | 0.000 | 0.635 | 0.040 |

### 4.2 Sensitivity of the 2BDA index to non-algal factors

Given that dust may change the spectral reflectance of bare ice and affect the 2BDA index, we analyzed the sensitivity of 2BDA index to dust presence based on the SNICAR simulations for varying dust sizes and concentrations. We should note

here that there has been some discussion in past literature regarding hematite-rich dust (e.g. Tedesco et al., 2013; Cook et al., 2020), which could produce a different spectral response. However, the field study of Cook et al. (2020) found very low concentrations of such dust, and therefore we consider its impact to be negligible. Using the simulated spectra, we calculated the 2BDA and impurity indices for each dust size and concentration. Figure 4 shows the 2BDA index vs. impurity index calculated for the SNICAR simulations (with circle diameters representing the magnitude of dust concentrations for four

different dust sizes), along with the density scatterplots of impurity vs. 2BDA index calculated from the MERIS data. The SNICAR simulations show that the impurity index is more sensitive to dust than the 2BDA index. Figure 4 illustrates that the upper bound of the impurity index calculated from the MERIS data is around 1.0. This maximum value corresponds to a dust concentration of ~500 ppm (for the 5.0-10.0 μm dust range), which is consistent with the measurements of Cook et al. (2020), who reported mean and maximum dust concentrations of 342 ppm and 519 ppm respectively over a field site within

the study area. However, SNICAR simulations indicate that an impurity index of 1.0 corresponds to a maximum 2BDA value of 0.99. Therefore, the presence of dust alone cannot explain the high 2BDA index values present in Fig. 4. This comparison suggests that for our study area, areas with a 2BDA index greater than 0.99 are not likely to be false positives caused by dust.

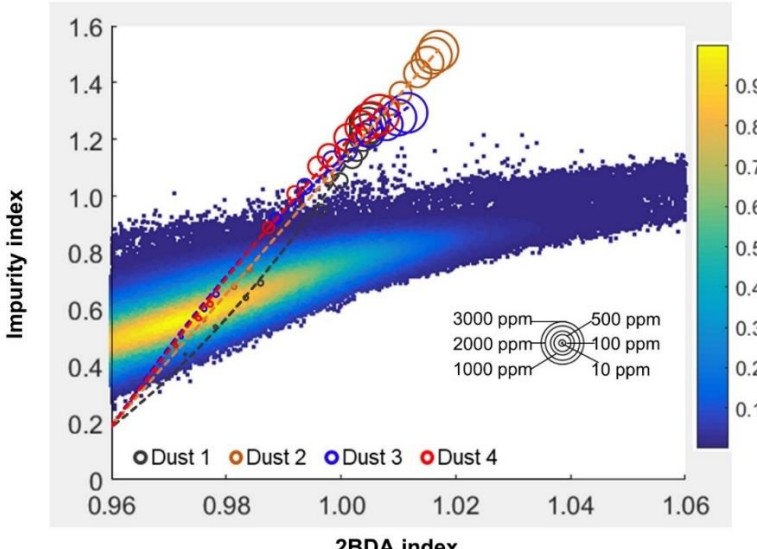

**Figure 4. Impurity index vs. 2BDA index for MERIS bare ice pixels (density scatter plot with colours indicating relative frequency), excluding missing data in our study area, between 2004 and 2011. Circles show impurity vs. 2BDA index from SNICAR simulations with varying concentrations of dust (with four different dust sizes). The circle size corresponds to the dust concentration, and dashed lines show the polynomial regression for each of the different dust sizes.**

Although the bare ice spectrum can also be affected by other factors such as air bubbles and meltwater presence, there is no

evidence suggesting that these factors can generate the chlorophyll-a-like spectral signature with a higher reflectance at 709 nm as compared with 665 nm. In fact, ice with different concentrations of air bubbles has a consistent spectral shape between 665 nm and 709 nm (Condom et al., 2018), and meltwater exhibits a similar pattern to ice at this wavelength range (Fig. 3b),

with both the ice and water spectra characterized by a decreased reflectance from 665 nm to 709 nm. The sensitivity of the 2BDA index to glacier algae can be further demonstrated using the field dataset of Cook et al. (2020). Table C1 and Figure
C2 (Appendix C) indicate strong positive correlations between measured cell abundance and the 2BDA index calculated from coincident in-situ hyperspectral data, particularly for those samples with a measured cell abundance of greater than 10000 cells/ml, which have an average 2BDA index of 1.09±0.073. In comparison, the samples with a measured cell abundance of lower than 10000 cells/ml have an average 2BDA index of 0.98±0.015.

### 4.3 Spatial variability

To examine spatial variability on a broader scale, Fig. 5 shows the spatial patterns of the mean 2BDA index, impurity index, reflectance at 620 nm, and MODIS broadband albedo for the bare ice zone in our study area, averaged over 135 days when MERIS images are available between 2004 and 2011. Figure 5a, which shows patterns of the 2BDA index, suggests glacier algae are abundant at the DS region close to the Jakobshavn Isbrae Glacier between the altitudes of 600 m and 1200 m, and in the middle ablation area (68.5°N-66.5°N) between 1200 m and 1400 m. These patterns are consistent with glacier algal
maps derived from the Sentinel-3 OLCI data for the 2016 and 2017 summer season (Wang et al., 2018). The relative magnitude of the 2BDA values between the DS, KAN_L, and KAN_M sites also matches the relative magnitude of field measurements of glacier algal abundance (Stibal et al., 2015; circles on Fig. 5), with the highest 2BDA index and algal abundance at the DS site, a lower value at the KAN_M site, and the lowest value at the KAN_L site.

A comparison between Figs. 5a and 5b and an examination of variation of the indices with elevation (Fig. A1) indicate a
similarity in the spatial distribution of the two indices but also notable differences. In particular, the 2BDA index reaches a peak at an elevation of 1300 m, while the impurity index peaks at 1180 m. As suggested by our sensitivity analysis discussed in Section 4.2, the 2BDA index is primarily sensitive to chlorophyll-a, while the impurity index is sensitive to materials that darken the electromagnetic spectrum in visible wavelengths, including abiotic impurities (e.g. outcropping particulates; Wientjes et al., 2012), biological impurities, and liquid water. The map of reflectance at 620 nm, the band commonly used to
delineate dark ice using a threshold (determined to be 0.47 for MERIS), is shown in Fig. 5c. Similar to the impurity index, the 620 nm reflectance and MODIS broadband albedo (Fig. 5d) reach a minimum value at 1180 m in elevation (Fig. A1; Fig. 5d). Comparison between the three indices and MODIS albedo suggests that algal abundance is highest between 1200 and 1400 m in elevation, contributing to reduced albedo, while other factors may play a more important role in albedo reduction below 1200 m in elevation. In particular, the darkening between 1000 m and 1200 m in elevation could be attributed to
longer exposure of bare ice resulting in increased consolidation of particulates with melt (Tedesco et al., 2016), where "wavy" patterns of outcropping dust can be observed (Wientjes and Oerlemans, 2010; Fig. A2). In contrast, imagery (WorldView-2) suggests that these "wavy" patterns may not be present at higher elevations where the appearance of dark material is more consistent with distributed algal material (Fig. A2). Other factors that may contribute to a reduction in MODIS albedo include liquid water and surface crevasses (e.g. Ryan et al., 2018), though their fraction is small relative to
other surface types (Ryan et al., 2018).

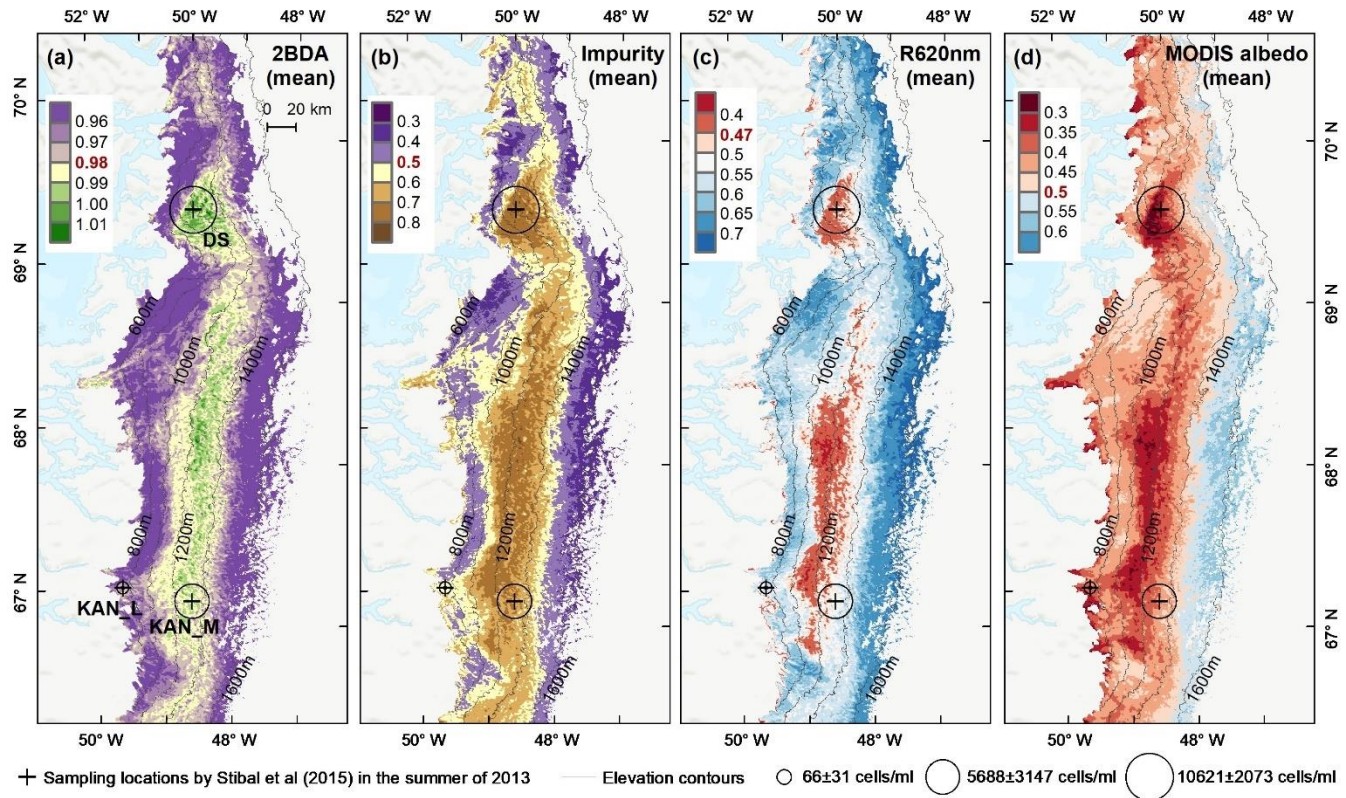

**Figure 5. Spatial patterns of the mean 2BDA index (a), impurity index (b), reflectance at 620 nm (c), and MODIS broadband albedo (d) over the bare ice zone during July and August from 2004 to 2011. The elevation contours illustrate the spatial variations of each variable with altitude. The cross labels show the spatial locations of the field sites DS, KAN_L, and KAN_M and magnitude of glacier algal abundance (circle labels) measured by Stibal et al. (2015) in 2013.**

## 4.4 Interannual variability

The annual time series (July-August mean) of the 2BDA index (Fig. 6a) and the impurity index (Fig. 6b) show the interannual variability of algal abundance and impurity content, indicating a general increasing trend in bare ice area, algal abundance, and total impurity content between 2004 and 2011, particularly after 2006. The spatial extent of glacier algae also expanded towards higher elevations (1200 m – 1400 m) over this period. Between 2004 and 2011, the 2BDA index reached a maximum in 2010 when high air temperatures and intensive surface melt occurred over Greenland (Tedesco et al., 2011). The impurity index exhibits similar interannual variability compared with the 2BDA index, but also exhibits different spatiotemporal variations between 1000–1200 m and 1200–1400 m in the middle ablation area. Figure A3 illustrates the interannual variability of the average 2BDA and impurity indices at different elevation levels (600-800m, 800-1000m, 1000-1200m, and 1200-1400m). In particular there are notable differences in variability of the 2BDA index between the 1000-1200m and 1200-1400m levels. The interannual variability of the 2BDA and impurity indices is also coherent with

variability in Greenland ice sheet-wide summer albedo, which was lowest in 2010 and highest in 2006 for the period 2004–2011 (Tedesco et al., 2018).

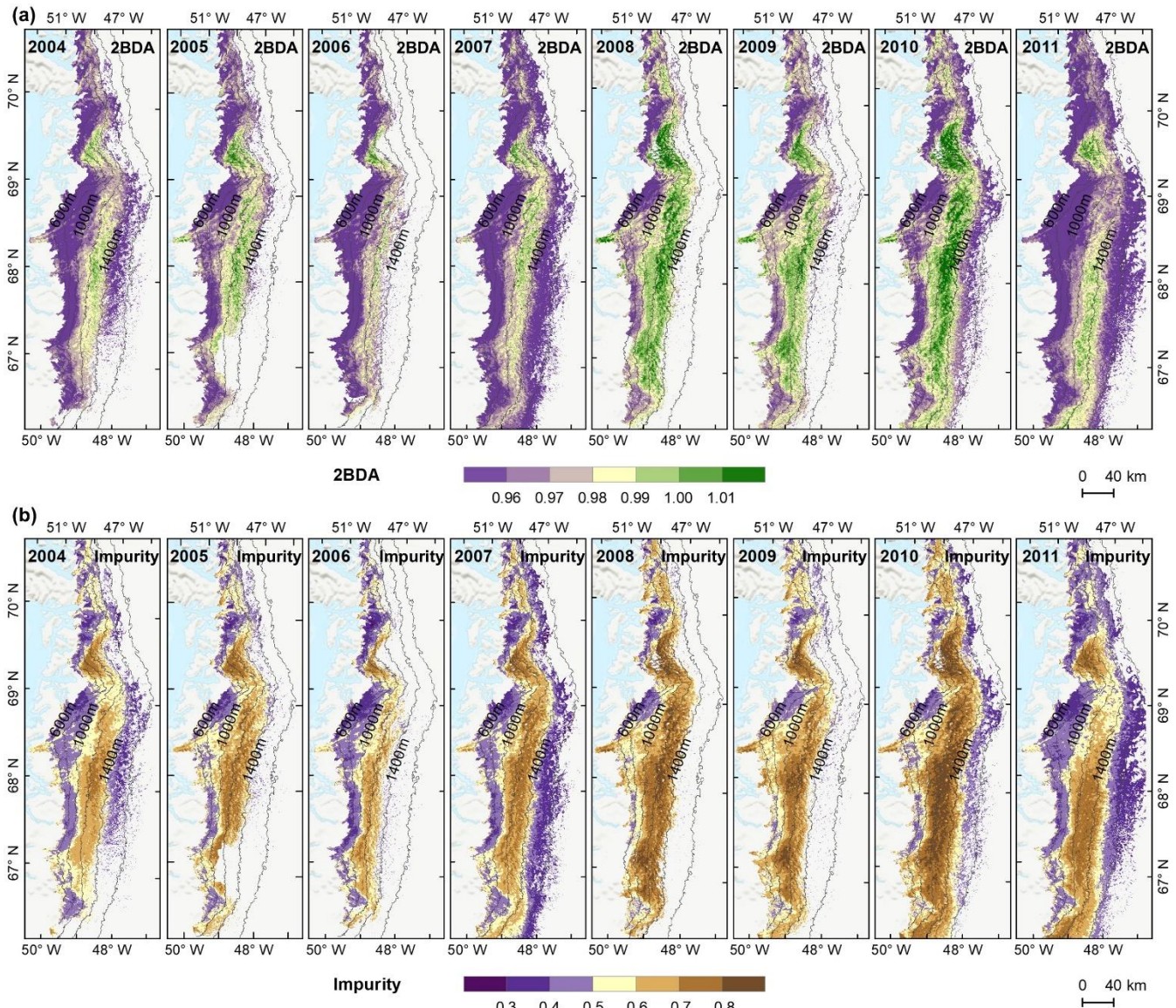

**Figure 6. Maps of mean 2BDA index (a) and impurity index (b) over July and August from 2004 to 2011.**

We also calculated interannual trends in the 2BDA index, impurity index, and MODIS broadband albedo using linear regression analysis, with the mean 2BDA index, impurity index, or MODIS albedo for each year as the dependent variable and the year as the independent variable. Pixels with observations during fewer than five years were discarded from the analysis. Figure 7 shows the regression coefficients for 2BDA and impurity indices, and MODIS albedo vs. time. The corresponding $R^2$ and P-value estimates are shown in Fig. A4. There were two primary regions within our study area that

exhibited significant increases in algal abundance from 2004 to 2011 (Fig. 7a), the DS region in the north and the southern region (68.5°N–66.5°N) between 1200 m and 1400 m in elevation. Other areas do not show statistically significant trends. The interannual trend of the impurity index (Fig. 7b) shows a larger spatial extent with a significant increasing trend as compared with the 2BDA index. Figure 7c shows that the areas with increasing algal abundance and increasing impurity index also had significant albedo (July-August mean) reduction from 2004 to 2011. The albedo reduction was roughly -0.025 to -0.04 per year over the K-transect area (between 1200 m and 1400m in elevation) and within the DS area. The spatial patterns of declining albedo more closely resemble the patterns of impurity index as opposed to the 2BDA index, suggesting that the impurity index quantifies multiple processes related to surface darkening in addition to glacier algae.

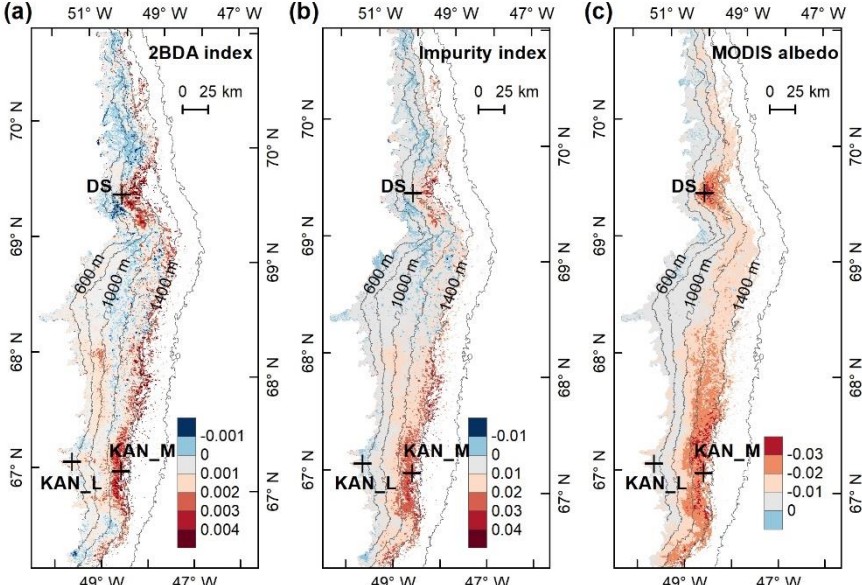

**Figure 7. Interannual trends (regression coefficients with year) of the 2BDA index (a), Impurity index (b), and MODIS albedo (c) from 2004 to 2011.**

### 4.5 Seasonal trends of algal growth over July and August

To better understand seasonal dynamics of glacier algae, we examined intra-annual trends in the 2BDA index during the months of July and August. We estimated the temporal trend of the 2BDA index from July to August for each MERIS pixel. For each pixel and each day, we calculated the average 2BDA index using the same-day 2BDA indices of multiple years. To account for the differences between different years, we applied a temporal smoothing function with a window size of three days to the daily average 2BDA data. Pixels with more than 15 days of observations were kept for linear regression analysis, with the daily 2BDA index as the dependent variable and the time (in days) as the independent variable.

Figure 8 illustrates the pattern of seasonal trends across the southwest Greenland ablation area. Figures 8b and 8c show the spatial distribution of seasonal trends across the area, while Fig. 8a shows examples of the daily 2BDA time series at the

three field sites KAN_L, KAN_M, and DS. At the coastal KAN_L site, which had the lowest algal cell concentration (66 $\pm$ 31 cells/ml), the average 2BDA index is less than 0.98 during July and August, and there is no significant temporal trend. At the KAN_M and DS sites, with higher cell concentrations (5688$\pm$3147 and 10621$\pm$2073 cells/ml respectively), 2BDA values were mostly greater than 0.98, and there were significant increases in the 2BDA index during July and August (of 0.0004 and 0.0007 day$^{-1}$ respectively), suggesting dramatic algal growth. The results indicate that the higher concentrations of algae are associated with a significant increasing trend over the course of a season. Indeed, the highest seasonal trends in the middle ablation area (68.5°N-66.5°N) are found in the 1200 to 1400 m elevation band, also the region of highest 2BDA index.

The time series in Fig. 8a suggest that the period of algal growth at KAN_M occurred primarily between mid-July and mid-August, beginning later than at the DS site. Between 20 July and 20 August, the regression slope was 0.0009 day$^{-1}$ for both DS and KAN_M. This time window is consistent with the rapid algal colonization observed in field (Stibal et al., 2017; Williamson et al., 2018; Yallop et al., 2012; Lutz et al. 2018) and the patterns of temporal variability derived from Sentinel-3 data (Wang et al., 2018). To test whether higher growth rates later in the season were present across the region, we also examined region-wide trends between 20 July and 20 August (Fig. 8c). The magnitude of trends for the shorter period are higher over a broad region, and $R^2$ values are higher, indicating a shorter growth period across much of the region, with the exception of the area around the DS site in the north. We also explored the interannual variability of seasonal patterns over the DS and KAN_M sites (Fig. A5). Despite the interannual variations of 2BDA index, the regression slopes of 2BDA versus time (day) through mid-July to mid-August for different years were comparable to the slope of the aggregated time series between 2004 and 2011, particularly for the KAN_M site. Over the DS site, the algal growth rates were above-average during the growth seasons in 2005, 2009, and 2011 (Fig. A5). The DS site is located in lower elevations, where warmer temperatures may promote a faster growth rate.

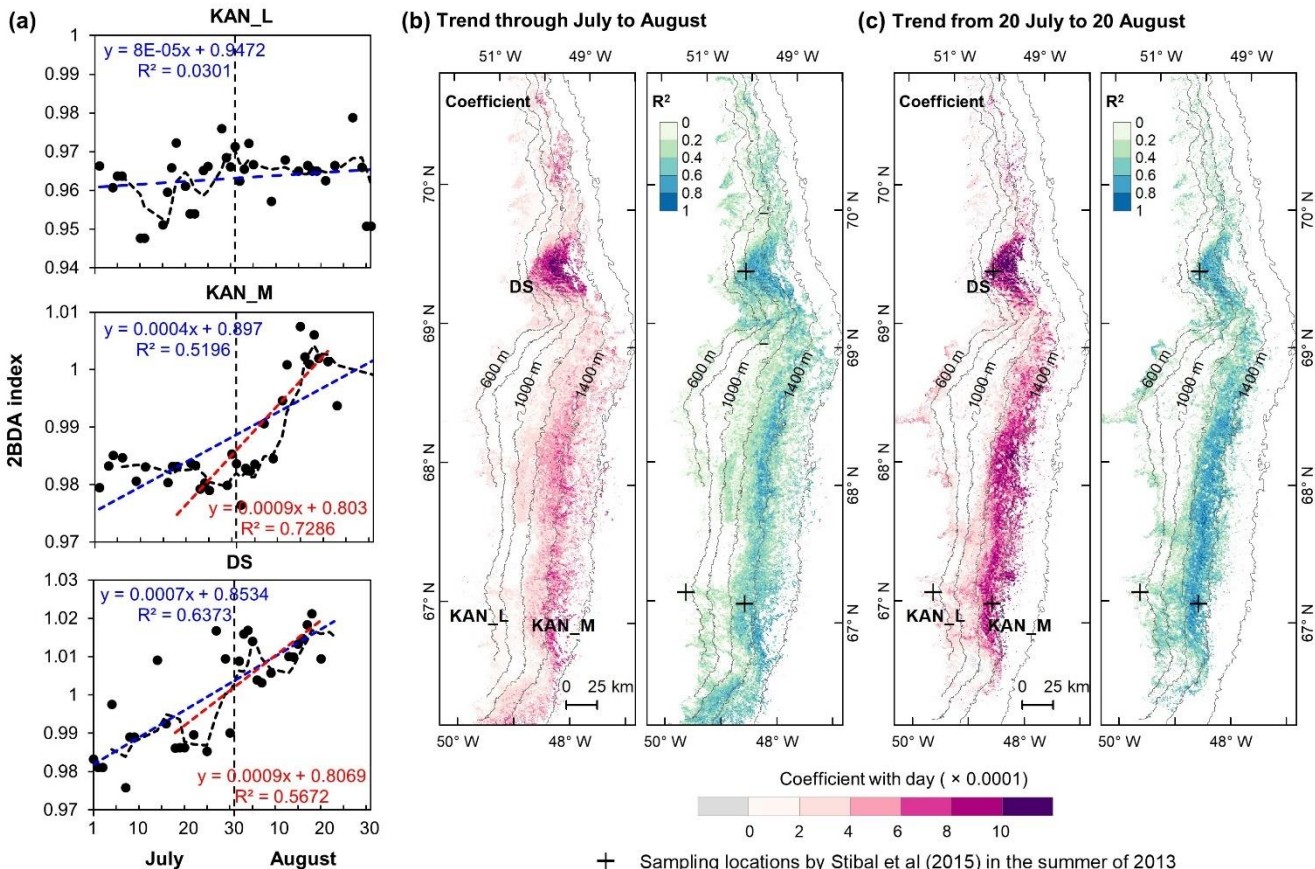

**Figure 8. Temporal trends of the 2BDA index over July and August. (a) 2BDA time series and temporal trend analysis over the**
**KAN_L, KAN_M, and DS sites. (b) Regression slope and R² estimates of the temporal trend analysis for the period of July–August**
**(for areas where the p value <=0.05). (c) Regression slope and R² estimates of the temporal trend analysis for the period of 20**
**July–20 August (for areas where the p value <=0.05).**

To make our estimates comparable to previous studies, we calculated the algal population doubling time from the seasonal
trends in the 2BDA index. Wang et al. (2018) derived an empirical relationship between the Sentinel-3 OLCI reflectance
ratio $R_{709nm}/R_{673nm}$ and the glacier algal abundance (Stibal et al., 2015; Williamson et al., 2018) using an exponential fit. The
empirical relationship is represented as

$$y = 10^{-35} \, e^{87.015x} \tag{3}$$

where $x$ denotes the reflectance ratio and $y$ denotes the algal abundance (cells/ml). Given the similarity between the OLCI
and MERIS band configurations and the negligible differences between the 673 nm and 665 nm reflectance, we used Eq. (3)
to derive the relationship between the MERIS 2BDA index ($x$) and algal population doubling level ($\log_2 y$ cells/ml, i.e. the
number of times the observed number of cells has doubled). Taking the base-2 logarithm of Eq. (3) gives the equation:

$$\log_2 y = \log_2 e * 87.015 * x - 35 * \log_2 10 \tag{4}$$

The algal population doubling time can then be derived by examining the regression coefficient (denoted as *a*) of 2BDA versus time, which gives the rate of change of 2BDA with time (*dx/dt*). From Eq. (4), the rate of change of population doubling with respect to the 2BDA index (*dlog₂y/dx*) is $\log_2 e * 87.015$. Combining *dlog₂y/dx* with the regression coefficient *a* (*dx/dt*) the change in population doubling level with time (*dlog₂y/dt)* is $\log_2 e * 87.015 * a$, and the time for one algal population doubling is therefore estimated as $1 / (\log_2 e * 87.015 * a)$. Values of population doubling time corresponding to various values of *a* are listed in Table 3. The areas with significant algal growth trend ($R^2 > 0.5$) between 20 July and 20 August (Fig. 8c) had a mean regression coefficient of $0.00076 \pm 0.0002$, corresponding to a mean algal population doubling time of $11.2 \pm 2.6$ days. The DS area had faster algal growth rate than other areas, with a doubling time of $9.6 \pm 2.7$ days. This estimate is comparable to the field study by Williamson et al. (2018) who reported a doubling time of $7.18 \pm 1.04$ days for algae-abundant ice (at the K-transect).

**Table 3. Algal population doubling time for different regression coefficients of 2BDA index with day**

| Regression coefficient | Population doubling time (days) |
|---|---|
| 0.0004 | 19.91 |
| 0.0005 | 15.93 |
| 0.0006 | 13.28 |
| 0.0007 | 11.38 |
| 0.0008 | 9.96 |
| 0.0009 | 8.85 |
| 0.0010 | 7.97 |
| 0.0015 | 5.31 |
| 0.0020 | 3.98 |

## 4.6 Impact of glacier algal blooms on surface albedo in July and August

To investigate potential impact of algal changes on albedo variability, we quantified the relationship between glacier algal blooms and surface albedo in July and August based on the daily time series data of the 2BDA index and MODIS broadband albedo. A daily albedo time series obtained by averaging and smoothing the MODIS daily albedo data from 2004 to 2011 was derived using the same method for deriving the 2BDA seasonal trends. Figure 9 shows the derived temporal trends in MODIS albedo from 1 July to 20 August. The days after 20 August were excluded from the analysis since snowfalls often happen in late August. The DS area had the most significant albedo reduction over July and August, up to 0.4~0.6% per day. In the middle ablation zone between the altitudes of 1000 m and 1200 m the albedo reduction rate was 0.2~0.4% per day, and the reduction rate was 0.2~0.3% per day in the zone between 1200 m and 1400 m in elevation.

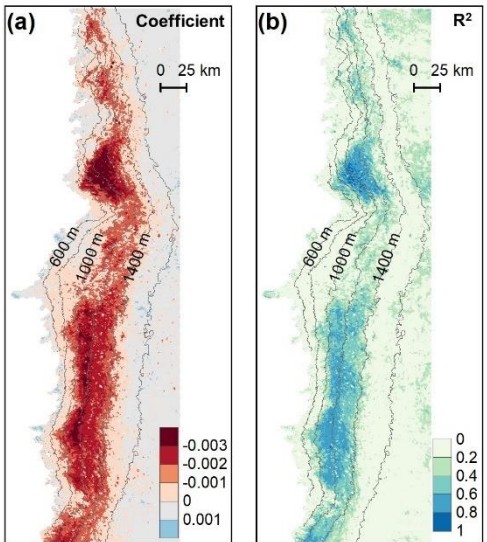

**Figure 9. Temporal trends in MODIS broadband albedo during July and August (over 2004-2011). (a) Regression coefficients of surface albedo with time (day) from 1 July to 20 August. (b) Corresponding $R^2$ estimates.**

We converted the 2BDA index (x) to the algal population doubling level ($\log_2 y$) using the derived Eq. (4) in section 4.5.
Figure 10 shows results of a regression analysis for algal population doubling vs. surface albedo. The analysis shows a statistically significant correlation between algal growth and albedo reduction at the DS area between the altitudes of 800 m and 1200 m, the middle ablation zone between the altitudes of 1200 m and 1400 m, and the 1000–1200 m area nearby the K-transect. Over these areas, the regression coefficient ranged between -0.04 to -0.02, indicating a surface albedo decrease of 2~4% for each algal population doubling. This estimate is comparable to results from the field study by Stibal et al. (2017)
who estimated a net albedo reduction of 0.038±0.0035 for each algal population doubling based on in-situ measurements of glacier algal abundance and coincident surface albedo. In general, glacier algal growth explains most of the temporal variability of surface albedo in July and August between 1000 and 1400 m in elevation, except the middle area between 1000 m and 1200 m in elevation where other factors likely contribute to the observed albedo reduction.

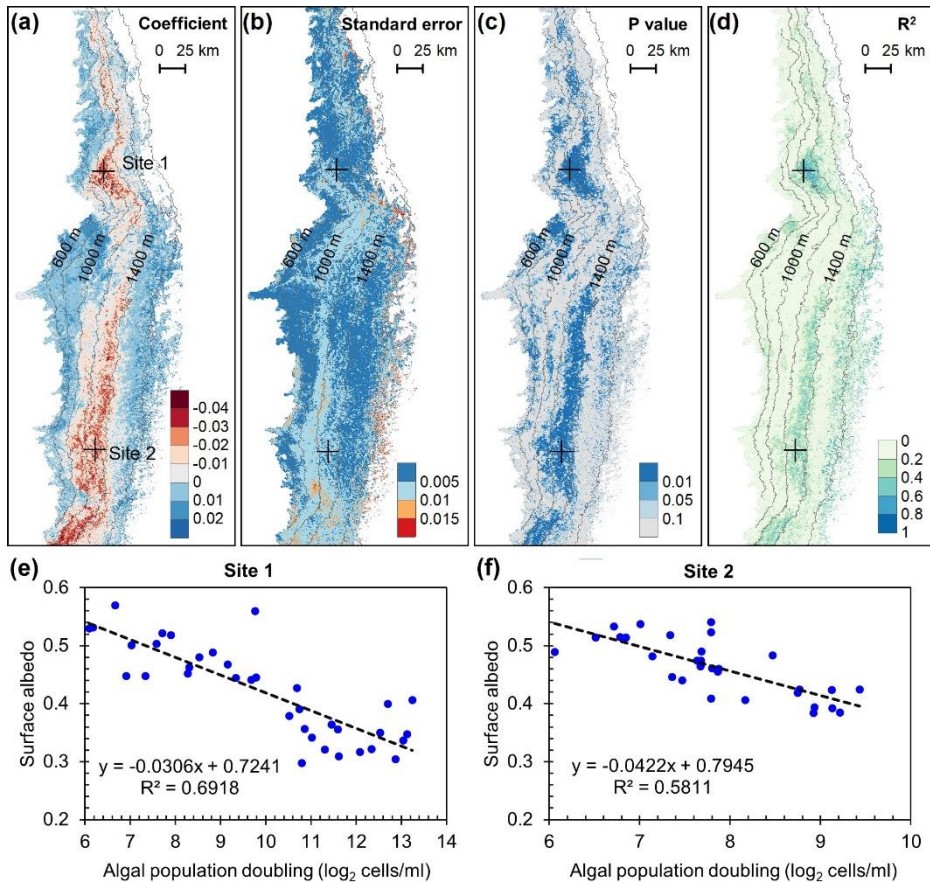

Figure 10. Relationship between algal population doubling and surface albedo. (a) Regression coefficients. (b) Standard errors of the correlation coefficients. (c) P values. (d) $R^2$ values. (e) and (f) show algal population doubling versus surface albedo at Site 1 and Site 2, respectively.

## 5 Discussion

### 5.1 Sensitivity to subpixel variability

In this study, we utilized the chlorophyll-a signal generated by glacier algae in the red-NIR region (Fig. 2d) to quantify the spatiotemporal variability of glacier algae at a regional scale for the summer seasons of 2004-2011 in southwest Greenland. The specific wavelengths and narrow bandwidths of MERIS designed for chlorophyll-a detection make MERIS archive data a powerful tool for studying supraglacial algal communities. The chlorophyll-a signal present in the MERIS spectra is consistent with (nearly) coincident WorldView-2 data and hyperspectral field measurements collected over dark ice with high algal abundance (Fig. 2d). Similar to the Sentinel-3 OLCI ratio index $R_{709nm}/R_{673nm}$, the MERIS 2BDA index $R_{709nm}/R_{665nm}$ can effectively quantify the algal growth pattern during July and August (Fig. 8). Using SNICAR simulations, we examined the potential impact of dust on the 2BDA index. The comparison between SNICAR simulations and MERIS

ratio indices indicates a high sensitivity of the 2BDA index to glacier algae as compared to dust (Fig. 4). Here we explore the sensitivity of the 2BDA index to subpixel variability using a linear mixing method based on the field spectral measurements

of Cook et al. (2020) and the SNICAR-simulated spectra for dust (size 4 with a concentration of 500 ppm). The spectra used for linear mixing experiments are shown in Fig. C3a. By specifying the areal percentage of the impurity-covered (algae or dust) surface at subpixel scale, we calculated the mixed spectra by linearly combining the algae (four samples with different measured algal abundance) or dust spectra (SNICAR-simulated) with the bare ice spectrum (measured algal abundance as 0) weighted by areal percentage. Figure C3b shows the 2BDA index calculated from the mixed spectra varying with the areal

percentage of algae or dust at the subpixel scale. It is shown that the 2BDA index dramatically increases with the areal percentage of glacier algae, being consistent with the positive correlation between the 2BDA index and algal abundance. In contrast, the 2BDA index is much less sensitive to dust areal coverage. The results indicate that even with sub-pixel variability of surface materials, the satellite-derived 2BDA index is still strongly sensitive to the presence of algae. High-resolution UAV mapping by Ryan et al. (2018) suggests that the areal percentage of distributed impurities is up to 65%~95%

within individual MODIS pixels (500-meter resolution) over the dark zone in southwest Greenland, indicating that a high sub-pixel areal percentage of algae is possible. Our linear mixing experiments (Fig. 3Cb) indicate that the relatively high 2BDA values derived from satellite are unlikely to be achieved without the presence of glacier algae, and that the MERIS 2BDA index can effectively capture the glacier algae variability, especially within the dark zone.

## 5.2 Relationship between regional climate model albedo bias and glacier algae

Our analysis suggests that surface albedo decreases by 0.02~0.04 for each algal population doubling during July and August primarily in algae-abundant areas close to the Jakobshavn Isbrae Glacier and within the middle ablation zone (68.5°N-66.5°N) between 1200 m and 1400 m in elevation. It is also important to know whether the MERIS 2BDA index could explain the discrepancy between the satellite-measured albedo and bare ice albedo in climate models which do not currently simulate the effects of biology and other impurities. The MAR regional climate model, for instance, exhibits a positive

albedo bias along the southwestern Greenland ice sheet margin because of this (e.g. Alexander et al., 2014). Figure 11a shows a comparison between MODIS albedo and MAR albedo over the study area (including both bare ice and snow), indicating the overestimation of MAR albedo for the dark areas with MODIS albedo less than 0.5. There is a significant negative correlation between the albedo difference (MODIS albedo minus MAR albedo) and the 2BDA index (Fig. 11b), indicating that the positive MAR albedo bias increases with algal abundance.

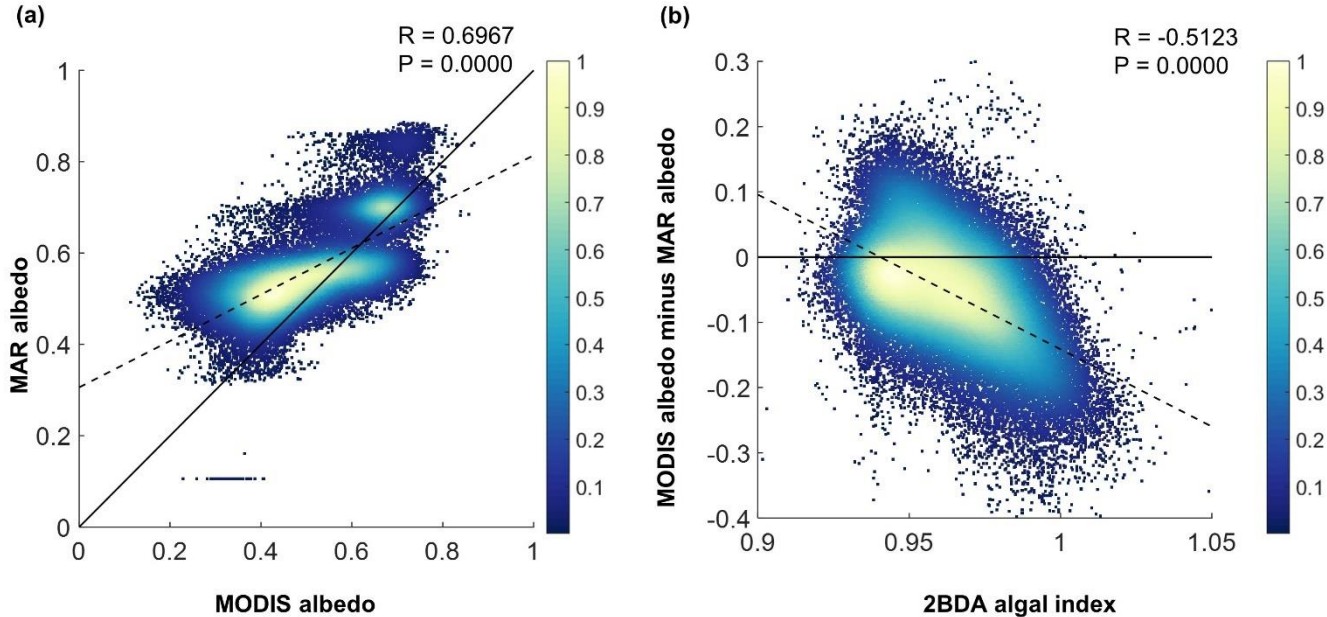


**Figure 11. (a) Comparison between MAR albedo and MODIS albedo over the study area for July and August from 2004 to 2011. The dashed line shows the linear fit between MODIS albedo and MAR albedo. The black line is the 1:1 reference line. (b) Relationship between the MERIS 2BDA index and the albedo difference between MODIS and MAR, with the dashed-line showing the linear fit. The colour scheme in both (a) and (b) illustrates the relative data distribution density (yellow means higher density,**

**and blue means lower density).**

The spatial pattern of the MAR albedo bias (Fig. 12a) is consistent with the satellite-derived impurity distribution (e.g. Fig. 5b). Over the dark areas, the MAR albedo was overestimated by 0.16±0.03 as compared to the MODIS albedo. We further examined the relationship between the albedo bias (MODIS albedo minus MAR albedo) and the algal population for the seasonal trend between 1 July and 20 August, finding a significant correlation in the DS site region (Figs. 12b, 12c, 12d).

Figure 12d indicates that each population doubling can explain a -0.0274 bias between the MODIS and MAR albedos at the DS site. In comparison, the albedo bias in the middle zone between 1000 m and 1400 m is less well-explained by glacier algae. This is partially consistent with our previous analysis that the albedo reduction at 1000 m–1200 m is poorly related to algal growth. Between 1200 m and 1400 m the correlation between the derived algal population and the MAR bias is not strong (Fig. 12c) even though there is a fairly strong correlation between algal population and MODIS albedo. This suggests

that in this area, although MAR does not include the effects of algae, the decrease in albedo associated with liquid water ponding in MAR may approximate the trends associated with increasing algae concentrations. In addition to parameterizing glacier algal growth, other processes related to albedo reduction such as consolidation of impurities melted from snow should be also accounted for in the future.

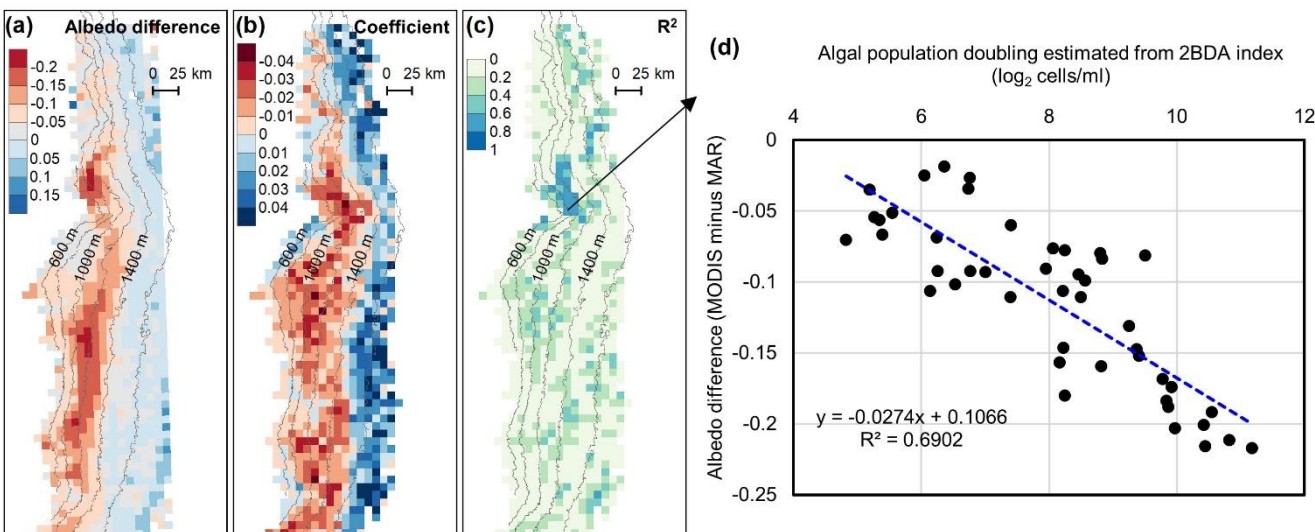

Figure 12. (a) Albedo difference between MODIS albedo and MAR albedo (MODIS albedo minus MAR albedo) averaged over the study period. (b) Regression coefficients of albedo difference with 2BDA-derived algal population doubling ($\log_2$ cells/ml). (c) $R^2$ estimates for the regression analysis. (d) Scatterplot of albedo difference versus algal population doubling over the DS algae-abundant area and the equation for the linear fit.

## 5.3 Potential drivers for glacier algae variability

Due to the impact of glacier algal blooms on bare ice albedo, it is fundamental to understand the factors affecting algal growth. Lutz et al. (2018) analysed the composition of glacier algal communities near the K-transect between 27 July and 14 August 2016 using high-throughput sequencing and subsequent oligotyping techniques. The glacier algae species of Ancylonema nordenskiöldii and Mesotaenium berggrenii were found as the dominant taxa. Glacier algae lack a flagellated stage and are less capable of migrating upwards to snow layers at the beginning of melting season (Anesio et al., 2017). Therefore, glacier algal growth is restricted to the bare ice surface, which is consistent with our finding that glacier algal blooms tend to occur extensively from late-July to mid-August when the bare ice is exposed. However, somewhat paradoxically, the areas at lower altitude have longer duration of bare ice exposure, whereas intense glacier algal blooms occur at higher altitude up to 1200-1400 m along the middle ablation zone. One possible reason for this discrepancy could be that the growth of glacier algae is influenced by liquid water (e.g. Tedstone et al., 2017) and nutrient availability. Although liquid water is a prerequisite for algal growth, Wang et al. (2018) found a negative correlation between algal abundance and meltwater production, which was attributed to hydrological flushing of algae during periods of excessive meltwater and surface runoff (Takeuchi, 2001; Uetake et al., 2010). These results do not contradict the importance of liquid water to algal growth as indicated by Tedstone et al. (2017), but rather suggest that there is an optimal amount of melt that may be required to support algal growth, with too little or too much melt resulting in lower algal concentrations.

To examine potential drivers of algal growth, we explored the relationships between the 2BDA index and topographic variables as well as near surface temperature and meltwater production simulated by MAR (Fig. B1), by separating the data

into two-dimensional bins and calculating the average 2BDA index for each bin. The comparison suggests that glacier algae are mostly distributed over flat areas with fewer topographic undulations (Fig. B1a). The areas suitable for glacier algal growth have moderate but not excessive melting (Fig. B1b). This further supports the hypothesis that high melt has a negative effect on algal development. In regard to the suitable temperature, glacier algae are so far known to be well adapted to temperatures close to 0°C (Anesio et al., 2017). Although no significant correlations have been found between algal abundance and air temperature, Figure B2 shows dips in measured daily algal abundance (Stibal et al., 2017) coinciding with below-freezing near-surface MAR-simulated daily air temperatures at the K-transect S6 station during the 2014 summer, suggesting that freezing temperatures negatively impact algal growth.

We also examined interannual variations in climate variables in relation to the 2BDA index. Figure 13 shows the MAR-simulated shortwave and longwave downward radiation fluxes, cloud cover, snowfall, rainfall, meltwater production, and near surface air temperature averaged over July and August across the study area from 2004 to 2011. The high 2BDA algal index during 2008-2010 (Fig. 5a and Fig. 13a) coincides with reduced cloud cover and higher incoming shortwave radiation (Fig. 13b). This period is also characterized by less rainfall (Fig. 13c), reducing the possibility of hydro-flushing. Figure 13d shows that the high algal index years of 2008 and 2009 exhibited less melting and lower temperature than the other years, suggesting that these variables may play a less important role in algal growth than shortwave radiation. Given the importance of shortwave radiation for photosynthesis of glacier algae, the results suggest that air temperature, surface melt, and bare ice exposure may be important factors at the beginning stage of glacier algal habitat development, while downward shortwave radiation could be most important during the proliferation stage. These dynamics could be influenced by recent atmospheric circulation changes in Greenland, with patterns of anomalous anticyclonic circulation and higher 500 hPa geopotential height becoming more frequent (e.g. Hanna et al., 2016; Mioduszewski et al., 2016), associated with reduced cloud cover (Hofer et al., 2017) and increased downward shortwave radiation. However, more research is required to fully understand these relationships and quantify the effects of various factors on glacier algal growth. In the context of future ice sheet change, it is therefore vital to understand the interactions between the supraglacial microbiome and climate change (Cavicchioli et al., 2019) for better projection of future ice sheet mass balance.

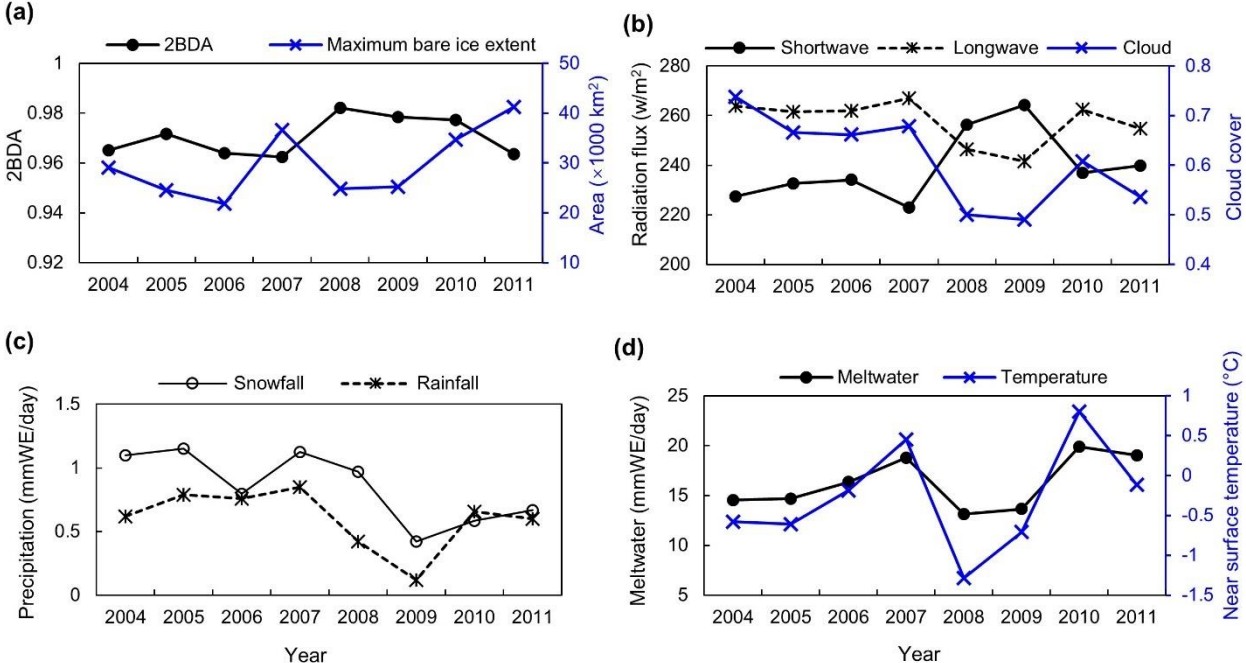

**Figure 13. (a) Average 2BDA index over bare ice and maximum bare ice area from 2004 to 2011 (MERIS). (b) July-August mean of downward shortwave and longwave radiation fluxes and cloud cover over the study area from 2004 to 2011 (MAR). (c) July-August mean of rainfall and snowfall (MAR). (d) July-August mean of meltwater production and near surface temperature (MAR).**

## 6 Conclusions

We examined the spatiotemporal variability of glacier algal blooms in southwest Greenland during July and August from 2004 to 2011 using the chlorophyll-a detection capability of MERIS. We calculated a number of remote sensing ratio indices including chlorophyll-a indices and the impurity index. The results indicate that similar to the Sentinel-3 OLCI ratio index of $R_{709nm}/R_{673nm}$, the MERIS 2BDA index of $R_{709nm}/R_{665nm}$ can effectively quantify the spatial distribution and seasonal growth pattern of glacier algae, with results highly consistent with field measurements. There was an increasing trend of glacier algal abundance and impurity content at the dark area close to Jakobshavn Isbrae Glacier and the area close to K-transect at altitude of 1200 m–1400 m, in conjunction with a declining trend of surface albedo over the 2004 to 2011 period. We quantified the impact of glacier algal growth on surface albedo over July and August, and found that each algal population doubling decreases the surface albedo by 2~4 percent. Our analysis points to the great potential of using satellite ratio indices to parameterize the impact of glacier algae on surface albedo, thereby reducing the albedo bias in regional climate models. Nevertheless, the surface darkening along the middle ablation zone between 1000 m and 1200 m in elevation cannot be well explained by algal growth, indicating that other processes related to surface darkening need further investigation and

quantification. Future research should also be directed toward understanding the climate drivers of glacier algae variability
and parameterizing their growth dynamics using regional climate model outputs.

**Data availability**

MERIS level-2 data are available at the MERCI file archive (https://merisfrs-merci-ds.eo.esa.int/), courtesy of the European Space Agency. MODIS MOD09GA and MOD10A1 data can be accessed from the NASA Land Processes Distributed Active Archive Center (https://search.earthdata.nasa.gov/). WorldView-2 imagery were provided by the Polar Geospatial
Center (PGC, https://www.pgc.umn.edu/) at the University of Minnesota. MAR v3.9.3 outputs are available at ftp://ftp.climato.be/fettweis/.MARv3.9.3.

**Appendix A**

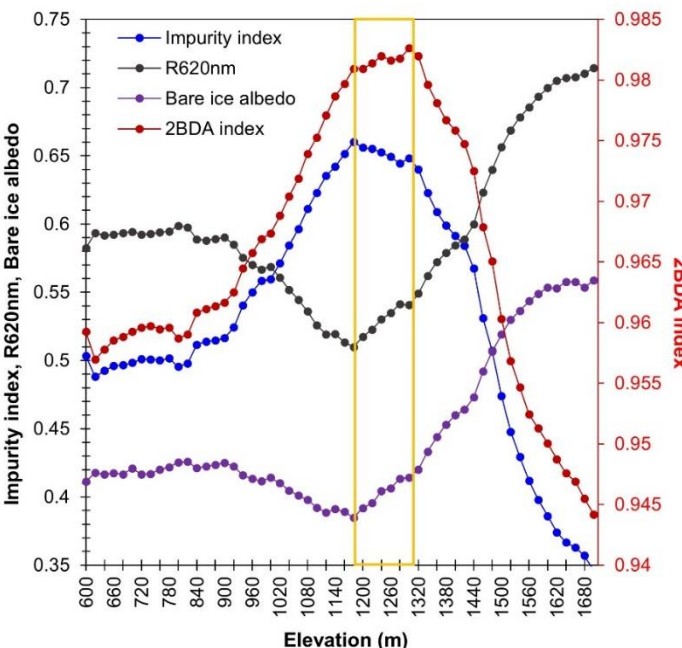

**Figure A1. Spatial variations of the average 2BDA index, impurity index, 620 nm reflectance, and MODIS albedo over bare ice at
different elevations within the study area (20-meter elevation interval). For surface elevation, we used the 30-meter resolution
MEaSUREs Greenland Ice Mapping Project (GIMP) Digital Elevation Model (Howat et al., 2014; 2015).**

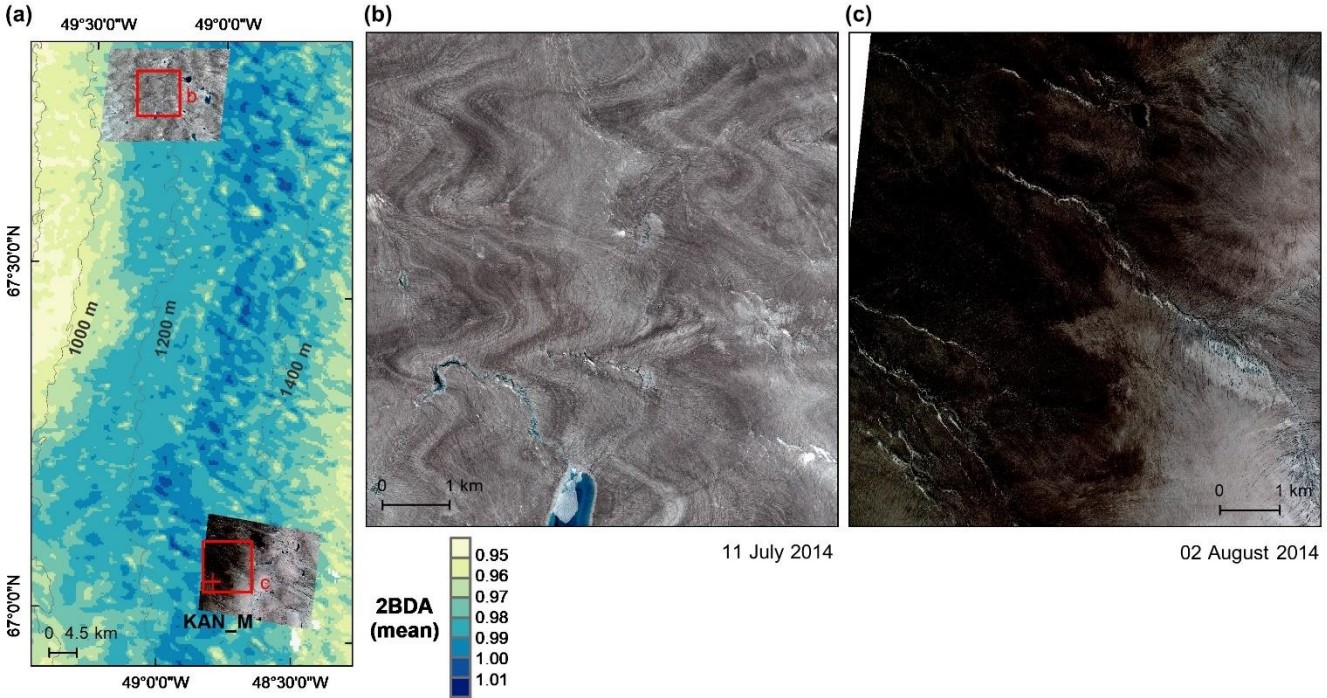

**Figure A2.** Average 2BDA index (2004-2011) for a subset of our study area (a) and comparison between WorldView-2 imagery over a dark ice site with low 2BDA index at 1000-1200m elevation (b) and a dark ice site with high 2BDA index at 1200-1400m elevation (c). The WorldView-2 image in (b) illustrates the 'wavy' pattern that Wientjes and Oerlemans (2010) suggested was caused by ancient ice outcropping.

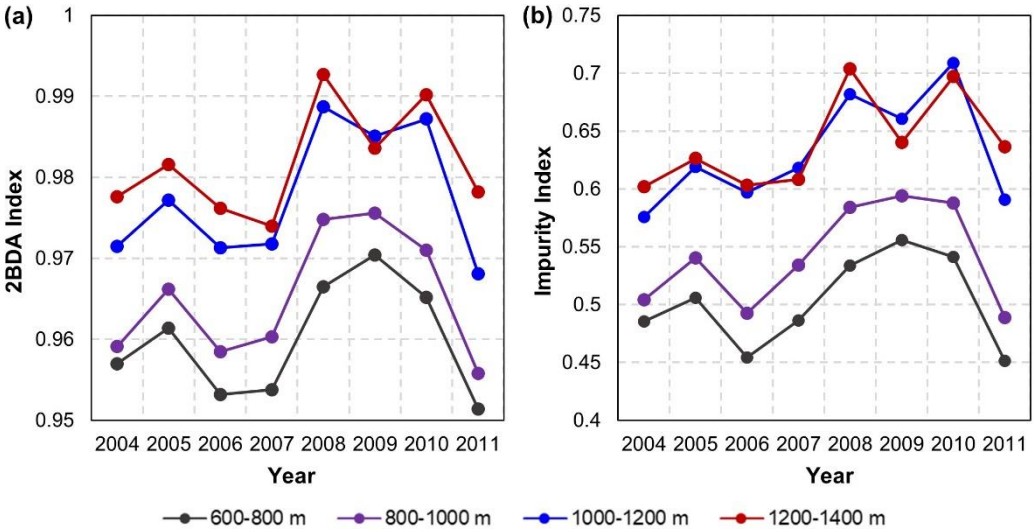

**Figure A3.** Interannual variability of the 2BDA index (a) and impurity index (b) at the elevation levels of 600-800m, 800-1000m, 1000-1200m, and 1200-1400m within the study area.

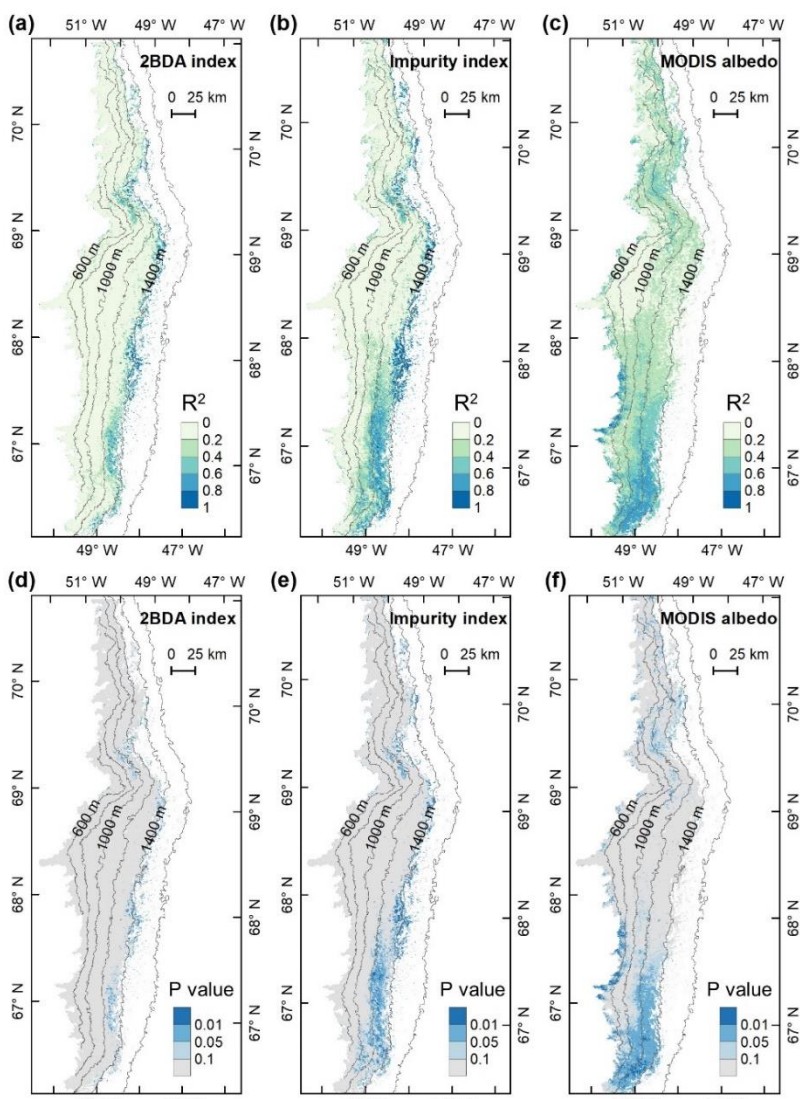

**Figure A4. R² and P values for the interannual trends of the 2BDA index, impurity index, and MODIS albedo from 2004 to 2011.**


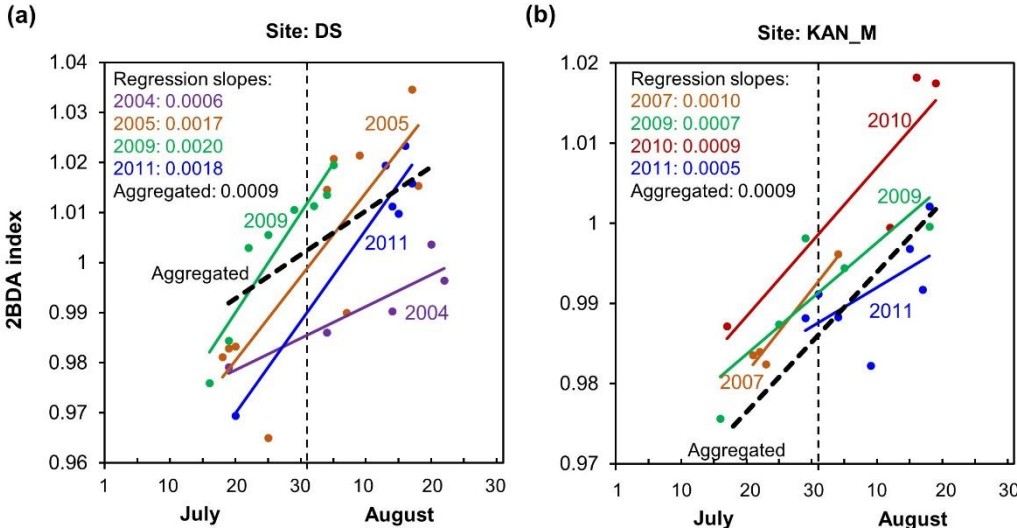

**Figure A5.** Temporal trends of 2BDA index from mid-July to Mid-August in different years at sites DS (a) and KAN_M (b).

## Appendix B

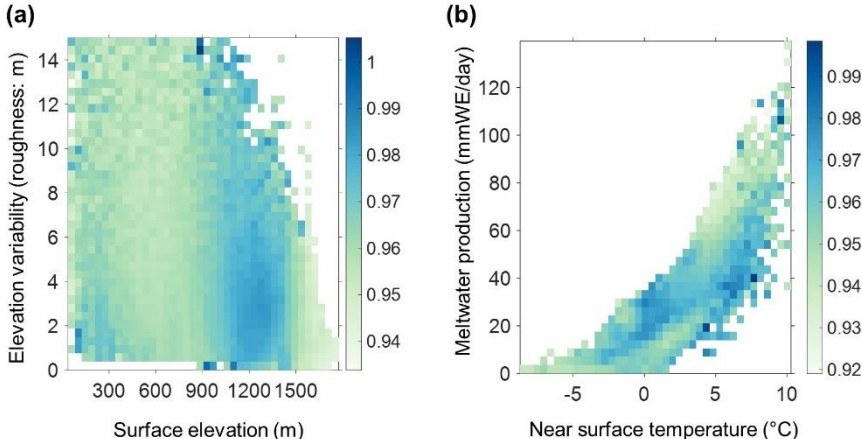

**Figure B1. (a) 2BDA index versus surface elevation and roughness (elevation variability within each MERIS pixel). (b) 2BDA index versus near surface temperature and meltwater production simulated by MAR. The colour bars in (a) and (b) indicate the average 2BDA index for each two-dimensional bin defined by the two variables on the horizontal and vertical axes.**

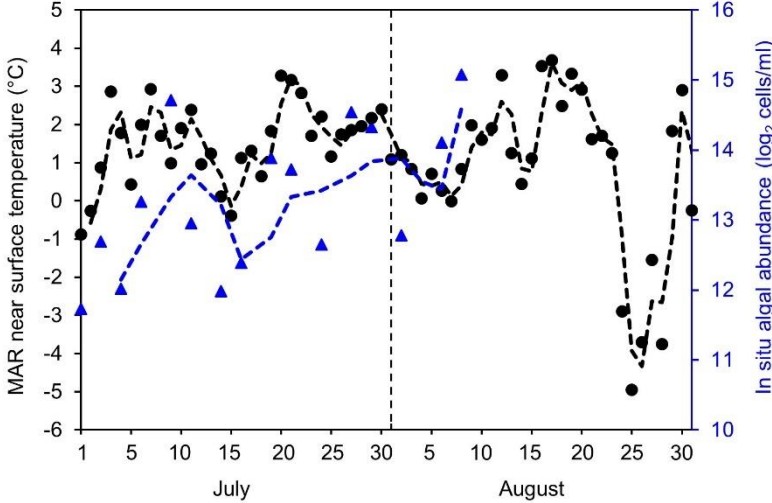

**Figure B2. MAR-simulated near surface temperature (°C, black circle, left axis) and in situ measured algal abundance ($\log_2$ cells/ml, blue triangles, right vertical axis) at the S6 weather station at the K-transect during July-August 2014 by Stibal et al. (2017).**

## Appendix C

**Table C1. Measured algal cell abundance from the field dataset of Cook et al. (2020) with the 2BDA index calculated from coincident hyperspectral measurements. The highlighted rows are samples with cell abundance of greater than 10000 cells/ml.**

| Sample ID | Cell abundance (cells/ml) | 2BDA index |
|---|---|---|
| 13_7_SB1 | 2688 | 0.9614 |
| 13_7_SB2 | 13375 | 1.0075 |
| 13_7_SB3 | 938 | 0.9813 |
| 13_7_SB4 | 4500 | 1.0371 |
| 13_7_SB5 | 0 | 0.9653 |
| 14_7_SB1 | 30313 | 1.0953 |
| 14_7_SB2 | 3063 | 0.9868 |
| 14_7_SB3 | 7938 | 0.9554 |
| 14_7_SB4 | 17938 | 0.9939 |
| 14_7_SB5 | 41000 | 1.2218 |
| 14_7_SB6 | 0 | 0.9555 |
| 14_7_SB7 | 12438 | 1.0850 |
| 14_7_SB8 | 0 | 0.9863 |
| 14_7_SB9 | 21875 | 1.0808 |
| 14_7_SB10 | 24875 | 1.1257 |
| 15_7_SB1 | 1438 | 0.9908 |
| 15_7_SB2 | 7250 | 0.9497 |
| 15_7_SB3 | 30313 | 1.0810 |
| 15_7_SB4 | 4250 | 0.9665 |
| 15_7_SB5 | 938 | 0.9839 |
| 20_7_SB1 | 11375 | 1.0536 |
| 20_7_SB2 | 7563 | 0.9939 |
| 20_7_SB3 | 7625 | 1.0122 |
| 21_7_SB1 | 92250 | 1.2635 |
| 21_7_SB2 | 44861 | 1.1411 |
| 21_7_SB3 | 750 | 0.9922 |
| 21_7_SB4 | 14313 | 1.0296 |
| 21_7_SB5 | 1063 | 0.9731 |
| 21_7_SB7 | 33229 | 1.0794 |
| 21_7_SB8 | 1188 | 0.9440 |
| 21_7_SB9 | 313 | 1.0097 |
| 21_7_SB10 | 17563 | 1.0763 |
| 23_7_SB1 | 250 | 0.9765 |
| 23_7_SB2 | 938 | 0.9754 |
| 23_7_SB3 | 8563 | 1.0141 |
| 23_7_SB4 | 21125 | 0.9975 |
| 23_7_SB5 | 28563 | 1.1131 |
| 24_7_SB1 | 1875 | 0.9985 |

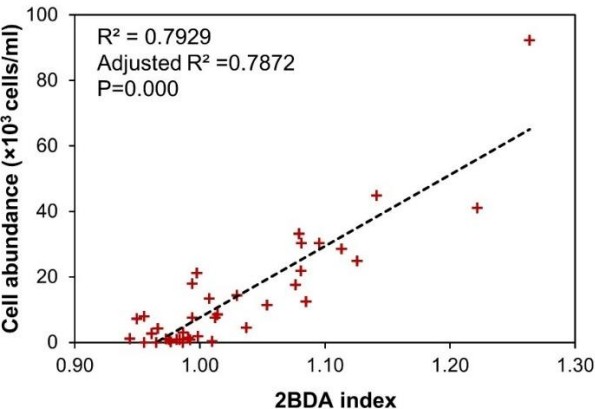

**Figure C2. Scatterplot of measured cell abundance versus 2BDA index listed in Table C1.**

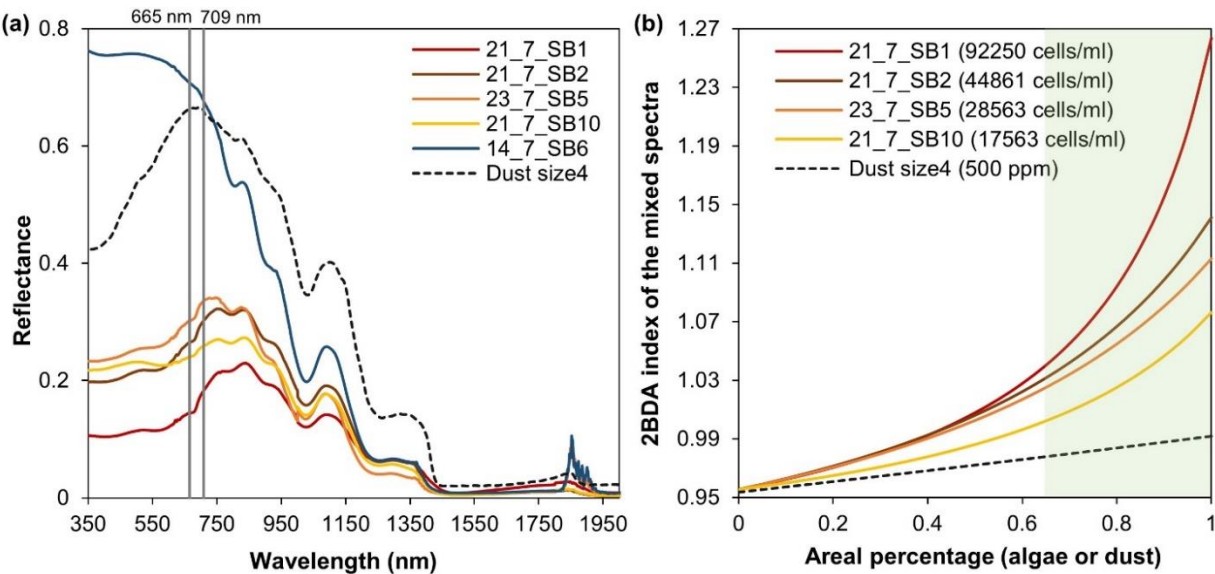

**Figure C3. Spectral linear mixing experiments. (a) Field hyperspectral measurements of four algae-abundant samples (21_7_SB1, 21_7_SB2, 23_7_SB5, and 21_7_SB10) and one bare ice sample (zero algal abundance, 14_7_SB6) from Cook et al. (2020), and the**
**SNICAR-simulated spectra for the dust scenario (size 4 at concentration of 500 ppm). (b) 2BDA index calculated from the linearly mixed spectra with varying areal percentage at subpixel scale for algae (different algal abundances) and dust scenarios.**

## Author contribution

S. W., M. T., and P. A. designed the study. S. W. processed the MERIS, MODIS, and WorldView-2 data. S. W. and M. X.
tested the MERIS ratio indices. X. F. provided the MAR v3.9 outputs. S. W., M. T., and P. A. analysed the results and generated figures. S. W. wrote the manuscript. All authors discussed the results and contributed to the final manuscript.

## Competing interests

The authors declare that they have no conflict of interest.

## Acknowledgements

This work was supported by National Science Foundation ANS #1713072, National Science Foundation PLR-1603331, NASA Exobiology award #80NSSC18K0814, NASA MAP #80NSSC17K0351, NASA #NNX17AH04G, and the Heising-Simons foundation. We would like to thank the Polar Geospatial Center (https://www.pgc.umn.edu/) for providing the WorldView-2 imagery, the European Space Agency for distributing the MERIS data, and the NASA Land Processes Distributed Active Archive Center for distributing the MODIS data. Thanks to David Porter (Lamont-Doherty Earth
Observatory, Columbia University) and Rafael Antwerpen (Utrecht University) for providing comments on the final manuscript.

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
