# Peer review of "Quantifying spatiotemporal variability of glacier algal blooms and the impact on surface albedo in southwest Greenland"

_The Cryosphere, 2019_

## Short Comment (SC1) · 11 Nov 2019

Dear Authors,

Thank you for this valuable contribution. I'd like to give a comment about the algae playing a role in albedo change: I think the term "ice algae" is incorrect, because this refers to Diatoms living on the bottom of icebergs in sea water environment. However, the algae living on melting freshwater ice-surfaces should be called "glacial algae": The main prominent members belong to the genera Ancylonema, Mesotaenium or Cylindrocystis, and they are not related to diatoms at all.

Line 34, I'd change "and" to "or", because this list of geographical regions where snow and glacial algae occur (according to Anesio et al. 2017) is not complete. For example, the Andes in South America provide a lot of suitable habitats but are missing.

All the best!

---

## Referee Comment (RC1) · Anonymous Referee #1 · 2 Dec 2019

*Wang et al. (2019): Quantifying spatiotemporal variability of ice algal blooms and the impact on surface albedo in southwest Greenland*

**Overall:**

The authors have presented a paper that attempts to quantify biomass over the western Greenland Ice Sheet using the well-known "red-edge" technique that they refer to as "2BDA", which is often used for detecting chlorophyll-containing biota such as photosynthetic algal blooms in oceanic and lacustrine environments, vegetation and crop mapping. Biomass quantification over the Greenland Ice Sheet is a worthy research goal because Greenland Ice Sheet glacier algae very likely play an important role in controlling the ablation zone albedo that is not yet accounted for in energy balance models. This is well within the scope of The Cryosphere and the scientific question is worthy of consideration in this journal. There are some very useful aspects to the paper, including demonstration that there are ablation zone albedo processes that SMB models currently do not account for, the albedo time series over the western ablation zone, and the comparison between different band ratios. However, there are some major issues that need to be addressed before I can recommend publication. More detail is provided below.

**Major Comments:**

1) There is past literature that emphasises the importance of discounting abiotically-generated red-edge signals before assuming them to be diagnostic of photosynthetic life (see Sparks et al. 2009; Seager et al. 2005). The vulnerability of the red-edge to false positives is demonstrable using a simple radiative transfer model (easily replicated in-browser via http://snow.engin.umich.edu/). Fig 1A shows the results from SNICAR runs where all variables were held constant except the mass concentrations of completely inorganic impurities (Flanner et al.'s (2009) "global average dusts, type 4"). The lowest 2 spectral albedo curves returned a false positive 2BDA result (1.002 and 1.005). In Figure 1B, the model is run again identically but with a hematite-rich mineral dust taken from Polashenski et al. (2015), giving eight 2BDA false positives (1.004, 1.015, 1.026, 1.031, 1.039, 1.043 1.046). Tedesco et al. (2013) suggested that hematite-rich red dusts are present in the GrIS ablation zone (we note that Cook et al. (*in review*) disagreed about that but their paper remains unpublished). Taking Tedesco et al. (2013) to be correct about the prevalence of red dusts on the Greenland ablation zone therefore invalidates the red-edge as a biomarker due to the demonstrable potential for false positives. Convincing empirical data is required to demonstrate that the 2BDA signal is exclusively biological and robust to these types of false positive results.

A)
B)

[Figure]

*Fig 1: A) SNICAR runs with diffuse irradiance, homogenous snow with grain diameter 500 micron, density 400 kg m$^{-3}$ and Flanner et al. (2009)'s "dust 4" in the upper 1 mm, in mass concentrations of 0.1, 0.3, 0.5, 0.8, 1.0, 1.5, 2.0, 2.5, 3.0 ug$_{dust}$/g$_{ice}$. B) Identical SNICAR runs but with Polashenski et al. (2015)'s high hematite dust.*

2) The authors do not account for the spectral albedo of the ice itself. Ice albedo can vary dramatically independently of light absorbing particles and cause the 2BDA value to change,

undermining the biomass quantification. Figure 2 shows identical simulations to Fig 1, except the grain size is increased to 1500 microns. False positive results are returned as before, but the value of the 2BDA indexes - and therefore the retrieved biomass – change (Flanner et al. (2009) dust = 1.0009 and 1.004; Polashenski et al. (2015) dust = 1.0018, 1.014, 1.026, 1.032, 1.040, 1.043, 1.045, 1.046). The retrieved biomass therefore changed without any change in impurity loading. On real glacier ice where the ice albedo can vary by tens of percent independently of impurity concentration due to weathering crust development, meltwater accumulation and drainage, topography, impurity mixing and glaciological structure, the potential for highly error-prone retrievals is likely very high. The authors need to demonstrate that their band ratio is not vulnerable to this uncertainty, or that they can quantify and correct for it.

A)                                                    B)

[Figure]

*Fig 2: A) SNICAR runs with diffuse irradiance, homogenous 1500 micron, 400 kgm-3 snow with Flanner et al. (2009)'s "dust 4" in the upper 1 mm, in mass concentrations of 0.1, 0.3, 0.5, 0.8, 1.0, 1.5, 2.0, 2.5, 3.0 ug$_{dust}$/g$_{ice}$. B) Identical SNICAR runs but with Polashenski et al. (2015)'s high hematite dust.*

3) The authors do not adequately address the problem of scale mismatches between the normal length scales of typical algal blooms (biomass varies dramatically over 1-10 m length scales) and the satellite used to gather their data (300 x 300m). Surface heterogeneity must surely introduce major uncertainties as the spectral reflectance of each pixel is the combined product of many highly variable surfaces. How would, for example, cryoconite, surface water, dust, crevasses and surface topography at the sub-pixel scale affect their biomass quantification? These unquantified factors must influence the predicted cell concentration independently of real-world changes to the mass concentration of algae, but they are not discussed in the paper.

4) There is insufficient detail regarding the use of field spectroscopic measurements as "ground truth". Only one single field spectrum is presented in the paper and the measurement conditions are not reported. Has it been picked because it matches well with the MERIS spectrum, is it a mean (in which case of how many samples, and what do the error bars look like) or is it the only available spectrum? How do other spectra in the field dataset compare? Can the authors provide evidence to suggest that centimeter scale field spectroscopy measurements are truly representative of the biomass over entire MERIS pixels?

5) The authors are selective with their citing of literature under review. If they wish to include papers still under discussion they should explain why the issues of spatial scale encountered in 20 m Sentinel-2 pixels discussed by Cook et al (in review) and Tedstone et al. (in review) do not also apply to their 300 m MERIS pixels. If they decide to stick to published literature they should explain how the presence of hematite-rich dusts on the ablating Greenland Ice Sheet as reported by Tedesco et al. (2013) does not invalidate their assumption that the red-edge is uniquely biological.

**Specific Comments:**

Title: As suggested by Daniel Remias in the open discussion forum, please adopt the generally accepted terminology "glacier algae" that distinguishes these algae from those found in sea ice.

General point about chlorophyll: Referring to "chlorophyll" is somewhat ambiguous as it could imply total chlorophyll or one of several chlorophyll variants. Please be specific that you mean chlorophyll-a.

L39: Any citation for yellow/orange snow algae? They are normally thought of as green or red.

L71: The authors rightly criticise carotenoid-based remote sensing methods because of possible false positives due to "dirt" but ignore the potential for equivalent red-edge false positives due to dust.

L75: The authors claim chlorophyll is the appropriate pigment to use to identify ice algae despite also stating that the coloration of the algae is primarily due to purpurogallin pigments. Why, then, is that not the appropriate pigment to use to identify glacier algae?

L75: "owing to its unique spectral signatures between 665-710 nm (Gitelson, 1992; Painter et al., 2001; Wang et al., 2018)": Chlorophyll-a absorbs in narrow bands around 680 nm and 440 nm. Any effects extending up to 710 nm are due to interactions with the surrounding medium. This is why Painter et al. (2001) was able to use the narrow 680 nm absorption feature as a diagnostic tool for chlorophyll-a detection.

L79: "Quantification of ice algae biomass from satellite data based on the chlorophyll-a feature has received less attention since the chlorophyll-related satellite bands designed for land generally have coarse spectral resolutions." This is just one of many reasons why remotely detecting algae over glacier ice is not simple. Other complexities include the complex pigmentation of the algae, the spatial resolution of the remote sensing instruments relative to the typical length scales of individual surface features (including algal blooms) and critically the optics of the underlying ice that vary dramatically over space and time and which are not yet well described. These issues are as important as the spectral resolution and must be acknowledged.

L100 – 120: Much more detail is required here. For example, how many actual field samples were used to validate your remote sensing retrievals? What were the biomasses measured at those sites? What were the measurement conditions? On which dates were spectra available at which sites? Were the measurement times conistent and how do they compare to the satellite overpass times? What was the sensor footprint size for the field measurements and how were these upscaled to the satellite pixel scale?

Figure 1: Please provide details of the field spectrum presented as the dashed line. Where/when was it collected and how does it compare to other field spectra presented in this paper?

L209: Chlorophyll-a is the primary photosynthetic pigment, but not the primary light-absorbing pigment. In both the studies you have cited the chlorophyll-a absorption feature is actually extremely subtle – in fact in Cook et al. (in review) it was only really discernable in the derivative spectra and indistinguishable in the raw reflectance. In Stibal et al. (2017) the spectrum are presented with a very truncated y-axis to make the pigment feature discernable.

L211: "pure ice" has lower reflectance at red wavelengths compared to shorter wavelengths.

L216-220: This line of reasoning borrows heavily from studies of chlorophyll-a dominated species in other environments and still requires the red-edge to be validated over glacier ice where LAP and meltwater mixing, complex pigmentation and ice optics are potential confounding variables.

Figure 2: How did the authors select the field spectrum to plot on this figure? Is this the average of all available? If so please provide error bars and number of observations. Also, 184 cells/mL reported in the legend is a tiny amount of algae, unlikely to explain the albedo reduction observed – is this a typo? What was the mineral dust type and concentration in the same area – could it also explain the red-edge? How much of the albedo reduction can be attributed to the algae and how much to melt water/dust? If the absorption is mostly due to chlorophyll-a as the authors suggest, why is the absorption maximum outside of the chlorophyll-absorption range shown in Fig 2c and why does it extend across the visible wavelengths? Why do the field spectra and remotely sensed spectra diverge below ~640 nm?

Figure 3: A) "Agust" → August; B) The authors present spectra for "dark ice (more chlorophyll)" and "dark ice (less chlorophyll)". However, there does not seem to be any positive 2BDA signal in the latter spectrum at all. Is it actually "dark ice (no chlorophyll)"? If so, there are additional darkening processes occurring on the ice. What processes are darkening the ice in those areas and to what extent do those ice darkening processes also influence the biomass retrievals in areas where there is a positive 2BDA result? What effect does this have on retrieved biomass? What is the detection limit for the 2BDA method?

L413: Cook et al. (in review) mention that a red-edge signal was present in most of their algal hyperspectral data but they do not mention false positive rates and they opted not to use that method for their spatial upscaling. It would therefore be useful to know the false positive rate in the present study and how it scales to 300m MERIS pixels.

L416: It is not clear to me from the manuscript precisely how you have inferred algal cell abundance. Please provide further methodological details.

L450-460: Another explanation for this is that the overall ice albedo is lower, there may be smoother ice and more water at the surface, and rather than there being less algae, the red-edge signal is simply erased by an overall dampening of the spectrum across all wavelengths (i.e. putting dark impurities on dark ice has a less detectable effect that putting the same impurities on otherwise bright ice). Can the authors demonstrate that this is not the case?

**References:**

Cook et al. (in review) Glacier algae accelerate melt rates on the south western Greenland Ice Sheet, The Cryopshere, https://www.the-cryosphere-discuss.net/tc-2019-58/#discussion

Painter, T. H., Duval, B., and Thomas, W. H.: Detection and quantification of snow algae with an airborne imaging spectrometer, Appl. Environ. Microbiol., 67, 5267–5272, https://doi.org/10.1128/AEM.67.11.5267-5272.2001, 2001.

Tedesco, M., Foreman, C., Anton, J., Steiner, N., Schwartzman, T.: Comparative analysis of morphological, mineralogical and spectral properties of cryoconite in Jakobshavn Isbræ, Greenland, and Canada Glacier, Antarctica. Annals of Glaciology, 54(63), 147-157. doi:10.3189/2013AoG63A417, 2013.

Tedstone, A. J., Cook, J. M., Williamson, C. J., Hofer, S., McCutcheon, J., Irvine-Fynn, T., Gribbin, T., and Tranter, M.: Algal growth and weathering crust structure drive variability in Greenland Ice Sheet ice albedo, The Cryosphere Discuss., https://doi.org/10.5194/tc-2019-131, in review, 2019.

---

## Referee Comment (RC2) · Anonymous Referee #2 · 3 Dec 2019

Overview

This manuscript uses data from the MERIS satellite sensor to seek to quantify glacier algae bloom dynamics over the south west Greenland ablation zone. They justify their use of this sensor for detecting algal blooms by reference to their previous work using the very similar Sentinel-3 OLCI on the same topic (Wang et al, 2018), by selected references to some field observations, and by wider reference to remote sensing of ocean-borne algal blooms.

Major comments

The manuscript tests several remote sensing ratio indices and shows that, to some

extent, the 2BDA approach retrieves a different signal to that obtained by the 'bulk' Impurity Index or simple red band threshold approaches. This is a useful exercise in seeking to understand what signals can be retrieved from the MERIS/OLCI sensors.

As the manuscript is presented currently, I have some major concerns which prevent me from recommending publication in The Cryosphere.

There are known problems with seeking to apply band ratios/indices designed for chlorophyll-a retrieval from water bodies but which this manuscript does not engage with. I appreciate that the main studies which highlight these problems, by Cook et al. (TCD) and Tedstone et al. (TCD), are currently undergoing review and so were unlikely to have been available at the time when this study was started. But nevertheless, there is a lack of discussion of the wider literature on this issue; instead, Wang et al (2018) is cited as proof that chlorophyll-a-focused band ratios are appropriate for detecting glacier algae blooms, but I find that discussion of these problems is lacking there too, and so their 2018 paper is not an especially strong foundation on which to base the present study.

As I understand it, the core of this problem is two-fold: (1) the optics of bare ice are insufficiently well understood to be able to guarantee that the reduction in reflectance around 667 nm compared to 710 nm is uniquely biological; and (2) other light absorbing impurities may interfere or present the same signal. Thus, based on published field evidence, there is little evidence that the band ratio approach is uniquely biological. Cook et al. (TCD) and Tedstone et al. (TCD) have more information on this and note that phenolic compounds for in the dominant glacier algae species can obscure potentially diagnostic spectral features. This being the case, NDCI etc may simply be measuring some combination of slightly different surface characteristics to the Impurity Index approach, rather than yielding information specifically on glacier algae growth. Thus, regarding inter-annual mapping of 'dark ice' vs glacier algae, there may be little advance on Shimada et al. (2016) or Tedstone et al. (2017), both of whom considered inter-annual variability in 'dark ice' dynamics over the timescales addressed here.

On justification of the 2BDA, Wang et al. (2018) point to Painter et al. (2001) as evidence that glacier algae can detected using chlorophyll-a indices. However, Painter et al refers to the specific case of snow algae growing on snow surfaces, which is not relevant here as this study engages only with bare ice surfaces. Thus, retrievals in this study can in fact be based only on paired cell counts and field spectra acquired by Stibal et al. (2017), a study which also indicates that chlorophyll-a-based approaches could be useful for remote sensing. However, the spectra that Stibal et al (2017, Fig. 3) present refers only to high algal abundance ice, over centimetres patch scales, which is not representative of OLCI or MERIS 300 m data. Some consideration of the scale mismatch is therefore required.

Possibly a more minor concern: the cell counts used as field validation in this manuscript are very high, at 105 cells ml (Figure 2d), but I'm not sure that we would expect to see such high counts over these larger spatial scales (e.g. Williamson et al., 2018, FEMS). Furthermore, the field spectra seem to have quite high reflectance for the quoted cell counts compared to other field spectra in the literature, e.g. Figure A1 in Tedstone et al. (2017, TC). The field spectra shown here seems to be that in Stibal et al. (2017, GRL, Figure 3), but a cell count is not quoted there and so I raise this question here in case there has been an error in transforming Stibal et al's data for this study.

The study also presents data that undermines its application of a Chlorophyll-a based band ratio approach. Figure 3b shows some averaged MERIS surface reflectance curves. Dark Site (Less Chlorophyll) has higher reflectance at 665 than 709 nm and so with 2BDA this site would presumably diagnose as 'clean ice' by comparison to the Clean Ice spectrum plotted above it. I do not see any comment upon this issue elsewhere in the text.

I'm very confused about how the algal population doubling times were calculated. This is a critical part of the manuscript as it underpins the assertion that there is a 0.02-0.04 reduction rate in albedo for each algal population doubling.
Overall, I would urge nuanced engagement with the question of how confident can we be that the differences between 2BDA, Impurity Index and Dark Ice metrics are due solely to algae and not to other processes that might affect this band ratio? I suggest that this needs much clearer explanation in the methods about how Stibal et al's field data were used in this manuscript, and some nuanced discussion of the uncertainties surrounding Chlorophyll-a indices on ice surfaces. If these issues are addressed then the revised manuscript may be suitable for publication.

Minor comments

I agree with the short comment by Daniel Remias that this manuscript should use the terminology 'glacier algae' in preference to 'ice algae'.

The introduction includes wide-ranging references to both glacier algae and snow algae. Detailed discussion of the snow algae literature is not relevant here as this study focuses only on bare ice surfaces, so the introduction would benefit from being focussed solely on glacier algae.

P3 L71: define what is meant by 'dirt'.

P8 L209. Cook et al. (2019, Cryosphere Discussions) are cited for the first time here. If it is being cited then it should be introduced earlier during the lit. review section of the Introduction. Alternatively, if taking the view that Cook et al is under discussion and that it isn't 'referenceable', then all references to it should be removed.

P9 L226: please quantify how the 'best' means of quantifying ice algae was obtained. This is not clear, either here or in the subsequent text.

P9 L229: Dumont et al. (2014) focussed on impurity loading upon snow surfaces. Please comment further on the suitability of the Impurity Index for ice surfaces.

Results, section 4.1: I find this section very difficult to read. It would benefit from re-writing and introduction of paragraph breaks.

Fig. 3a: typo, August spelt 'Agust'

Fig. 3b: provide MERIS band numbers at top of plot to aid cross-comparison back to Table 1. The colours of the two dark ice spectra lines are too similar to be able to tell them apart in print.

P11 L275: full stop missing after '1400 m'.

P12 L278-290: I do not follow the arguments being made in this section. Further, I disagree with the statement made in reference to Fig. 4, that 'Similar to the Impurity Index, the dark ice area is not only limited to the algae-abundant areas'. My examination of Fig 4 suggests that this is cherry-picking as conversely I saw plenty of evidence of a very good match between the two indices. As the authors central premise is that the 2BDA is 'uniquely biological' and so therefore yielding details not provided by the Impurity index or Dark ice index I propose that quantification beyond eye-balling the associated plots is required – ideally some statistical approach.

P12 L288-290: this study has no field data for the wavy patterns caused by ancient ice outcropping and does not provide any zoomed satellite imagery which shows them, so the reference to Wientjes and Oerlemans (2010) strikes me as somewhat speculative.

P13 L302: 'exhibits different spatiotemporal variations'. Are these differences statistically significant? They are almost impossible to identify by eye, apart from in one or two years of record 2BDI. Consider doing some elevational binning to support your case.

Fig. 5: add 2BDA and Impurity index labels to each row of subplots.

Fig. 6: What p-value where these trends culled at, if at all? I also note that the R2 values in the referenced appendix plot are very small.

Fig. 7: Please provide some indication of measurement spread at each point, e.g. +/- 1 s.d. I would also prefer to see just the 2014 MERIS data for comparison with the 2014 algal abundance time series, rather than the aggregate 2004-2011 time series which is shown currently. Previous work e.g. Shimada et al. (2016) and Tedstone et al. (2017)

has shown that there is considerable inter-annual variability and so I think more value here could come from detailed analysis of how algal growth proceeds in each season.

P16 L342-359: very wordy. Requires paragraphing. Also consider in here which assertions can be retained once major review comments are addressed. It remains particularly difficult to follow the links with the field data despite close reading of the m/s.

Fig. 8: (a) panels use inter-annual averages of each day and are therefore not especially useful at the process-level: like any other process, algal growth is not actually dependent on time but on a range of processes. Examination of individual years with varying melt season characteristics would therefore be more useful. At the very least, it would be good to see faint lines for each year plotted into the background of these panels. Associated question: how much 'noise' is there in individual years relative to the 'climatological' averages being shown?

Fig. 8c and section 4.4: is the chosen breakpoint of 20 August statistically significant?

Discussion: excessively wordy in places, can be shortened without loss of meaning. Fig. 11: provide colorbar to interpret density colors. Consider providing R2 values instead of just R.

Text of page 22: this paragraph is overly long. It requires a re-structure.

P22 L456-459: might be worth noting here that this is opposite to the results of Tedstone et al. (2017).

L460: 'For each of the two variables'

P23 L474-481: reads hugely speculatively, especially given the relative lack of process-level understanding about ice algae available in the literature.

Fig. 13a,b: why was a white mid-point of $\sim$0.97 chosen? Aren't algae judged to be present at values < 1?

Fig. 13c,d,e: I am not sure what the relevance of providing these data are. At the very

least it would be useful to add some kind of annual 2BDI and Impurity Index time series for comparison with the provided metrics.

---

## Author Comment (AC2) · 1 Feb 2020

**Overview**

This manuscript uses data from the MERIS satellite sensor to seek to quantify glacier algae bloom dynamics over the south west Greenland ablation zone. They justify their use of this sensor for detecting algal blooms by reference to their previous work using the very similar Sentinel-3 OLCI on the same topic (Wang et al, 2018), by selected references to some field observations, and by wider reference to remote sensing of ocean-borne algal blooms.

**Response:** We greatly appreciate the reviewer's careful reading and useful suggestions, which have improved our manuscript. In this revision, we believe that we have solved most of the concerns raised by the reviewer, please see our responses below.

**Major comments**

The manuscript tests several remote sensing ratio indices and shows that, to some extent, the 2BDA approach retrieves a different signal to that obtained by the 'bulk' Impurity Index or simple red band threshold approaches. This is a useful exercise in seeking to understand what signals can be retrieved from the MERIS/OLCI sensors.

As the manuscript is presented currently, I have some major concerns which prevent me from recommending publication in The Cryosphere.

There are known problems with seeking to apply band ratios/indices designed for chlorophyll-a retrieval from water bodies but which this manuscript does not engage with. I appreciate that the main studies which highlight these problems, by Cook et al. (TCD) and Tedstone et al. (TCD), are currently undergoing review and so were unlikely to have been available at the time when this study was started. But nevertheless, there is a lack of discussion of the wider literature on this issue; instead, Wang et al (2018) is cited as proof that chlorophyll-a-focused band ratios are appropriate for detecting glacier algae blooms, but I find that discussion of these problems is lacking there too, and so their 2018 paper is not an especially strong foundation on which to base the present study.

**Response:** In this revision, we added a section (5.1, shown below) to analyze and discuss the sensitivity of 2BDA index to dust presence using SNICAR simulations with variant dust sizes and concentrations (more details are in the response to reviewer#1). Our results indicate that the 2BDA index is much less sensitive to dust presence than the impurity index, and in our context (with impurity index mostly less than 1.0), the high 2BDA index (greater than 0.99) is unlikely caused by dust. Given that the 2BDA index is specifically designed for chlorophyll-a retrieval and the narrow bandwidths of MERIS, the 2BDA index (especially at values greater than 0.99) is uniquely biological due to glacier algae.

Since the discussion paper is not referenceable, we removed all discussion papers from our citations. The paper by Cook et al. (2020) was just published, and we added this citation to our introduction and discussion. Cook et al. (2020) also discussed the 'red-edge' feature present in their field spectra and argued that this feature is a unique chlorophyll-a feature.

**5.1 Sensitivity analysis of 2BDA index to dusts**

In this study, we utilized the chlorophyll-a spectral signature generated by glacier algae in the red-NIR region (Fig. 2d) to quantify the spatiotemporal variability of glacier algae over bare ice in southwest Greenland. The narrow bandwidths and wavelengths of MERIS designed for chlorophyll-a detection make the MERIS archive data a powerful tool to study supraglacial algal communities. The chlorophyll-a signal present in the MERIS spectra is consistent with (nearly) coincident WorldView-2 data and field hyperspectral measurements collected over dark ice with high algal abundance. Similar to the Sentinel-3 OLCI ratio index R709nm/R673nm, the MERIS 2BDA index R709nm/R665nm can effectively quantify the algal growth pattern through July to August. However, given that dusts may complicate the spectra and affect the 2BDA index, we performed radiative transfer modelling experiments using the Snow, Ice, and Aerosol Radiation (SNICAR) model (Flanner et al., 2007; Flanner et al., 2009) to test the sensitivity of 2BDA index to dust presence. Using SNICAR-Online (Flanner et al., 2007), we simulated the spectra for varying sizes and concentrations of dusts, and then calculated the corresponding 2BDA and impurity indices. The SNICAR input parameters include incident radiation (direct), solar zenith angle (60 degrees), clear- vs. cloudy sky (Summit Greenland), snow grain effective radius (1500 microns, considering the ice surface), snowpack thickness (100 m), snowpack density (400 kg/m3), and dust concentration (0.1, 0.3, 0.5, 0.8, 1, 1.5, 2, 2.5, 3, 5, 8, 10, 30, 50, 80, 100, 300, 500, 800, 1000, 1500, 2000, 2500, and 3000 ppm) for four dust sizes (dust 1: 0.1–1.0µm; dust 2: 1.0–2.5µm; dust 3: 2.5–5.0µm; dust 4: 5.0–10.0 µm). We should note here that there has been some discussion in past literature of hematite-rich dust (e.g. Tedesco et al., 2013; Cook et al., 2020), which could produce a different spectral response. However, the study of Cook et al. (2020) finds very low concentrations of such dust, and therefore we consider its impact to be negligible. Figure 10 shows the scatterplots of impurity index vs. 2BDA index calculated for the SNICAR simulations (with circle diameters representing the magnitude of dust concentrations for four different dust sizes), and the density scatterplots from the MERIS data (impurity vs. 2BDA indices over bare ice). Figure 10 indicates that the impurity index is more sensitive to dust presence than the 2BDA index. In contrast, the upper bound of the impurity index we calculated from the MERIS data is around 1.0. According to the SNICAR simulations, the impurity index of 1.0 corresponds to a maximum 2BDA value of 0.99 (~500 ppm concentration for dust 4). This indicates that for our study area, the glacier algae identified with 2BDA index greater than 0.99 are unlikely to be false positives caused by dusts.

Figure 10: Impurity index vs. 2BDA index for MERIS pixels, excluding missing data in our study area between 2004 and 2011 (density scatter plot). Circles show impurity vs. 2BDA index from SNICAR simulations with varying concentrations of surface dust (with 4 different dust sizes). The circle size corresponds to the dust concentration, and dashed lines show the polynomial regression for each of the different dust sizes.

As I understand it, the core of this problem is two-fold: (1) the optics of bare ice are insufficiently well understood to be able to guarantee that the reduction in reflectance around 667 nm compared to 710 nm is uniquely biological; and (2) other light absorbing impurities may interfere or present the same signal. Thus, based on published field evidence, there is little evidence that the band ratio approach is uniquely biological. Cook et al. (TCD) and Tedstone et al. (TCD) have more information on this and note that phenolic compounds for in the dominant glacier algae species can obscure potentially diagnostic spectral features. This being the case, NDCI etc may simply be measuring some combination of slightly different surface characteristics to the Impurity Index approach, rather than yielding information specifically on glacier algae growth. Thus, regarding inter-annual mapping of 'dark ice' vs glacier algae, there may be little advance on Shimada et al. (2016) or Tedstone et al. (2017), both of whom considered inter-annual variability in 'dark ice' dynamics over the timescales addressed here.

**Response:** We agree that the optics of bare ice and light absorbing impurities can complicate the spectral signal, but we respectfully disagree that "based on published field evidence, there is little evidence that the band ratio approach is uniquely biological". Current field studies (Stibal et al., 2017; Cook et al., 2020) presented the field hyperspectral data of dark ice with abundant glacier algae, and their data show the chlorophyll-a signature at the red-NIR region. However, they did not apply the band ratio approach, which doesn't necessarily mean that 'there is little evidence that

the band ratio approach is uniquely biological'. As we discuss below, the ice/snow optics have little impact on the 2BDA index, and based on radiative transfer modeling experiments (response to reviewer #1, and revised discussion section 5.1), the upper limit of the dust impact on the 2BDA index is around 0.99. In contrast, the impurity index is more sensitive to dust presence. In the revised text, we have added more discussion and figures to show the difference between impurity index and 2BDA index. We also respectfully disagree with the statement 'Thus, regarding interannual mapping of 'dark ice' vs glacier algae, there may be little advance on Shimada et al. (2016) or Tedstone et al. (2017), both of whom considered inter-annual variability in 'dark ice' dynamics over the timescales addressed here', since according to our results (Figure 4), there are differences between 2BDA index, impurity index, and the R620nm reflectance that Shimada et al. (2016) and Tedstone et al. (2017) used for dark ice delineation. We would like to argue that their method doesn't account for any biological signal specific to glacier algae, and is more likely to be influenced by meltwater presence, ice optics and other impurities.

On justification of the 2BDA, Wang et al. (2018) point to Painter et al. (2001) as evidence that glacier algae can detected using chlorophyll-a indices. However, Painter et al refers to the specific case of snow algae growing on snow surfaces, which is not relevant here as this study engages only with bare ice surfaces. Thus, retrievals in this study can in fact be based only on paired cell counts and field spectra acquired by Stibal et al. (2017), a study which also indicates that chlorophyll-a-based approaches could be useful for remote sensing. However, the spectra that Stibal et al (2017, Fig. 3) present refers only to high algal abundance ice, over centimetres patch scales, which is not representative of OLCI or MERIS 300 m data. Some consideration of the scale mismatch is therefore required.

Response: We respectfully disagree with the reviewer on the point that the specific case of snow algae growing on snow surfaces is not relevant with the glacier algae growing on bare ice surface. There are differences between snow and ice spectra, but both of them are characterized by the decreased reflectance at 709 nm as compared with 665 nm. The spectral signature of ice and snow themselves exhibit a slope opposite to that of the chlorophyll-a spectra at this region. Although snow algae and glacier algae are distinct species, they both generate chlorophyll-a for photosynthesis activity and chlorophyll-a is their major photosynthetic pigment. The colours of snow algae and glacier algae are different mainly because snow algae generate secondary carotenoids which have reflectance peak at red band. However, according to Painter et al. (2001), this carotenoid feature does not block the chlorophyll-a absorption signal around 680 nm, so they detected snow algae based on the chlorophyll-a signature between 630 nm and 700 nm using the absorption at 680 nm and the reflectance feature at 630 nm and 700 nm. Glacier algae have brownish-grey colour because they generate purpurogallin pigments, and at the same time, they also generate chlorophyll-a for photosynthesis (similar to snow algae). However, as we responded to the first reviewer, compared with the purpurogallin pigment, Chlorophyll-a is more appropriate for mapping glacier algae for the following reasons:

 Chlorophyll-a is the primary photosynthetic pigment of glacier algae (Williamson et al., 2018). The ocean color satellite sensors like Envisat MERIS and Sentinel-3 OLCI are designed to capture the Chlorophyll-a signal from highly-absorptive and optically complex water bodies, which means that the ocean color sensors are highly sensitive to the chlorophyll-a presence, making them very useful tools for glacier algae detection based on the biological signatures.

2) According to the studies by Remias et al. (2012) and Williamson et al. (2018), the spectral signatures (absorption peaks) of the purpurogallin pigment are concentrated in the UV region (278 nm, 304 nm, and 389 nm, Remias et al.,2012). To our knowledge, no satellite sensor can detect these spectral signatures. Although the purpurogallin pigment is very likely to account for the brownish-grey colour of glacier algae, its absorption over the entire visible spectrum is quite uniform, making it difficult to differentiate from other dark impurities. In contrast, chlorophyll-a can generate very strong spectral signatures in the red and NIR region, which is supported by field hyperspectral measurements for both snow algae and glacier algae. (e.g. Ganey et al., 2017; Painter et al., 2001; Stibal et al., 2017; Cook et al., 2020).

As we clarified in the text, we used the field measurements by Stibal et al. (2017) for qualitative evidence to show that the MERIS spectra, WorldView-2 spectra, and field hyperspectral data are consistent in terms of the spectral shape over algae-abundant ice. In this revision, we revised Fig. 2d to include more field spectra data from Stibal et al. (2017) to illustrate that the chlorophyll-a spectral signature at the red-NIR region is present across multiple measurement samples and dates. Additionally, the recently published paper by Cook et al. (2020) also discussed the 'red-edge' feature present in their field data, which is attributed to the chlorophyll-a generated by glacier algae. In regard to the scale issues, the MERIS (300 meter) spectra and WorldView-2 (2 meter) spectra are quite similar, and previous studies (e.g. Ryan et al., 2018) show that the areal percentage of the distributed impurities is up to >90% within individual MODIS pixels (500-meter resolution). Therefore, MERIS data can capture well the glacier algae signal over southwest Greenland; nevertheless, we agree with the reviewer that more investigations on the scale and spectral mixing issues are needed in future studies. We have revised the discussion to acknowledge those issues. Besides, as we responded to reviewer #1, we excluded the possibility of false positives to detect glacier algae caused by dusts when the 2BDA index is greater than 0.99.

Possibly a more minor concern: the cell counts used as field validation in this manuscript are very high, at 105 cells ml (Figure 2d), but I'm not sure that we would expect to see such high counts over these larger spatial scales (e.g. Williamson et al.,2018, FEMS). Furthermore, the field spectra seem to have quite high reflectance for the quoted cell counts compared to other field spectra in the literature, e.g. Figure A1 in Tedstone et al. (2017, TC). The field spectra shown here seems to be that in Stibal et al. (2017, GRL, Figure 3), but a cell count is not quoted there and so I raise this question here in case there has been an error in transforming Stibal et al's data for this study.

**Response:** In this revision, we clarified in the text on how we used the field data by Stibal et al. (2017). The field hyperspectral measurements collected by Stibal et al. (2017) were used for qualitative purposes for comparison with the MERIS spectra over dark ice to validate the chlorophyll-a spectral signature at the red-NIR region, specifically the bands of 709 nm and 665 nm used for 2BDA index calculation. We have revised Fig.2d by adding multiple in situ spectra collected over the algae-abundant dark ice (R620nm<0.4, and algal concentration >=10000 cells/ml) to illustrate that the chlorophyll-a spectral signature is present across multiple

measurement samples and dates. We have double checked the original data published by Stibal et al. (2017) and ensured the correctness of our plotted spectra.

---

## Author Comment (AC3) · 1 Feb 2020

Thank you for the suggestions. We have changed the text accordingly.

---

## Author Response (AR1)

**Editor**

*Your TCD manuscript "Quantifying spatiotemporal variability of ice algal blooms and the impact on surface albedo in southwest Greenland" received two constructive reviews where both reviewers identified methodological issues and questions, which you already partly clarified in your author comments.*

*Now I would like to ask you to upload a revised version of the manuscript together with an author's response where you clearly address the comments of the reviewers. In this revised version I think it is specially important to address the topic of sensitivity of the 2BDA methods to changes in dust, ice properties, etc. and this may even include this sensitivity in your results (and not only discussion).*

**Response:** We sincerely thank the editor and two anonymous reviewers for their insightful and constructive comments and suggestions. We have endeavored to address all the comments and improve the manuscript to our best. In particular, to address the comments regarding the sensitivity of the 2BDA index to glacier algae as compared to other factors, we included new sections (**3.3 Sensitivity analysis based on radiative transfer modelling, 4.2 Sensitivity analysis of 2BDA index to non-algal factors, and 5.1 Sensitivity to subpixel variability**) to analyze and discuss the impacts of dust presence, ice properties, and scale issues on 2BDA index. We used the radiative transfer model SNICAR and the spectral linear mixing method to analyze and discuss those factors. To support our statements, we added substantial details in the appendix and improved our figure presentations overall.

In this document, we provide point-by-point responses to the reviewer comments, a summary list of all relevant changes made in the manuscript, and a copy of the manuscript with changes highlighted. We believe that the manuscript has been greatly improved in this revision and we hope that the revised manuscript will be suitable for publication.

**Anonymous Referee #1**

*Overall:*

*The authors have presented a paper that attempts to quantify biomass over the western Greenland Ice Sheet using the well known "rededge" technique that they refer to as "2BDA", which is often used for detecting chlorophyll containing biota such as photosynthetic algal blooms in oceanic and lacustrine environments, vegetation and crop mapping. Biomass quantification over the Greenland Ice Sheet is a worthy research goal because Greenland Ice Sheet glacier algae very likely play an important role in controlling the ablation zone albedo that is not yet accounted for in energy balance models. This is well within the scope of The Cryosphere and the scientific question is worthy of consideration in this journal. There are some very useful aspects to the paper, including demonstration that there are ablation zone albedo processes that SMB models currently do not account for, the albedo time series over the western ablation zone, and the comparison between different band ratios. However, there are some major issues that need to be addressed before I can recommend publication. More detail is provided below.*

**Response:** We greatly appreciate the reviewer's careful review of our manuscript. Many thanks for suggesting to use SNICAR model to assess the sensitivity of 2BDA index to various dust concentrations. In this revision, we believe the manuscript has been greatly improved by incorporating the reviewer comments. Please see our responses below.

*Major Comments:*

*1)There is past literature that emphasises the importance of discounting abiotically generated rededge signals before assuming them to be diagnostic of photosynthetic life (see Sparks et al. 2009; Seager et al. 2005). The vulnerability of the rededge to false positives is demonstrable using a simple radiative transfer model (easily replicated inbrowser via http://snow.engin.umich.edu/). Fig 1A shows the results from SNICAR runs where all variables were held constant except the mass concentrations of completely inorganic impurities (Flanner et al.'s (2009) "global average dusts, type 4"). The lowest 2 spectral albedo curves returned a false positive 2BDA result (1.002 and 1.005). In Figure 1B, the model is run again identically but with a hematiterich mineral dust taken from Polashenski et al. (2015), giving eight 2BDA false positives (1.004, 1.015, 1.026, 1.031, 1.039, 1.043 1.046). Tedesco et al. (2013) suggested that hematiterich red dusts are present in the GrIS ablation zone (we note that Cook et al. (in review) disagreed about that but their paper remains unpublished). Taking Tedesco et al. (2013) to be correct about the prevalence of red dusts on the Greenland ablation zone therefore invalidates the rededge as a biomarker due to the demonstrable potential for false positives. Convincing empirical data is required to demonstrate that the 2BDA signal is exclusively biological and robust to these types of false positive results.*

A)                                                    B)

[Figure]

*Fig 1: A) SNICAR runs with diffuse irradiance, homogenous snow with grain diameter 500 micron, density 400 kg m$^3$ and Flanner et al. (2009)'s "dust 4" in the upper 1 mm, in mass concentrations of 0.1, 0.3, 0.5, 0.8, 1.0, 1.5, 2.0, 2.5, 3.0 ugdust/gice. B) Identical SNICAR runs but with Polashenski et al. (2015)'s high hematite dust.*

**Response**: We thank the reviewer for pointing out the issue of potential impacts by dusts. In this revision, we addressed this concern by conducting SNICAR simulations with various parameter settings as the reviewer suggested. We would like to clarify that our objective is not to define a universal biomarker for detecting photosynthetic life. To our understanding, the two papers the reviewer mentioned that address the potential false signal resulting from dusts (Sparks et al. 2009 and Seager et al. 2005; not in the reference list), are in the extraterrestrial context.

However, our research is conducted based on the understanding that widespread glacier algal blooms occur on the bare ice zone in southwest Greenland, which have been confirmed by numerous studies (Cook et al., 2020; Lutz et al., 2014; Remias et al., 2012; Ryan et al., 2018; Stibal et al., 2015; Stibal et al., 2017; Williamson et al., 2019; Yallop et al., 2012). Nevertheless, to evaluate the sensitivity of the 2BDA index to various dust sizes and concentrations, we performed radiative transfer modelling experiments using SNICAR, by setting the grain size of snow to 500 microns and 1500 microns. However, we cannot generate the same results using the dust concentrations specified by the reviewer (0.1, 0.3, 0.5, 0.8, 1.0, 1.5, 2.0, 2.5, 3.0 $\mu g_{dust}/g_{ice}$). Using these parameters, the 2BDA index is less than 0.97 for all dust sizes (dust 1, dust 2, dust 3, and dust 4) when the grain size is 1500 microns, and less than 0.98 when grain size is 500 microns. The 2BDA index would be over 1.0 only when the dust concentrations are greater than ~800 ppm.

In this revision, we added the section 3.3 (Sensitivity analysis based on radiative transfer modelling) and section 4.2 (Sensitivity analysis of 2BDA index to non-algal factors) to specifically analyze this issue. Using a grain size of 1500 microns produces a spectral curve that is closest to the MERIS bare ice spectra. The SNICAR experiments were performed with the following parameters: direct incident radiation, a solar zenith angle of 60 degrees, clear-sky conditions (for Summit Greenland), a snow grain effective radius of 1500 micron (to approximate the ice surface), a snowpack thickness of 100 m (to avoid any influence of the sub-snowpack albedo), a snowpack density of 400 kg/m$^3$, and dust concentrations of (0.1, 0.3, 0.5, 0.8, 1, 1.5, 2, 2.5, 3, 5, 8, 10, 30, 50, 80, 100, 300, 500, 800, 1000, 1500, 2000, 2500, and 3000 ppm) for four dust sizes (dust 1: 0.1–1.0μm; dust 2: 1.0–2.5μm; dust 3: 2.5–5.0μm; dust 4: 5.0–10.0 μm). We also tested different density values but

these did not affect the simulation results. In addition to the 2BDA index, we also calculated the impurity index for the SNICAR simulations, and found that the impurity index is more sensitive to dusts than the 2BDA index (Figure 4 in the revised manuscript, shown below). The figure (below) shows scatterplots of impurity index vs. 2BDA index calculated for the SNICAR simulations (with the diameter of circles representing the magnitude of dust concentrations for four different dust sizes), and density scatterplots from the MERIS data (impurity vs. 2BDA indices over bare ice). The results indicate that relatively high concentrations of dust would increase the 2BDA index, but would also result in a large increase in the impurity index. By contrast, the upper bound of the impurity index we calculated from the MERIS data is around 1.0, below the impurity index values for the highest dust concentrations. These results suggest that relatively high 2BDA values (especially above 0.99, corresponding to an impurity index of 1.0) are unlikely to be caused by dusts, because the presence of dusts would also result in an impurity index of above 1.0. This indicates that for our study area, the glacier algae identified with a 2BDA index greater than 0.99 are not likely to be false positives caused by dusts. Finally, even below this threshold, the slope of the 2BDA vs. impurity index is shallower than the SNICAR-generated curves, suggesting that the 2BDA index is generally more sensitive to chlorophyll-a than to dust.

With regard to the possible presence of hematite-rich dust, the samples of Tedesco et al. (2013) were from cryoconite holes, and are not necessarily representative of the ice surface; and the hematite concentration in those samples was actually very low. Cook et al. (2020) also found that the local bare-ice mineral dust is poor in hematite and rich in weakly absorbing quartz and feldspar minerals. Therefore, the hematite has a negligible influence on the detected chlorophyll-a signal at the red-NIR region.

[Figure]

**Figure 4 (in the revised manuscript). Impurity index vs. 2BDA index for MERIS bare ice pixels (density scatter plot with colours indicating relative frequency), excluding missing data in our study area, between 2004 and 2011. Circles show impurity vs. 2BDA index from SNICAR simulations with varying concentrations of dust (with four different dust sizes). The circle size corresponds to the dust concentration, and dashed lines show the polynomial regression for each of the**

**different dust sizes. The circle size corresponds to the dust concentration, and dashed lines show the polynomial regression for each of the different dust sizes.**

*2) The authors do not account for the spectral albedo of the ice itself. Ice albedo can vary dramatically independently of light absorbing particles and cause the 2BDA value to change, undermining the biomass quantification. Figure 2 shows identical simulations to Fig 1, except the grain size is increased to 1500 microns. False positive results are returned as before, but the value of the 2BDA indexes and therefore the retrieved biomass – change (Flanner et al. (2009) dust =1.0009 and 1.004; Polashenski et al. (2015) dust = 1.0018, 1.014, 1.026, 1.032, 1.040, 1.043, 1.045, 1.046). The retrieved biomass therefore changed without any change in impurity loading. On real glacier ice where the ice albedo can vary by tens of percent independently of impurity concentration due to weathering crust development, meltwater accumulation and drainage, topography, impurity mixing and glaciological structure, the potential for highly error prone retrievals is likely very high. The authors need to demonstrate that their band ratio is not vulnerable to this uncertainty, or that they can quantify and correct for it.*

[Figure]

*Fig 2: A) SNICAR runs with diffuse irradiance, homogenous 1500 micron, 400 kgm³ snow with Flanner et al. (2009)'s "dust 4" in the upper 1 mm, in mass concentrations of 0.1, 0.3, 0.5, 0.8, 1.0, 1.5, 2.0, 2.5, 3.0 ugdust/gice. B) Identical SNICAR runs but with Polashenski et al. (2015)'s high hematite dust.*

**Response:** The sensitivity of the 2BDA index to dust presence (over snow with a 1500 microns grain size to approximate ice) has been discussed above.

We agree on the point that ice albedo changes with impurity concentration, meltwater presence, topography and crevasses, as discussed by Ryan et al. (2018). But also, as suggested by Ryan et al. (2018), the distributed impurities explain most of the spatial variability of surface albedo. In addition, it should be noted that variations in albedo due to other factors including water, other impurities, and ice albedo does not necessarily affect the 2BDA index. Ice with different concentrations of air bubbles has a consistent spectral shape between 665 nm and 709 nm (Condom et al., 2018), and meltwater exhibits a similar pattern at this wavelength range, both of which are characterized by a decreasing reflectance from 665 nm to 709 nm. The sensitivity of the 2BDA index to glacier algae can be further demonstrated using the field dataset of Cook et al. (2020), as illustrated by Table C1 and Figure C2 (in the revised manuscript, shown below). The MERIS band ratio between 709 nm and 665 nm (both bands have bandwidths of 10 nm) is specifically designed

for chlorophyll-a, and less affected by dusts as we discussed above. To our knowledge, the meltwater, weathering crust, and crevasses do not cause the pattern of increasing reflectance from 665 nm to 709 nm. In our revised Figure 2d (shown below), it is clearly shown that the MERIS exhibits the red-NIR spectral signature caused by chlorophyll-a, which matches multiple field hyperspectral data measurements over algae-abundant dark ice (which are likely subject to varying ice conditions). We have added discussions in section 4.2 accordingly.

[Figure]

**Figure C2 (in the revised manuscript). Scatterplot of measured cell abundance versus 2BDA index listed in Table C1 based on the published field dataset of Cook et al. (2020).**

[Figure]

**Figure 2 (in the revised manuscript). Comparison between MERIS, WorldView-2, and field spectra over algae-abundant dark ice. (a) MERIS Level-2 image (true colour composite) acquired on 5 July 2010. Pixels with missing data are shown in light blue. (b) WorldView-2 surface reflectance image acquired on 9 July 2010 over the square area in (a). (c) Zoomed-in WorldView-2 image, with the area (red square) corresponding to the selected MERIS pixel in (a). (d) Reflectance spectra for MERIS and WorldView-2 (2010), and field hyperspectral measurements collected over the algae-abundant dark ice at S6 by Stibal et al. (2017) in 2014.**

*3) The authors do not adequately address the problem of scale mismatches between the normal length scales of typical algal blooms (biomass varies dramatically over 110 m length scales) and the satellite used to gather their data (300 x 300m). Surface heterogeneity must surely introduce major uncertainties as the spectral reflectance of each pixel is the combined product of many highly variable surfaces. How would, for example, cryoconite, surface water, dust, crevasses and surface topography at the subpixel scale affect their biomass quantification? These unquantified factors must influence the predicted cell concentration independently of realworld changes to the mass concentration of algae, but they are not discussed in the paper.*

**Response:** In this revision, we added section 5.1 (Sensitivity to subpixel variability) to discuss the scale issues. Based on our SNICAR experiments, and analysis of the 2BDA and impurity indices, the 2BDA index is less sensitive to the presence of dust, which means that the high 2BDA index is uniquely biological. Given the sensitivity of MERIS to the presence of chlorophyll-a, the 2BDA index can capture well the chlorophyll-a signal generated by glacier algae. To examine the potential impact of spatial heterogeneity on the MERIS 2BDA index, we performed spectral linear mixing experiments using the field hyperspectral measurements of Cook et al. (2020) for glacier algae and bare ice and the SNICAR-simulated spectra for dust. We obtained the mixed spectra (Figure C3 in the manuscript, shown below) by specifying the different areal percentage of algae/dust vs. bare ice, and calculated the corresponding 2BDA index for the mixed spectra. It is shown that the 2BDA index dramatically increases with the areal percentage of glacier algae, being consistent with the assumption that the 2BDA index is positively correlated with the algal abundance. In contrast, the 2BDA index has much less sensitivity to dust. The high-resolution UAV mapping by Ryan et al. (2018) suggests that the areal percentage of the distributed impurities is up to 65%~95% within individual MODIS pixels (500-meter resolution) over the dark zone in southwest Greenland. Our linear mixing experiments (Figure C3b) suggest that the MERIS 2BDA index can capture the glacier algae variability within the dark zone. In addition, our comparison between the MERIS spectra, WorldView-2 spectra, and field hyperspectral data (Figure 2 in the manuscript, shown above) shows that the chlorophyll-a signature at the red-NIR region is quite consistent between different source measurements with different spatial scales.

[Figure]

**Figure C3 (in the revised manuscript). Spectral linear mixing experiments. (a) Field hyperspectral measurements of four algae-abundant samples (21_7_SB1, 21_7_SB2, 23_7_SB5, and 21_7_SB10) and one bare ice sample (zero algal abundance, 14_7_SB6) from Cook et al. (2020), and the SNICAR-simulated spectra for the dust scenario (size 4 at concentration of 500 ppm). (b) 2BDA index calculated from the linearly mixed spectra with varying areal percentage at subpixel scale for algae (different algal abundances) and dust scenarios.**

*4) There is insufficient detail regarding the use of field spectroscopic measurements as "ground truth". Only one single field spectrum is presented in the paper and the measurement conditions are not reported. Has it been picked because it matches well with the MERIS spectrum, is it a mean (in which case of how many samples, and what do the error bars look like) or is it the only available spectrum? How do other spectra in the field dataset compare? Can the authors provide evidence to suggest that centimeter scale field spectroscopy measurements are truly representative of the biomass over entire MERIS pixels?*

**Response**: We have revised the text to include more details on how we used the field data by Stibal et al. (2015) and Stibal et al. (2017). In our study, we used those field data in a qualitative way to validate the spatial variations of algal concentration magnitude derived from the satellite data, and to compare the field hyperspectral measurements over algae-abundant ice with the MERIS spectra and WorldView-2 spectra. The field measurements are collected after the period of MERIS measurements, precluding direct comparison with field data. In this revision, we revised Figure 2d (shown above) to include additional field spectra collected over dark ice (R620nm<0.4) with high algal abundance (cell concentrations greater than 10000 cells/ml). As illustrated by Figure 2d, the spectral characteristics at the red-NIR region match well between MERIS spectra, WorldView-2 spectra and field spectra. The match between MERIS spectra (300 meter) and WorldView-2 spectra (2 meter) also indicate that the chlorophyll-a signal cannot be masked out because of large spatial scales, given the high areal percentage of the distributed impurities within the MERIS pixel, as illustrated by Figure 2 (in the revised manuscript) and given the estimation of Ryan et al. (2018) that the areal percentage of the distributed impurities is about 65~95% within individual MODIS pixels (500-meter resolution) in dark ice areas.

*5) The authors are selective with their citing of literature under review. If they wish to include papers still under discussion they should explain why the issues of spatial scale encountered in 20 m Sentinel2 pixels discussed by Cook et al (in review) and Tedstone et al. (in review) do not also apply to their 300 m MERIS pixels. If they decide to stick to published literature they should explain how the presence of hematiterich dusts on the ablating Greenland Ice Sheet as reported by Tedesco et al. (2013) does not invalidate their assumption that the rededge is uniquely biological.*

**Response:** We removed the references to all discussion papers since they are not referenceable. The paper by Cook et al. (2020) is now published, and we have included this citation in our introduction section and discussion section. We also mentioned the potential impact of hematite dust (Tedesco et al. 2013) on the 2BDA index in section 4.2. As we mentioned above, the hematite has a negligible impact on the 2BDA index in our context.

**Specific Comments:**

*Title: As suggested by Daniel Remias in the open discussion forum, please adopt the generally accepted terminology "glacier algae" that distinguishes these algae from those found in sea ice.*

**Response:** As suggested, we have changed 'ice algae' to 'glacier algae' throughout the text.

*General point about chlorophyll: Referring to "chlorophyll" is somewhat ambiguous as it could imply total chlorophyll or one of several chlorophyll variants. Please be specific that you mean chlorophylla.*

**Response:** We have revised the text as suggested.

*L39: Any citation for yellow/orange snow algae? They are normally thought of as green or red.*

**Response:** We have added the citation (Anesio et al., 2017) for the yellow/orange pigmentation of snow algae.

*L71: The authors rightly criticise carotenoid based remote sensing methods because of possible false positives due to "dirt" but ignore the potential for equivalent rededge false positives due to dust.*

**Response:** In this revision, we ran a number of SNICAR simulations with variant dust sizes and concentrations. Based on the SNICAR simulations, we calculated both the 2BDA and impurity indices for different dust configurations, and evaluated the potential impact of dust presence on 2BDA index. Please see details in the revised section 4.2 (Sensitivity analysis of 2BDA index to non-algal factors).

*L75: The authors claim chlorophyll is the appropriate pigment to use to identify ice algae despite also stating that the coloration of the algae is primarily due to purpurogallin pigments. Why, then, is that not the appropriate pigment to use to identify glacier algae?*

**Response:** Compared with the purpurogallin pigment, Chlorophyll-a is more appropriate for mapping glacier algae for the following reasons:

1) Chlorophyll-a is the primary photosynthetic pigment of glacier algae (Williamson et al., 2018). The ocean color satellite sensors like Envisat MERIS and Sentinel-3 OLCI are designed to capture the Chlorophyll-a signal from highly-absorptive and optically complex water bodies, which means that the ocean color sensors are highly sensitive to the chlorophyll-a presence, making them very useful tools for glacier algae detection based on the biological signatures.

2) According to the studies by Remias et al. (2012) and Williamson et al. (2018), the spectral signatures (absorption peaks) of the purpurogallin pigment are concentrated in the UV region (278 nm, 304 nm, and 389 nm, Remias et al.,2012). To our knowledge, no satellite sensor can detect these spectral signatures. Although the purpurogallin pigment is very likely to account for the brownish-grey colour of glacier algae, its absorption over the entire visible spectrum is quite uniform, making it difficult to differentiate from other dark impurities. In contrast, chlorophyll-a can generate very strong spectral signatures at the red and NIR region, which are supported by field hyperspectral measurements for both snow algae and glacier algae. (e.g. Ganey et al., 2017; Painter et al., 2001; Stibal et al., 2017; Cook et al., 2020).

We have revised the text (introduction section) to discuss and compare the suitability of purpurogallin vs. chlorophyll-a for glacier algae mapping.

*L75: "owing to its unique spectral signatures between 665 710nm (Gitelson, 1992; Painter et al., 2001; Wang et al., 2018)": Chlorophylla absorbs in narrow bands around 680 nm and 440 nm. Any effects extending up to 710 nm are due to interactions with the surrounding medium. This is why Painter et al. (2001) was able to use the narrow 680 nm absorption feature as a diagnostic tool for Chlorophylla detection.*

**Response:** By 'unique spectral signatures between 665-710nm', we are referring to the absorption between 665-681 nm and the reflectance peak around 710nm. Painter et al. (2001) used the 680nm absorption feature by calculating the integral of the absorption scaled by its continuum spectra. Painter et al. (2001) retrieved the continuum spectrum by linearly interpolating the reflectance peaks at 630 nm and 700 nm, which similarly to our study, essentially used the relative difference between the absorption and reflectance features at the red-NIR region. Their method is specifically applicable for hyperspectral data like AVIRIS, but is limited for satellite multispectral data. We have revised the text to improve the clarity.

*L79: "Quantification of ice algae biomass from satellite data based on the chlorophylla feature has received less attention since the chlorophyll related satellite bands designed for land generally have coarse spectral resolutions." This is just one of many reasons why remotely detecting algae over glacier ice is not simple. Other complexities include the complex pigmentation of the algae, the spatial resolution of the remote sensing instruments relative to the typical length scales of individual surface features (including algal blooms) and critically the optics of the underlying ice that vary dramatically over space and time and which are not yet well described. These issues are as important as the spectral resolution and must be acknowledged.*

**Response:** We have revised the text (introduction section) to acknowledge these issues.

*L100 – 120: Much more detail is required here. For example, how many actual field samples were used to validate your remote sensing retrievals? What were the biomasses measured at those sites? What were the measurement conditions? On which dates were spectra available at which sites? Were the measurement times conistent and how do they compare to the satellite overpass times? What was the sensor footprint size for the field measurements and how were these upscaled to the satellite pixel scale?*

**Response:** We have revised this section and made clarifications on how we used the field measurements. It should be noted that we utilized those field data in a qualitative way for comparison with the satellite signals, rather than in a quantitative way for direct algal biomass inversion. The Envisat MERIS was operational from March 2002 to April 2012. To our knowledge, there were no algal field data coincident with the satellite data. In our study, we estimated the growth rate (population doubling time) and albedo reduction rate (for each population doubling) using a simple mathematical conversion and empirical relationship established from the Sentinel-3 OLCI data and field data (Wang et al. 2018). We did not directly estimate the algal biomass or abundance from MERIS data since no coincident field data are available. We attempted to apply the Sentinel-3 retrieved empirical relationship to estimate the population doubling time and albedo reduction rate due to algae, and the results match well with the spatial variability from previous field observations.

*Figure 1: Please provide details of the field spectrum presented as the dashed line. Where/when was it collected and how does it compare to other field spectra presented in this paper?*

**Response:** We have added more details in the figure caption (below) to describe the field spectrum.

**Figure 1. Spectral response functions of (a) MERIS (red), OLCI (blue), and (b) MODIS (black), and WorldView-2 (orange) over the wavelength range of 350-1050 nm. All the MERIS and OLCI bands are within the 350-1050 nm range, where photosynthetic and photoprotective pigments have spectral responses. Four MODIS bands (over land) and eight WorldView-2 bands are within this spectral range, but with much coarser spectral resolutions. In both sub-plots, the dashed line shows hyperspectral ASD field spectrometer data (right vertical axis) collected over algae-abundant ice by Stibal et al. (2017), containing chlorophyll-a signal at the red-NIR wavelengths (red highlighted region). The plotted field spectrum (sample code: Ab.25.06.14.D1) was measured on 25 June 2014 at 67°04.779'N, 49°24.077'W (near the automatic weather station S6 along the K-transect), with an algal abundance measurement of 121664 cells/ml (Stibal et al., 2017).**

*L209: Chlorophylla is the primary photosynthetic pigment, but not the primary light absorbing pigment. In both the studies you have cited the chlorophylla absorption feature is actually extremely subtle – in fact in Cook et al. (in review) it was only really discernable in the derivative spectra and indistinguishable in the raw reflectance. In Stibal et al. (2017) the spectrum are presented with a very truncated yaxis to make the pigment feature discernable.*

**Response:** We agree with the reviewer on the point that Chlorophyll-a is the primary photosynthetic pigment but not the primary light absorbing pigment. However, this does not mean that Chlorophyll-a cannot be used as the biomarker to detect glacier algae. According to the literature, the purpurogallin pigments are the primary light absorbing pigments for glacier algae. As we mentioned above, the characteristic spectral signatures generated by purpurogallin pigments (that might be used as biomarker for glacier algae detection) are concentrated in the ultraviolet region (278 nm, 304 nm, and 389 nm). To our knowledge, current satellite sensors cannot capture

the spectral signals at these wavelengths, which means that the spectral properties of purpurogallin pigments in the UV region cannot be utilized for glacier algae detection from space. The absorption features of the purpurogallin pigments are quite uniform over the entire visible spectrum, with no characteristic spectral signatures that can be used by satellite sensors to differentiate glacier algae from other dark materials. Although the Chlorophyll-a spectral signature (between 665 nm and 709 nm) generated by glacier algae is not as strong as the algal blooms in aquatic systems, the spectral characteristics of Chlorophyll-a are indeed present on the spectral curve, which are particularly obvious in the derivative spectra (shown in Cook et al. 2020) and the normalized spectra (revised Figure 3c, shown below).

[Figure]

**Figure 3 (in the revised manuscript). MERIS spectra of different surface types. (a) MERIS Level-2 image (false colour composite) acquired on 14 August 2011 and locations of the four sample sites. Each site has an area of 1.2 km by 1.2 km, composed of 16 MERIS pixels. (b) MERIS reflectance in 13 spectral bands over the four sites, illustrated by mean and standard deviation values for each band over each site. (c) Normalized reflectance relative to the clean ice spectra.**

*L211: "pure ice" has lower reflectance at red wavelengths compared to shorter wavelengths.*

**Response:** We have revised the text as 'Pure ice has lower reflectance at 709 nm compared to shorter wavelengths (Hall and Martinec, 1985).'

*L216220: This line of reasoning borrows heavily from studies of chlorophylla dominated species in other environments and still requires the rededge to be validated over glacier ice where LAP and meltwater mixing, complex pigmentation and ice optics are potential confounding variables.*

**Response:** We agree that the mixture of dusts, algal pigments, meltwater, and ice optics could complicate the surface spectra. In this revision, we discussed the potential impacts of these variables on the 2BDA index by incorporating the SNICAR simulations. Although the 2BDA index is developed and well-validated for ocean color applications, the rationale for glacier algae detection based on the chlorophyll-a spectral signature at the red-NIR region is very similar to that of algal detection in aquatic environments, particularly for turbid case 2 waters (Blondeau-Patissier

et al., 2014; Matthews, 2011). Similar to the dark ice surface, the case 2 waters are also optically complex, largely affected by the colored dissolved organic matter (CDOM) and suspended sediments. The 2BDA index based on the 665nm and 709nm bands utilizes the reflectance peak near 709 nm, which has been widely tested and validated for the case 2 waters. Using Figure 2 (in the revised manuscript, shown above), we intend to show that the algae-laden ice has the chlorophyll-a spectral signature, which is consistent between the 300-meter MERIS spectra, 2-m WorldView-2 spectra, and the in-situ hyperspectral data. Multiple in situ spectra have been added to Figure 2 illustrating that the chlorophyll-a spectral signature is present across multiple measurement samples and dates.

*Figure 2: How did the authors select the field spectrum to plot on this figure? Is this the average of all available? If so please provide error bars and number of observations. Also, 184 cells/mL reported in the legend is a tiny amount of algae, unlikely to explain the albedo reduction observed –is this a typo? What was the mineral dust type and concentration in the same area – could it also explain the rededge? How much of the albedo reduction can be attributed to the algae and how much to melt water/dust? If the absorption is mostly due to chlorophylla as the authors suggest, why is the absorption maximum outside of the chlorophyll absorption range shown in Fig 2c and why does it extend across the visible wavelengths? Why do the field spectra and remotely sensed spectra diverge below ~640 nm?*

**Response:** The field spectrum we selected from Stibal et al. (2017) is used here as an example to show that the chlorophyll-a spectral characteristics (665-709 nm) over algae-abundant ice, and the satellite data (2-meter resolution WorldView-2 and 300-meter MERIS imagery) have similar spectral features at this red-NIR region. The selection criteria include high measured algal abundance (184184 cells/ml) and dark appearance (R620nm<0.45, consistent with the dark ice delineation criteria by Shimada et al., 2016 and Tedstone et al. 2017). To further illustrate the red-NIR spectral signature of glacier algae, we added multiple field spectra from locations where algal cell concentrations were measured at greater than 10,000 cells/ml to Figure 2, showing consistent spectral shapes in this wavelength region. For detailed information about the field data, including dust composition and albedo reduction caused by different variables, please refer to Stibal et al. (2017). As mentioned above, we have added a section analyzing the impact of dusts on 2BDA index in the revised discussion. 'The absorption maximum outside of the chlorophyll absorption range' can be explained by the purpurogallin pigments. The low absorption and uniform absorption in this range actually emphasizes the importance of using the 'red-edge' feature to detect glacier algae. We are not suggesting that 'the absorption is mostly due to chlorophyll-a', instead, we are suggesting to use the chlorophyll-a feature (absorption at 665 nm and reflectance peak at 709 nm) to detect glacier algae. We have revised the text to clarify this point. The divergence between the field spectra and satellite spectra below ~640 nm may be caused by two factors: 1) the uncorrected Rayleigh scattering effect that affects shorter visible wavelengths (particularly the blue band). and 2) spectral mixing with ice. Both can make the reflectance at shorter wavelengths higher, however, the reflectance ratio between 709 nm and 665 nm is less affected by these factors.

*Figure 3: A) "Agust"->August;*

**Response:** The typo has been corrected.

*B) The authors present spectra for "dark → ice (more chlorophyll)" and "dark ice (less chlorophyll)". However, there does not seem to be any positive 2BDA signal in the latter spectrum at all. Is it actually "dark ice (no chlorophyll)"? If so, there are additional darkening processes occurring on the ice. What processes are darkening the ice in those areas and to what extent do those ice darkening processes also influence the biomass retrievals in areas where there is a positive 2BDA result? What effect does this have on retrieved biomass? What is the detection limit for the 2BDA method?*

**Response:** We have corrected the figure to refer to "high chlorophyll-a" and "low chlorophyll-a". To illustrate the chlorophyll-a signal better, we also plotted the relative surface reflectances (MERIS) for different surface types normalized to the clean ice spectra (Figure 3c in the revised manuscript) since the primary background spectral signal is from ice. For both water and ice, the spectrum shows a decrease in reflectance from 665 nm to 709 nm, which is opposite to the chlorophyll-a spectrum. A 2BDA signal of less than one therefore does not imply that there is no chlorophyll-a present. A smaller rate of decrease could still be produced by low amounts of chlorophyll-a. Using the 2BDA index, we do not intend to classify the ice surface into 'algae' vs. 'no algae'. We use the 2BDA index to show the magnitude of glacier algal blooms varying over space and time. We think it is more appropriate to use 'high chlorophyll-a' and 'low chlorophyll-a' to describe those two sites. We agree with the reviewer that more discussions and investigations are needed to quantify the impacts of other darkening processes on 2BDA index. In this revision, we added the analysis of dust impacts on 2BDA index based on SNICAR simulations in section 4.2 (Sensitivity analysis of 2BDA index to non-algal factors). We found that by combining the 2BDA index with the Impurity Index, we can exclude the possibility of false positives when the 2BDA index is greater than 0.99.

*L413: Cook et al. (in review) mention that a rededge signal was present in most of their algal hyperspectral data but they do not mention false positive rates and they opted not to use that method for their spatial upscaling. It would therefore be useful to know the false positive rate in the present study and how it scales to 300m MERIS pixels.*

**Response**: Cook et al. (2020, published online) showed that the 'red-edge' spectral signal due to chlorophyll-a is present in their hyperspectral measurements for algae-covered ice, which further supports the chlorophyll-a signal we observed in the 300-meter MERIS image and the 2-meter WorldView-2 image. In this revision, we revisited the field dataset of Cook et al. (2020). They provided coincident data of cell abundance and hyperspectral measurements for a number of field sites (Table C1 in the revised manuscript). Based on their field datasets, we calculated the 2BDA index for each sample, and Figure C2 (in the revised manuscript, shown above) shows the strong correlation between cell abundance and 2BDA index based on the Cook et al. (2020) field data. To investigate potential scale issues, we added a discussion section (5.1 Sensitivity to subpixel variability) to analyze the sensitivity of 2BDA index to subpixel variability of glacier algae vs. dust by performing spectral linear mixing experiments based on the Cook et al. (2020) field spectral data and the SNICAR-simulated dust spectral data. As we discussed above, the MERIS 2BDA index is capable of capturing the glacier algae at the 300-meter resolution scale. This is further supported by the comparison between the WorldView-2 image (2-meter resolution) and

the MERIS image (Figure 2 in the revised manuscript, shown above). Although we can observe spatial heterogeneity within one MERIS pixel, the dark materials are widespread over the entire area. This is also consistent with the UAV mapping results by Ryan et al. (2018), showing that the areal percentage of the distributed impurities is up to 90% within individual MODIS pixels (500-meter resolution) over the dark zone in southwest Greenland. Nevertheless, it is important to investigate the pixel mixture problems more rigorously in the future and the limit of algae distribution within each pixel that can cause detectable chlorophyll-a signal. Based on our discussion on dust impacts and the spatial scale of MERIS imagery, we think that using MERIS data is more likely to cause false negatives instead of false positives given the sensor detection limit to weaker chlorophyll-a signals.

*L416: It is not clear to me from the manuscript precisely how you have inferred algal cell abundance. Please provide further methodological details.*

**Response:** The methods for computing algal population doubling were described in Section 4.3 (Lines 363-376 in the original manuscript). However, this section may have been somewhat unclear. In this revision, we have clarified how the population doubling time was estimated based on the fitted coefficients between 2BDA and time (section 4.5 in the revised manuscript). We did not directly infer the algal cell abundance using the 2BDA index, instead, we used the empirical relationship established based on the Sentinel-3 OLCI band ratio and previous field measurements (Wang et al., 2018) and mathematical conversions.

*L450460: Another explanation for this is that the overall ice albedo is lower, there may be smoother ice and more water at the surface, and rather than there being less algae, the rededge signal is simply erased by an overall dampening of the spectrum across all wavelengths (i.e. putting dark impurities on dark ice has a less detectable effect that putting the same impurities on otherwise bright ice). Can the authors demonstrate that this is not the case?*

**Response:** As we have discussed above, the 2BDA index is sensitive to the absorption and reflectance peaks of chlorophyll-a, which is not a feature of other surface types. As the 2BDA index is a ratio of two different wavelength bands, a uniform reduction in "background" albedo should have a small effect. A change in the shape of the "background" spectrum (the relative reflectance at 709 nm relative to 665 nm would be required to have a large impact on 2BDA. Observed spectra shown in Figure 2 (in the revised manuscript, shown above) suggest that differences in the average magnitude of the reflectance spectrum do not appear have a strong impact on the shape of the reflectance spectra, and therefore likely do not strongly impact the 2BDA index either.

*As I understand it, the core of this problem is two-fold: (1) the optics of bare ice are insufficiently well understood to be able to guarantee that the reduction in reflectance around 667 nm compared to 710 nm is uniquely biological; and (2) other light absorbing impurities may interfere or present the same signal. Thus, based on published field evidence, there is little evidence that the band ratio approach is uniquely biological. Cook et al. (TCD) and Tedstone et al. (TCD) have more information on this and note that phenolic compounds for in the dominant glacier algae species can obscure potentially diagnostic spectral features. This being the case, NDCI etc may simply be measuring some combination of slightly different surface characteristics to the Impurity Index approach, rather than yielding information specifically on glacier algae growth. Thus, regarding inter-annual mapping of 'dark ice' vs glacier algae, there may be little advance on Shimada et al. (2016) or Tedstone et al. (2017), both of whom considered inter-annual variability in 'dark ice' dynamics over the timescales addressed here.*

**Response:** We agree that the optics of bare ice and light absorbing impurities can complicate the spectral signal, but we respectfully disagree that "based on published field evidence, there is little evidence that the band ratio approach is uniquely biological". Current field studies (Stibal et al., 2017; Cook et al., 2020) presented the field hyperspectral data of dark ice with abundant glacier algae, and their data show the chlorophyll-a signature at the red-NIR region. However, they did not apply the band ratio approach, which doesn't necessarily mean that 'there is little evidence that the band ratio approach is uniquely biological'. As we discuss below, the ice/snow optics have little impact on the 2BDA index, and based on radiative transfer modelling experiments (response to reviewer #1, and revised section 4.2), the upper limit of the dust impact on the 2BDA index is around 0.99. In contrast, the impurity index is more sensitive to dust presence. In the revised text, we have added more discussion and figures to show the difference between impurity index and 2BDA index. We also respectfully disagree with the statement 'Thus, regarding inter-annual mapping of 'dark ice' vs glacier algae, there may be little advance on Shimada et al. (2016) or Tedstone et al. (2017), both of whom considered inter-annual variability in 'dark ice' dynamics over the timescales addressed here', since according to our results (Figure 5 in the manuscript), there are differences between 2BDA index, impurity index, and the R620nm reflectance that Shimada et al. (2016) and Tedstone et al. (2017) used for dark ice delineation. We would like to argue that their method doesn't account for any biological signal specific to glacier algae, and is more likely to be influenced by meltwater presence, ice optics and other impurities.

*On justification of the 2BDA, Wang et al. (2018) point to Painter et al. (2001) as evidence that glacier algae can detected using chlorophyll-a indices. However, Painter et al refers to the specific case of snow algae growing on snow surfaces, which is not relevant here as this study engages only with bare ice surfaces. Thus, retrievals in this study can in fact be based only on paired cell counts and field spectra acquired by Stibal et al. (2017), a study which also indicates that chlorophyll-a-based approaches could be useful for remote sensing. However, the spectra that Stibal et al (2017, Figure 3) present refers only to high algal abundance ice, over centimetres patch scales, which is not representative of OLCI or MERIS 300 m data. Some consideration of the scale mismatch is therefore required.*

**Response**: We respectfully disagree with the reviewer on the point that the specific case of snow algae growing on snow surfaces is not relevant to the discussion of glacier algae growing on bare ice surfaces. There are differences between snow and ice spectra, but both of them are characterized by decreased reflectance at 709 nm as compared with 665 nm. The spectral signature of ice and snow themselves exhibit a slope opposite to that of the chlorophyll-a spectra at this region. Although snow algae and glacier algae are distinct species, they both generate chlorophyll-a for photosynthetic activity and chlorophyll-a is their major photosynthetic pigment. The colours of snow algae and glacier algae are different mainly because snow algae generate secondary carotenoids which have a reflectance peak in red band. However, according to Painter et al. (2001), this carotenoid feature does not block the chlorophyll-a absorption signal around 680 nm. Therefore, Painter et al. (2001) were able to detect snow algae using the chlorophyll-a signature between 630 nm and 700 nm using the absorption at 680 nm and the reflectance features at 630 nm and 700 nm. Glacier algae have brownish-grey colour because they generate purpurogallin pigments, and at the same time, they also generate chlorophyll-a for photosynthesis (similar to snow algae). However, as we noted in response to the first reviewer, compared with the purpurogallin pigment, Chlorophyll-a is more appropriate for mapping glacier algae for the following reasons:

1) Chlorophyll-a is the primary photosynthetic pigment of glacier algae (Williamson et al., 2018). The ocean color satellite sensors like Envisat MERIS and Sentinel-3 OLCI are designed to capture the Chlorophyll-a signal from highly-absorptive and optically complex water bodies, which means that the ocean color sensors are highly sensitive to chlorophyll-a presence, making them very useful tools for glacier algae detection based on the biological signatures.

2) According to the studies by Remias et al. (2012) and Williamson et al. (2018), the spectral signatures (absorption peaks) of the purpurogallin pigment are concentrated in the UV region (278 nm, 304 nm, and 389 nm, Remias et al.,2012). To our knowledge, no satellite sensor can detect these spectral signatures. Although the purpurogallin pigment is very likely to account for the brownish-grey colour of glacier algae, its absorption over the entire visible spectrum is quite uniform, making it difficult to differentiate from other dark impurities. In contrast, chlorophyll-a can generate very strong spectral signatures in the red and NIR region, which is supported by field hyperspectral measurements for both snow algae and glacier algae. (e.g. Ganey et al., 2017; Painter et al., 2001; Stibal et al., 2017; Cook et al., 2020).

As we clarified in the text, we used the field measurements by Stibal et al. (2017) for qualitative evidence to show that the MERIS spectra, WorldView-2 spectra, and field hyperspectral data are consistent in terms of the spectral shape over algae-abundant ice. In this revision, we improved Figure 2d (in the revised manuscript) to include more field spectra data from Stibal et al. (2017) to illustrate that the chlorophyll-a spectral signature at the red-NIR region is present across multiple measurement samples and dates. Additionally, the recently published paper by Cook et al. (2020) also discussed the 'red-edge' feature present in their field data, which is attributed to the chlorophyll-a generated by glacier algae. Examining the field measurements and hyperspectral data of Cook et al. (2020), we find a strong sensitivity of the 2BDA index derived from field hyperspectral data with coincident measured cell abundance. We have added Table C1 and Figure C2 showing the relationship between cell abundance and the 2BDA index.

In regard to the scale issues, we added section 5.1 (Sensitivity to subpixel variability) to discuss the effects of spatial scale on the 2BDA index. Based on our SNICAR experiments, and analysis of the 2BDA and impurity indices, the 2BDA index is less sensitive to the presence of dust, which means that the high 2BDA index is uniquely biological. Given the sensitivity of MERIS to the presence of chlorophyll-a, the 2BDA index can effectively capture the chlorophyll-a signal generated by glacier algae. To examine the potential impact of spatial heterogeneity on the MERIS 2BDA index, we performed spectral linear mixing experiments using the field hyperspectral measurements of Cook et al. (2020) for glacier algae and bare ice and the SNICAR-simulated spectra for dust. We obtained the mixed spectra (Figure C3 in revised the manuscript, shown below) by specifying the different areal percentage of algae/dust vs. bare ice, and calculated the corresponding 2BDA index for the mixed spectra. It is shown that the 2BDA index dramatically increases with the areal percentage of glacier algae, being consistent with the assumption that the 2BDA index is positively correlated with algal abundance. In contrast, the 2BDA index has much less sensitivity to dust. The high-resolution UAV mapping by Ryan et al. (2018) suggests that the areal percentage of the distributed impurities is up to 65%~95% within individual MODIS pixels (500-meter resolution) over the dark zone in southwest Greenland. Our linear mixing experiments (Figure C3b, shown below) suggest that the MERIS 2BDA index can capture the glacier algae variability within the dark zone. In addition, our comparison between the MERIS (300-meter resolution) spectra, WorldView-2 (2-meter resolution) spectra, and field hyperspectral data (Figure 2 in the manuscript, shown below) shows that the chlorophyll-a signature at the red-NIR region is quite consistent between different source measurements at different spatial scales. Therefore, MERIS data can effectively capture the glacier algae signal over southwest Greenland; nevertheless, we agree with the reviewer that more investigations on the scale and spectral mixing issues are needed in future studies. Besides, as we noted in our response to reviewer #1, we excluded the possibility of false positives caused by dusts when the 2BDA index is greater than 0.99.

[Figure]

**Figure C3 (in the revised manuscript). Spectral linear mixing experiments. (a) Field hyperspectral measurements of four algae-abundant samples (21_7_SB1, 21_7_SB2, 23_7_SB5, and 21_7_SB10) and one bare ice sample (zero algal abundance, 14_7_SB6) from Cook et al. (2020), and the SNICAR-simulated spectra for the dust scenario (size 4 at concentration of 500 ppm). (b) 2BDA index calculated from the linearly mixed spectra with varying areal percentage at subpixel scale for algae (different algal abundances) and dust scenarios.**

[Figure]

**Figure 2 (in the revised manuscript). Comparison between MERIS, WorldView-2, and field spectra over algae-abundant dark ice. (a) MERIS Level-2 image (true colour composite) acquired on 5 July 2010. Pixels with missing data are shown in light blue. (b) WorldView-2 surface reflectance image acquired on 9 July 2010 over the square area in (a). (c) Zoomed-in**

**WorldView-2 image, with the area (red square) corresponding to the selected MERIS pixel in (a). (d) Reflectance spectra for MERIS and WorldView-2 (2010), and field hyperspectral measurements collected over the algae-abundant dark ice at S6 by Stibal et al. (2017) in 2014.**

*Possibly a more minor concern: the cell counts used as field validation in this manuscript are very high, at 105 cells ml (Figure 2d), but I'm not sure that we would expect to see such high counts over these larger spatial scales (e.g. Williamson et al.,2018, FEMS). Furthermore, the field spectra seem to have quite high reflectance for the quoted cell counts compared to other field spectra in the literature, e.g. Figure A1 in Tedstone et al. (2017, TC). The field spectra shown here seems to be that in Stibal et al. (2017, GRL, Figure 3), but a cell count is not quoted there and so I raise this question here in case there has been an error in transforming Stibal et al's data for this study.*

**Response:** In this revision, we clarified in the text on how we used the field data by Stibal et al. (2017). The field hyperspectral measurements collected by Stibal et al. (2017) were used for qualitative purposes for comparison with the MERIS spectra over dark ice to validate the chlorophyll-a spectral signature at the red-NIR region, specifically the bands of 709 nm and 665 nm used for 2BDA index calculation. We have revised Figure 2d (shown above) by adding multiple in situ spectra collected over the algae-abundant dark ice (R620nm<0.4, and algal concentration >=10000 cells/ml) to illustrate that the chlorophyll-a spectral signature is present across multiple measurement samples and dates. We have double checked the original data published by Stibal et al. (2017) and ensured the correctness of our plotted spectra.

*The study also presents data that undermines its application of a Chlorophyll-a based band ratio approach. Figure 3b shows some averaged MERIS surface reflectance curves. Dark Site (Less Chlorophyll) has higher reflectance at 665 than 709 nm and so with 2BDA this site would presumably diagnose as 'clean ice' by comparison to the Clean Ice spectrum plotted above it. I do not see any comment upon this issue elsewhere in the text.*

**Response:** We respectfully disagree with the reviewer that our presented data in Figure 3 undermines its application of a Chlorophyll-a based band ratio approach. As we responded to reviewer#1, we have corrected the figure to refer to "high chlorophyll-a" and "low chlorophyll-a". For both water and ice, the spectrum shows a decrease in reflectance from 665 nm to 709 nm, which is opposite that of the chlorophyll-a spectrum. A 2BDA signal of less than one therefore does not imply that there is no chlorophyll-a present. A smaller rate of decrease could still be produced by low amounts of chlorophyll-a. Using the 2BDA index, we do not intend to classify the ice surface into 'algae' vs. 'no algae'. We use the 2BDA index to show the magnitude of glacier algal blooms varying over space and time. We think it is more appropriate to use 'high chlorophyll-a' and 'low chlorophyll-a' to describe those two sites. To illustrate the chlorophyll-a signal better, we have revised the text to discuss this issue and added a subplot (Figure 3c in the manuscript, shown below) to show the normalized MERIS surface reflectances relative to the clean ice spectrum.

We agree with the reviewer that more discussions and investigations are needed to quantify the impacts of other darkening processes on 2BDA index. In this revision, we added the analysis of dust impacts on 2BDA index based on SNICAR simulations in section 3.3 (Sensitivity analysis

based on radiative transfer modelling) and section 4.2 (Sensitivity analysis of 2BDA index to non-algal factors). We found that by combining the 2BDA index with the Impurity Index, we can exclude the possibility of false positives when the 2BDA index is greater than 0.99.

[Figure]

**Figure 3 (in the revised manuscript). MERIS spectra of different surface types. (a) MERIS Level-2 image (false colour composite) acquired on 14 August 2011 and locations of the four sample sites. Each site has an area of 1.2 km by 1.2 km, composed of 16 MERIS pixels. (b) MERIS reflectance in 13 spectral bands over the four sites, illustrated by mean and standard deviation values for each band over each site. (c) Normalized reflectance relative to the clean ice spectra.**

*I'm very confused about how the algal population doubling times were calculated. This is a critical part of the manuscript as it underpins the assertion that there is a 0.02-0.04 reduction rate in albedo for each algal population doubling.*

**Response:** The methods for computing algal population doubling time were described in Section 4.3 (Lines 363-376 in the original manuscript). However, this section may have been somewhat unclear. In this revision, we have clarified how the population doubling time was estimated based on the fitted coefficients between 2BDA and time (section 4.5 in the revised manuscript).

*Overall, I would urge nuanced engagement with the question of how confident can we be that the differences between 2BDA, Impurity Index and Dark Ice metrics are due solely to algae and not to other processes that might affect this band ratio? I suggest that this needs much clearer explanation in the methods about how Stibal et al's field data were used in this manuscript, and some nuanced discussion of the uncertainties surrounding Chlorophyll-a indices on ice surfaces. If these issues are addressed then the revised manuscript may be suitable for publication.*

**Response**: As we mentioned above, we have analyzed the sensitivity of 2BDA index and impurity index to dust presence by using the SNICAR simulations with variant dust sizes and concentrations (Section 4.2, Figure 4 in the manuscript). We have excluded the possibility of false positives for glacier algae detection caused by dusts using the 2BDA index (particularly greater than 0.99).

We have clarified in the text on how we used the field data from Stibal et al. (2017). The match between the MERIS spectra, WorldView-2 spectra, and field spectra (Figure 2 in the manuscript, shown above) indicates the chlorophyll-a signal can be effectively captured by MERIS.

*Minor comments*

*I agree with the short comment by Daniel Remias that this manuscript should use the terminology 'glacier algae' in preference to 'ice algae'.*

**Response:** As suggested, we have changed 'ice algae' to 'glacier algae' through the text.

*The introduction includes wide-ranging references to both glacier algae and snow algae. Detailed discussion of the snow algae literature is not relevant here as this study focuses only on bare ice surfaces, so the introduction would benefit from being focused solely on glacier algae.*

**Response:** We respectfully disagree with the reviewer on this point. We think it is important to discuss the differences between snow algae and glacier algae, as the differences may not be clear to a broader audience, as well as the similarities and techniques that can help inform our current study.

*P3 L71: define what is meant by 'dirt'.*

**Response**: We used 'dirt' according to the studies by Painter et al. (2001) and Takeuchi et al. (2006). However, since glacier algae do not generate secondary carotenoids like snow algae, the spectral characteristics of carotenoids are not relevant to our study. To avoid ambiguity, we have removed the statement 'the spectral characteristics of dirt may resemble those of carotenoids' from the manuscript.

*P8 L209. Cook et al. (2019, Cryosphere Discussions) are cited for the first time here. If it is being cited then it should be introduced earlier during the lit. review section of the Introduction. Alternatively, if taking the view that Cook et al is under discussion and that it isn't 'referenceable', then all references to it should be removed.*

**Response**: This paper is currently published online (Cook et al., 2020). We have included it in the introduction and updated the reference.

*P9 L226: please quantify how the 'best' means of quantifying ice algae was obtained. This is not clear, either here or in the subsequent text.*

**Response:** We have rewritten the paragraph to improve the clarity.

*P9 L229: Dumont et al. (2014) focussed on impurity loading upon snow surfaces. Please comment further on the suitability of the Impurity Index for ice surfaces.*

**Response:** The Impurity Index (Dumont et al., 2014) is the ratio between the natural logarithms of the spectral albedos at 545–565 nm (visible green band) and at 841–876 nm (NIR band). This index was built upon the hypothesis that the surface reflectances at visible wavelengths are more sensitive to impurity content than at the NIR wavelengths, and Dumont et al. (2014) found that the

Impurity Index is almost insensitive to grain size based on their field measurements and radiative transfer modelling results. Given the similar spectral shapes of snow and ice and the general assumption in radiative transfer modelling that ice has larger grain size than snow, we think it is suitable to apply the Impurity Index to the ice surface. Besides, Dumont et al. (2014) quantified the impurity content using this index over the Greenland Ice Sheet for the May–July period from 2003 to 2013, including the bare ice zone in southwest Greenland. We have added more details in the text to describe the impurity index.

*Results, section 4.1: I find this section very difficult to read. It would benefit from re-writing and introduction of paragraph breaks.*

**Response:** This section has been rewritten as suggested and additional paragraph breaks have been added for clarity.

*Fig. 3a: typo, August spelt 'Agust' Fig. 3b: provide MERIS band numbers at top of plot to aid cross-comparison back to Table 1. The colours of the two dark ice spectra lines are too similar to be able to tell them apart in print.*

**Response:** We have revised the figure as suggested. We also added a subplot to show the normalized MERIS reflectances to the clean ice spectrum, showing the spectral signature of chlorophyll-a better.

*P11 L275: full stop missing after '1400 m'.*

**Response:** We have revised the text accordingly.

*P12 L278-290: I do not follow the arguments being made in this section. Further, I disagree with the statement made in reference to Fig. 4, that 'Similar to the Impurity Index, the dark ice area is not only limited to the algae-abundant areas'. My examination of Fig 4 suggests that this is cherry-picking as conversely I saw plenty of evidence of a very good match between the two indices. As the authors central premise is that the 2BDA is 'uniquely biological' and so therefore yielding details not provided by the Impurity index or Dark ice index I propose that quantification beyond eye-balling the associated plots is required – ideally some statistical approach.*

**Response:** In this revision, we changed the color scheme for each variable (2BDA, Impurity Index, R620nm, and MODIS bare ice albedo). The revised figure shows better the differences between those variables. Figure 5a (in the revised manuscript, shown below) clearly shows that along the central dark zone, 2BDA is highest at the elevation level of 1200-1400m, and comparatively the Impurity index is highest at the elevation level of 1000-1200m. The R620nm used for dark ice delineation in previous studies (Shimada et al. 2016; Tedstone et al. 2017) has the lowest value at the elevation level of 1000-1200m, more consistent with the Impurity Index. To show the spatial variation in more detail, we have also added a supplementary figure (Figure A1 in the revised manuscript, shown below) to illustrate the variations of different indices with elevation (at a 20-meter interval), which also shows the differences clearly. We have rewritten the text accordingly.

[Figure]

+ Sampling locations by Stibal et al (2015) in the summer of 2013    — Elevation contours    ○ 66±31 cells/ml    ⬭ 5688±3147 cells/ml    ◯ 10621±2073 cells/ml

**Figure 5 (in the revised manuscript). Spatial patterns of the mean 2BDA index (a), impurity index (b), reflectance at 620 nm (c), and MODIS broadband albedo (d) over the bare ice zone during July and August from 2004 to 2011. The elevation contours illustrate the spatial variations of each variable with altitude. The cross labels show the spatial locations of the field sites DS, KAN_L, and KAN_M and magnitude of glacier algal abundance (circle labels) measured by Stibal et al. (2015) in 2013.**

[Figure]

**Figure A1 (in the revised manuscript). Spatial variations of the average 2BDA index, impurity index, 620 nm reflectance, and MODIS albedo over bare ice at different elevations within the study area (20-meter elevation interval). For surface elevation, we used the 30-meter resolution MEaSUREs Greenland Ice Mapping Project (GIMP) Digital Elevation Model (Howat et al., 2014; 2015).**

*P12 L288-290: this study has no field data for the wavy patterns caused by ancient ice outcropping and does not provide any zoomed satellite imagery which shows them, so the reference to Wientjes and Oerlemans (2010) strikes me as somewhat speculative.*

**Response:** The study by Wientjes and Oerlemans (2010) indeed has no field data for the wavy patterns caused by ancient ice outcropping, but they do show the zoomed in ASTER satellite image (15-meter resolution) in their Figure 8 to illustrate the observed wavy patterns which are typical for outcropping tilted layers of ice. In our context, the 2BDA index indicates that along the central dark zone, there were more glacier algae distributed at the elevation level of 1200-1400m as compared with the 1000-1200m elevation level. However, the darkening index (R620nm, e.g. Shimada et al., 2016 and Tedstone et al., 2017) and the Impurity Index (Dumont et al., 2014) indicate that the 1000-1200m elevation zone also contains high impurity content, suggesting that other darkening processes potentially played an important role in this area. Therefore, we discussed the possibility of ancient ice outcropping in this area based on Wientjes and Oerlemans (2010). In this revision, we provided additional evidence (Figure A2 in the revised manuscript, shown below) to support this observation, using 2-meter resolution WorldView-2 images to show the differences between those two elevation zones. The WorldView-2 image (Figure A2b) clearly shows the wavy patterns mentioned by Wientjes and Oerlemans (2010) at 1000-1200m, and very different textures are visible at 1200-1400m (Figure A2c) where high algae content was identified using the 2BDA index.

[Figure]

**Figure A2 (in the revised manuscript). Average 2BDA index (2004-2011) for a subset of our study area (a) and comparison between WorldView-2 imagery over a dark ice site with low 2BDA index at 1000-1200m elevation (b) and a dark ice site with high 2BDA index at 1200-1400m elevation (c). The WorldView-2 image in (b)**

illustrates the 'wavy' pattern that Wientjes and Oerlemans (2010) suggested was caused by ancient ice outcropping.

*P13 L302: 'exhibits different spatiotemporal variations'. Are these differences statistically significant? They are almost impossible to identify by eye, apart from in one or two years of record 2BDI. Consider doing some elevational binning to support your case.*

**Response:** In this revision, we grouped the annual 2BDA and Impurity Index time series into different elevation bins (600-800m, 800-1000m, 1000-1200m, and 1200-1400m), to better illustrate the differences, and calculated the average values for each elevation bin. We also added a supplementary figure (Figure A3, shown below) in the appendix to show the annual time series of 2BDA and Impurity indices for different elevation levels, and revised the text accordingly.

[Figure]

**Figure A3 (in the revised manuscript). Interannual variability of the 2BDA index (a) and impurity index (b) at the elevation levels of 600-800m, 800-1000m, 1000-1200m, and 1200-1400m within the study area.**

*Fig. 5: add 2BDA and Impurity index labels to each row of subplots.*

**Response:** We have revised the figure as suggested.

*Fig. 6: What p-value where these trends culled at, if at all? I also note that the R2 values in the referenced appendix plot are very small.*

**Response:** We added the p-value maps to Figure A4 (in the revised manuscript) to show the spatial extent where the annual 2BDA index, Impurity Index, and MODIS broadband albedo have statistically significant annual trends from 2004 to 2011 which are limited to a few areas. Although the trends in most areas are not statistically significant at the 95% confidence level, we still think it is useful to examine the patterns of interannual trends for the different indices. We have revised the text to make note of the locations where Figure A4 shows statistically significant trends.

*Fig. 7: Please provide some indication of measurement spread at each point, e.g. +/- 1 s.d. I would also prefer to see just the 2014 MERIS data for comparison with the 2014 algal abundance time series, rather than the aggregate 2004-2011 time series which is shown currently. Previous work*

*e.g. Shimada et al. (2016) and Tedstone et al. (2017) has shown that there is considerable inter-annual variability and so I think more value here could come from detailed analysis of how algal growth proceeds in each season.*

**Response:** Since the Envisat MERIS was operational from March 2002 to April 2012, we are unable to provide a 2014 MERIS time series coincident with the 2014 field data. In this revision, we removed the figure from our manuscript. To consider the algal growth in different seasons, we extracted the seasonal growth functions (2BDA vs. time) for different seasons and compared with the growth function extracted from the 'climatological' averages over the two example sites KAN_M and DS (Figure A5 in the revised manuscript, shown below).

[Figure]

**Figure A5 (in the revised manuscript). Temporal trends of 2BDA index from mid-July to Mid-August in different years at sites DS (a) and KAN_M (b).**

*P16 L342-359: very wordy. Requires paragraphing. Also consider in here which assertions can be retained once major review comments are addressed. It remains particularly difficult to follow the links with the field data despite close reading of the m/s.*

**Response:** We have rewritten and restructured this section, and have added more text describing how and why we compared the remote sensing data to previous field data.

*Fig. 8: (a) panels use inter-annual averages of each day and are therefore not especially useful at the process-level: like any other process, algal growth is not actually dependent on time but on a range of processes. Examination of individual years with varying melt season characteristics would therefore be more useful. At the very least, it would be good to see faint lines for each year plotted into the background of these panels. Associated question: how much 'noise' is there in individual years relative to the 'climatological' averages being shown?*

**Response**: We aggregated the daily data in different years to estimate the overall seasonal trend since the data for some years are not sufficient to obtain a reliable seasonal trend. We agree that algal growth is a complicated process, being affected by multiple factors like nutrients, meltwater, sunlight, temperature, and so on, which needs further investigation in the future by combining in

situ and remote sensing observations with meteorological and regional climate modelling data. However, characterizing the seasonal trend of algal growth over time is still useful for understanding the average patterns of seasonal growth across multiple years. In this revision, we added a supplementary figure (Figure A5, shown above) to show the temporal variations of 2BDA index in different years over the sites KAN_M and DS (algae-abundant area), showing that the regression slope of 2BDA vs. time is quite consistent between different years despite some variability at the DS site.

*Fig. 8c and section 4.4: is the chosen breakpoint of 20 August statistically significant?*

**Response:** We choose 20 August as the breakpoint because for most of the years we studied, snowfall happened after 20 August and covered the bare ice surface where glacier algae grow. In this revision, we removed the points with p-value greater than 0.05 from Figure 8b and Figure 8c.

[Figure]

**Figure 8 (in the revised manuscript). Temporal trends of the 2BDA index over July and August. (a) 2BDA time series and temporal trend analysis over the KAN_L, KAN_M, and DS sites. (b) Regression slope and R2 estimates of the temporal trend analysis for the period of July–August (for areas where the p value <=0.05). (c) Regression slope and R2 estimates of the temporal trend analysis for the period of 20 July–20 August (for areas where the p value <=0.05).**

*Discussion: excessively wordy in places, can be shortened without loss of meaning. Fig. 11: provide colorbar to interpret density colors. Consider providing R2 values instead of just R.*

**Response**: We have revised the discussion section to improve clarity. We added the colorbar to Figure 11 as suggested. We use R instead of $R^2$ as our focus is to discuss the correlations between MAR albedo and MODIS albedo, and between the albedo bias and 2BDA index. The metric R

describes the strength of linear association between two variables, while $R^2$ is generally used in regression models to represent the amount of variability in y (dependent variable) that can be explained by the regression model, which is not the focus in our context.

*Text of page 22: this paragraph is overly long. It requires a re-structure.*

**Response:** We have restructured this part into three paragraphs and have revised the text.

*P22 L456-459: might be worth noting here that this is opposite to the results of Tedstone et al. (2017).*

**Response:** We have added discussion of the Tedstone et al. (2017) study as suggested.

*L460: 'For each of the two variables'*

**Response:** We have changed the text accordingly.

*P23 L474-481: reads hugely speculatively, especially given the relative lack of process-level understanding about ice algae available in the literature.*

**Response:** This statement is indeed somewhat speculative, but we have provided these suggestions based on our analysis as a point of discussion for future study. We restructured and rewrote this section to clarify that these statements are somewhat speculative and more work is necessary to better understand these relationships.

*Fig. 13a,b: why was a white mid-point of _0.97 chosen? Aren't algae judged to be present at values < 1?*

**Response**: We didn't intend to use 0.97 as a thresholding point. We changed the color scheme to avoid the confusion caused by color scheme. We also moved the subplots (a) and (b) to the appendix (Figure B1). As noted in the response to reviewer #1, there is no particular threshold for which algae are deemed to be present or not present. Algae may still be present at 2BDA values less than 1, as the background bare ice spectrum shows a decrease from 665 to 709 nm. However, as we note in response to reviewer #1, there is a higher likelihood of dust impacting the 2BDA index below values of 0.99.

[Figure]

**Figure B1 (in the revised manuscript). (a) 2BDA index versus surface elevation and roughness (elevation variability within each MERIS pixel). (b) 2BDA index versus near surface temperature and meltwater production simulated by MAR. The colour bars in (a) and (b) indicate the average 2BDA index for each two-dimensional bin defined by the two variables on the horizontal and vertical axes.**

*Fig. 13c,d,e: I am not sure what the relevance of providing these data are. At the very least it would be useful to add some kind of annual 2BDI and Impurity Index time series for comparison with the provided metrics.*

**Response:** We added the annual 2BDA time series to the figure as suggested. This analysis is meant to be a preliminary investigation of possible relationships between algae and climate forcing, and is provided as a discussion point for future research.

[revised manuscript text omitted]

**Summary list of all relevant changes made in the manuscript**

| | Sections | Changes |
|---|---|---|
| | Author info | We changed the correspondence email address as **wangshujie23@gmail.com** |
| **Text** | 1 Introduction | We rewrote paragraphs #5 and #6 to address the reviewers' comments, particularly about using the spectral feature of chlorophyll-a instead of the purpurogallin pigment. |
| | 2.1 Study area and previous field observations | We added more details about the field datasets and how we used them in this study. |
| | 3.2 Chlorophyll-a indices and impurity index | We added more details on the different indices, particularly the discussion about the impurity index as Reviewer#2 suggested. |
| | 3.3 Sensitivity analysis based on radiative transfer modelling | We added this new section in this revision to describe how we used the SNICAR model to examine the sensitivity of 2BDA index to dust presence. |
| | 4.1 Comparison between different ratio indices | We rewrote most of this section as Reviewer#2 suggested. In the course of making corrections to the manuscript we discovered a minor error in calculating the 3BDA index (which was not used to produce any results in the manuscript). The change does not affect any of our results or conclusions and we have corrected the manuscript with the correct 3BDA values. |
| | 4.2 Sensitivity analysis of 2BDA index to non-algal factors | We added this new section to analyze and discuss the sensitivity of the 2BDA index to dust presence and ice properties. |
| | 4.3 Spatial variability | We improved the readability of this section. |
| | 4.4 Interannual variability | For clarity, we moved some of the text to this new section and improved the writing for better readability. |
| | 4.5 Seasonal trends of algal growth over July and August | We improved the writing of this section and provided more methodological details about calculation of population doubling times. |
| | 5 Discussion | We restructured the Discussion into three sections: 5.1 Sensitivity to subpixel variability, 5.2 Relationship between regional climate model albedo bias and glacier algae, and 5.3 Potential drivers for glacier algae variability. |
| | 5.1 Sensitivity to subpixel variability | We added this new section to discuss the sensitivity of 2BDA index to the scale of MERIS data, by performing spectral linear mixing experiments using the field data and SNICAR-simulated data. |

| | | |
|---|---|---|
| | 5.3 Potential drivers for glacier algae variability | We restructured and rewrote this part by incorporating Reviewer#2's comments. |
| **Figures** | Figure 1 | Added more descriptions about the field spectrum in the figure caption |
| | Figure 2 | Added more field spectra of glacier algae-laden ice based on Stibal et al. (2017), to illustrate the consistency of the chlorophyll-a spectral signal between MERIS, WorldView-2, and field data. |
| | Figure 3 | Revised as Reviewer#2 suggested, and added a subplot to show the reflectance spectra of different surface types normalized to the bare ice spectra. |
| | Figure 4 | New figure added in this revision, to support the sensitivity analysis of the 2BDA index to dust presence based on SNICAR simulations and MERIS data. |
| | Figure 5 | Figure 4 in the original manuscript. Changed color scheme for better illustrating the spatial variability of different indices. |
| | Figure 6 | Figure 5 in the original manuscript. Changed as Reviewer#2 suggested. Changed color scheme for better illustration. |
| | Figure 8 | Changed subplots (b) and (c) by removing the pixels with P value greater than 0.05. |
| | Figure 11 | Changed as Reviewer#2 suggested by adding color bar. |
| | Figure 13 | Added the time series of 2BDA index and maximum bare ice extent as a subplot, and moved the original subplots (a) and (b) to Appendix B as Figure B1. |
| | Other changes | Removed the original Figure 7 from the manuscript. |
| **Appendix** | Figure A1 | New supplementary figure added in this revision, to show the different spatial variability of different indices varied with elevation levels. |
| | Figure A2 | New supplementary figure added in this revision, to illustrate the wavy patterns potentially caused by dust outcropping at 1000-1200m elevation based on high-resolution WorldView-2 imagery. |
| | Figure A3 | New supplementary figure added in this revision, to show the interannual variability of 2BDA index and impurity index at the elevation levels. |
| | Figure A4 | Changed from the original Figure A1. Added the maps of P values. |
| | Figure A5 | New supplementary figure added in this revision, to show the temporal trends of 2BDA index through mid-July to Mid-August in different years at sites DS (a) and KAN_M (b). |
| | Figure B1 | Moved from the original Figure 13(a) and (b). Changed color scheme to avoid ambiguity as noted by Reviewer#2. |
| | Table C1 | New supplementary table added in this revision, obtained from the field data of Cook et al. (2020), which provides additional evidence showing the positive correlations between 2BDA index and algal cell abundance. |
| | Figure C2 | New supplementary figure added in this revision, plotted based on Table C1. |
| | Figure C3 | New supplementary figure added in this revision, showing the spectral linear mixing results to support the discussion section 5.1 about the scale issues. |

[revised manuscript text omitted]

---

## Referee Report (RR1)

**Second review of Wang et al.: Quantifying spatiotemporal variability of glacier algal blooms and the impact on surface albedo in southwest Greenland.**

I thank the authors for a thorough response to my comments on the previous manuscript. Overall, I consider the revised version to be a big improvement, and I am sincerely appreciative of the work that has gone into addressing my comments. Several of my concerns have been mitigated by their detailed response, and I consider the support for glacier algal distribution estimates using the 2DBA index to be sufficient. I think there is a lot of value in the tracking of the 2DBA and albedo over the MERIS record, and the authors have presented these analyses well. However, I still feel there are some uncertainties where cell concentrations are quantified using the index. My comments below relate specifically to cell quantification.

Quantifying cell numbers requires that the index is insensitive to spectral mixing and variations in ice albedo. However, there is insufficient evidence that this is really the case. In their response to my initial comments, the authors argue that ice grain size does not influence the 2DBA index on the basis that: a) Ryan et al (2018) found distributed impurities to explain the majority of the albedo variation on the SW Greenland Ice Sheet; b) "to their knowledge" the 2DBA index is not affected by variations in ice albedo; c) selected MERIS spectra show similarities with selected field spectra. I do not find these to be terribly convincing arguments, because:

a) Ryan et al. (2018) also did not account for the changing albedo of the underlying ice, so it is plausible that the reason why "distributed impurities" were found to be responsible for the majority of the albedo variation is that the combined effect of the impurities and their feedbacks to the underlying ice structure were actually measured. Ryan et al. (2018) made no attempt to quantify impurity concentrations, and I do not see how their work supports the insensitivity of the 2DBA index to ice grain size.

b) I find this argument illogical. The 2DBA index compares reflectance at 665 and 709 nm. The latter is in the NIR range, which is precisely the region most sensitive to changes in grain size. Even if the visible wavelengths are insensitive to grain-size changes, i.e. the denominator in the 2DBA equation is fixed, the numerator is in the NIR wavelengths, so the index must surely be sensitive to grain size. I do not see convincing counter-arguments from the authors on this issue, but it is important because it undermines cell quantification using the 2DBA index.

c) Selecting similar spectra is only slightly helpful, since there are likely very many spectra that are less similar than the ones chosen.

The authors suggest in the manuscript that they conducted a sensitivity analysis for grain sizes 500 microns and 1500 microns but they do not comment on the results, only mentioning that changing ice density had a negligible effect. I would like to see the results of varying grain size on the 2DBA index but also note that the simulations have presumably been performed with dust as the primary light absorbing particle, which the authors assert does not produce a positive 2DBA index anyway.

There is also an assumption in the manuscript that 1500 microns is the appropriate grain size for representing glacier ice across the bare-ice zone, but there will be huge variability in this in the real world, which will feed forward into uncertainty in the 2DBA where it is used for quantifying cell abundance. The relationship established in Wang et al. (2018) is also based upon observed relationships between 300 x 300 m OLCI spectra with a very few centimeter-scale field spectra, so this association is also likely subject to the same uncertainties listed above. While their new linear mixing analysis is helpful, and adds value to the manuscript, it only really supports detection of algae, not quantification.

To reiterate, I think the authors have done a good job providing evidence and arguments in favour of their 2DBA index probably being suitable for assessing algal coverage/distribution. I think the majority of the paper is therefore publishable in TC. In my opinion, removing the sections that present cell quantifications derived from the index and the associated discussion would leave a much more robust paper, unless the authors can provide robust validation and estimates of uncertainty. This would be a relatively small change overall, as the majority of the paper does focus on spatial distribution.

---

## Author Response (AR2)

**Editor**

*Your revised TCD manuscript "Quantifying spatiotemporal variability of ice algal blooms and the impact on surface albedo in southwest Greenland" [tc-2019-226] has been seen again by both reviewers who both recommend that your manuscript can be accepted subject to minor revisions.*

*Before final acceptance, however, I recommend to address the comments raised by both reviewers. This includes the minor textual comments of reviewer 1&2, but also the important point raised by reviewer 2 regarding quantifying cell numbers.*

**Response:**

We are very grateful for all the comments and suggestions from the editor and two anonymous reviewers on this manuscript. We believe that we have resolved the issues raised by the reviewers and have improved the manuscript. In this revision, we made the corresponding changes suggested by the two reviewers. In particular, to further support our statement that the 2BDA index is not sensitive to the grain-size changes, we added a supplementary figure (Figure A1) in Appendix A to illustrate the low sensitivity of the 2BDA index to varying grain size and to validate the appropriateness of using 1500 microns to simulate the bare ice spectrum matching MERIS data. Additionally, we moved the quantification of algal population doubling time and albedo reduction rate due to algal population doubling to the discussion section 5.3. Since there are no field measurements coincident with the MERIS operational period, it is difficult to quantify the uncertainty of the 2BDA-inferred algal population variations. However, we think it is still meaningful to retain this discussion, since the magnitude inferred from the MERIS 2BDA index and Sentinel-3 empirical relationship compares well to data from previous field studies.

In this document, we provide point-by-point responses to the reviewer comments, a summary list of all relevant changes made in the manuscript, and a copy of the manuscript with changes highlighted. We believe that the manuscript has been greatly improved in this revision and we hope that the revised manuscript will be suitable for publication.

**Review #1**

The authors have made a thorough and engaged effort to answer my concerns and those of reviewer 1. This revised manuscript is hugely improved from the first submission: the scientific quality is high and the results are now communicated much more effectively and clearly.

I have only some minor comments remaining.

Line numbers refer to those in the revised manuscript.

**Response:** Thank you again for the careful reading and constructive comments. We have revised the manuscript as suggested. Please see our responses below.

*L196: Use of Alexander et al. (2014) citation is inappropriate. The reference to the meltwater scaling for albedo used in MAR should be cited to Zuo and Oerlemans (1996, Journal of Glaciology).*

**Response:** We have added the citation to Zuo and Oerlemans (1996) as suggested. As Alexander et al. (2014) discussed the meltwater scaling in the context of the MAR model, we have also retained this reference.

*Page 11-12. Presumably these definitions of "dark ice (low C-a)" versus "dark ice (high C-a)" are somewhat arbitrary, i.e. based on comparing the MERIS spectra of the two chosen pixels. This isn't a problem but I hope clarification of the definition can be added.*

**Response:** We have added the following sentence to the text for clarification:

*Since the magnitude of the chlorophyll-a-related spectral signal is directly related to algae concentration, we termed the northern site as 'dark ice (high chlorophyll-a)' and the southern site as 'dark ice (low chlorophyll-a)' in Figure 3 and Table 2.*

*Figure 4. Remove grey background to match other figures.*

**Response:** We have changed the figure as suggested.

*L374-378. Here, the suggestion is that darkening at 1000-1200 m may be due to inorganic particulates. Conversely, earlier, in Section 4.2, the authors argue that non-algal factors are relatively insignificant in darkening of the study region, citing Cook et al. (2020) to support their case. Yet, Cook et al. (2020) made their measurements at S6, which is at 1073 m asl – right in the middle of the elevation band being discussed here. As currently presented, this seems to be quite a contradiction and warrants further consideration.*

**Response:** In Section 4.2, we analyzed the sensitivity of the 2BDA index (indicating the content of chlorophyll-a or glacial algae, rather than 'darkening') to non-algal factors based on radiative transfer modeling experiments. In our context, 'darkening' refers to the reduction of surface albedo. Therefore, this statement is not contradictory to the discussion in Section 4.2. In Section 4.6, we mentioned that "The analysis shows a statistically significant correlation between algal growth and albedo reduction at the DS area between the altitudes of 800 m and 1200 m, the middle ablation zone between the altitudes of 1200 m and 1400 m, and the 1000–1200 m area near the K-transect". The field site S6 (Cook et al. 2020) is located at "the 1000–1200 m area near the K-transect". We have revised the text for better clarity.

*Figure A2 b,c: Please confirm processing steps undertaken on these images – are they both corrected to ground reflectance and therefore are the pixel values comparable between the images?*

**Response:** We have provided more information about the WorldView-2 imagery (Figure A2) in the figure caption. These two images are orthorectified Top-of-Atmosphere radiances, shown as a true color composite (Red channel: 659 nm band; Green channel: 546 nm band; Blue channel: 478 nm band). We use these two images to illustrate the 'wavy' pattern that Wientjes and Oerlemans (2010) suggested, as response to a question raised by Reviewer #2. Since our focus in this case is on the morphological pattern, it is not necessary to convert the radiance data to surface reflectance. However, it should be noted in Figure 2, we show surface reflectance data derived from WorldView-2 after atmospheric correction using ENVI FLAASH.

*Figure 8a: Please provide labelling information in the legend about the regression lines (presumably, one is for Jul-Aug and the other only for 20 Jul-20 Aug?)*

**Response:** We have added the labeling information to the figure as suggested.

*L464-468. These lines would be more readable if equations are typeset as per eqs. 3, 4.*

**Response:** As suggested by Reviewer #2, we have moved this part to the discussion section 5.3, and revised the text and equations to a more general and readable format.

*L471-473: Specify temporal bounds of Williamson study (i.e. 2016 only).*

**Response:** We have specified the time period (2016 summer) of the Williamson et al. field study.

*L478: Missing 'the' – 'to investigate THE potential impact'.*

**Response:** We have corrected the text.

*Figures 9, 10 and 12: Please format with graticules as per all other map figures.*

**Response:** As suggested, we have added graticules to Figures 9, 10, and 12.

*Please move Appendix B to after Appendix C, so represent the order that these are presented in the text.*

**Response:** We have changed the order as suggested.

**Review #2**

I thank the authors for a thorough response to my comments on the previous manuscript. Overall, I consider the revised version to be a big improvement, and I am sincerely appreciative of the work that has gone into addressing my comments. Several of my concerns have been mitigated by their detailed response, and I consider the support for glacier algal distribution estimates using the 2DBA index to be sufficient. I think there is a lot of value in the tracking of the 2DBA and albedo over the MERIS record, and the authors have presented these analyses well. However, I still feel there are some uncertainties where cell concentrations are quantified using the index. My comments below relate specifically to cell quantification.

**Response:** We sincerely appreciate the reviewer's constructive comments which have helped us further improve the manuscript. We believe that we have revolved the concerns raised by the reviewer. Please see our Reponses below.

Quantifying cell numbers requires that the index is insensitive to spectral mixing and variations in ice albedo. However, there is insufficient evidence that this is really the case. In their response to my initial comments, the authors argue that ice grain size does not influence the 2DBA index on the basis that: a) Ryan et al (2018) found distributed impurities to explain the majority of the albedo variation on the SW Greenland Ice Sheet; b) "to their knowledge" the 2DBA index is not

affected by variations in ice albedo; c) selected MERIS spectra show similarities with selected field spectra. I do not find these to be terribly convincing arguments, because:

a) Ryan et al. (2018) also did not account for the changing albedo of the underlying ice, so it is plausible that the reason why "distributed impurities" were found to be responsible for the majority of the albedo variation is that the combined effect of the impurities and their feedbacks to the underlying ice structure were actually measured. Ryan et al. (2018) made no attempt to quantify impurity concentrations, and I do not see how their work supports the insensitivity of the 2DBA index to ice grain size.

**Response:** With regard to quantifying cell numbers, in our original manuscript, we did not quantify the cell numbers based on the 2BDA index since we don't have field measurements of algal abundance coincident with the MERIS operational period. Instead, we attempted to quantify the temporal variation of algal population based on the 2BDA index and the correlation between algal population and surface albedo using the equation derived from the Sentinel-3 data in Wang et al. (2018). We agree that there are still uncertainties associated with the empirical relationship between 2BDA index and actual algal abundance that need to be quantified in future research. In this revision, we have therefore moved the material discussing algal population variability to the discussion section (5.3 Relationship of 2BDA index with algal population). It should be noted that this change does not affect the main results. Although we are unable to directly verify the applicability of the Sentinel-3-derived equation to MERIS data, we found the inferred algal population doubling time and albedo reduction rate due to algal population to be highly consistent with values obtained from previous field studies (Stibal et al., 2017; Williamson et al., 2018). Therefore, we think it is meaningful to retain this material as part of the discussion section in our manuscript.

With regard to varying grain size, we performed SNICAR simulations for dust-free scenarios with different values of snow grain effective radius ranging from 500 microns to 1500 microns. The simulation results are shown in Fig. A1a and Fig. A1b. We calculated the 2BDA values for different grain sizes using these spectra, finding the highest 2BDA value for the 500-micron case and the lowest 2BDA value for the 1500-micron case. These cases are labelled in Fig. A1c, which also provides a histogram of the 2BDA values calculated for the MERIS clean bare ice pixels. In addition, we have also provided the SNICAR simulation results with varying values of dust size and concentration using a grain size of 500 microns (Fig. A1d). Comparing Fig. A1d with the simulation results using 1500 microns (Fig. 4 in the manuscript), the high 2BDA values are still unlikely to be false positives caused by dust, that is, the high 2BDA values associated with glacial algae presence are insensitive to the dust presence and grain-size changes.

[Figure]

**Figure A1. (a) SNICAR dust-free simulations for different snow grain sizes (500–1500 μm). (b) Zoomed-in graph of (a) showing details of spectral albedos at 665 nm and 709 nm. (c) Histogram of the 2BDA index for the MERIS bare ice pixels with 620-nm reflectance greater than 0.65 (clean ice) and the corresponding 2BDA values for the SNICAR dust-free simulations with snow grain size of 500 and 1500 μm. (d) Impurity index vs. 2BDA index for MERIS bare ice pixels (density scatter plot with colours indicating relative frequency), and for SNICAR simulations (circles and dashed lines) with snow grain size of 500 μm for varying concentrations of dust (four different dust sizes). The circle size corresponds to the dust concentration, and dashed lines show the polynomial regression for each of the different dust sizes.**

b) I find this argument illogical. The 2DBA index compares reflectance at 665 and 709 nm. The latter is in the NIR range, which is precisely the region most sensitive to changes in grain size. Even if the visible wavelengths are insensitive to grain-size changes, i.e. the denominator in the 2DBA equation is fixed, the numerator is in the NIR wavelengths, so the index must surely be sensitive to grain size. I do not see convincing counter-arguments from the authors on this issue, but it is important because it undermines cell quantification using the 2DBA index.

**Response:** The band of 709 nm is located between the visible and NIR wavelengths. Using SNICAR, we simulated the spectra (without dust presence) for different values of snow grain

size (ranging from 500 µm to 1500 µm), as shown in Fig. A1a and Fig. A1b. It is shown that the effects of snow grain size on the 709 nm band are small compared with the effects on wavelengths greater than 750 nm. This is also the reason why it is important to use ocean color satellite sensors like Envisat MERIS and Sentinel-3 OLCI for computing this index. These sensors have narrow bandwidths (~10 nm) that are less sensitive to grain size changes compared with the broadband satellite sensors. Nevertheless, we also presented the simulation results with varying dust sizes and concentrations when the snow grain size is set as 500 µm (Fig. A1d). Comparing Fig. A1d and Fig.4 (in the manuscript), it is found that the variations of grain size primarily affect the lower range of 2BDA values. Therefore, the high values of 2BDA caused by glacial algae are not likely to be false positives caused by dust or grain-size changes.

c)  Selecting similar spectra is only slightly helpful, since there are likely very many spectra that are less similar than the ones chosen.

**Response:** In this revision, we provided the histogram distribution of the 2BDA values calculated for the MERIS clean bare ice pixels (all bare ice pixels with $R_{620nm} > 0.65$), as illustrated by Fig. A1c. We also calculated the 2BDA values for the dust-free SNICAR spectra simulated using different grain sizes, with the lowest 2BDA value for the 1500 µm, and highest 2BDA value for the 500 µm. We labeled these 2BDA values on Fig. A1c, which shows that the snow grain size of 1500 µm is a good representative value for the MERIS spectra. In fact, nearly 50% MERIS bare ice pixels have 2BDA values lower than the simulated value for the grain size of 1500 µm (the maximum grain size allowed in SNICAR). Therefore, the SNICAR simulations using 1500 µm are sufficient to test the sensitivity of 2BDA index to dust presence.

The authors suggest in the manuscript that they conducted a sensitivity analysis for grain sizes 500 microns and 1500 microns but they do not comment on the results, only mentioning that changing ice density had a negligible effect. I would like to see the results of varying grain size on the 2DBA index but also note that the simulations have presumably been performed with dust as the primary light absorbing particle, which the authors assert does not produce a positive 2DBA index anyway.

**Response:** Please see our response above and the newly added Figure A1 in Appendix A.

There is also an assumption in the manuscript that 1500 microns is the appropriate grain size for representing glacier ice across the bare-ice zone, but there will be huge variability in this in the real world, which will feed forward into uncertainty in the 2DBA where it is used for quantifying cell abundance. The relationship established in Wang et al. (2018) is also based upon observed relationships between 300 x 300 m OLCI spectra with a very few centimeter-scale field spectra, so this association is also likely subject to the same uncertainties listed above. While their new linear mixing analysis is helpful, and adds value to the manuscript, it only really supports detection of algae, not quantification.

**Response:** Please see our response above about the appropriateness of using the grain size of 1500 microns. Nevertheless, as we mentioned above, we also provided the simulation results for the 500 microns (Fig.A1d), which does not impact our dust sensitivity analysis. The high 2BDA

values are unlikely to be caused by either dust presence or grain-size changes. As for the quantification of algal population, please also see our response above.

To reiterate, I think the authors have done a good job providing evidence and arguments in favour of their 2DBA index probably being suitable for assessing algal coverage/distribution. I think the majority of the paper is therefore publishable in TC. In my opinion, removing the sections that present cell quantifications derived from the index and the associated discussion would leave a much more robust paper, unless the authors can provide robust validation and estimates of uncertainty. This would be a relatively small change overall, as the majority of the paper does focus on spatial distribution.

**Response:** As we mentioned above, we have moved the estimations of the algal population doubling time and the albedo reduction rates due to algal population doubling to the discussion section 5.3. We think it is meaningful to compare the inferred algal population doubling time and albedo reduction rates, but acknowledge the uncertainty in this estimate given the lack of coincident in situ measurements. However, it should be noted that this change does not affect the quantification of algal seasonal variations and the correlation between surface albedo and the 2BDA algal index.

**Summary list of all relevant changes made in the manuscript**

|  | Sections | Changes |
| --- | --- | --- |
|  | Author info | Updated the affiliation information of Xavier Fettweis |
| **Text** | 3.3 | We added the dust-free SNICAR simulations for varying grain size from 500 microns to 1500 microns. |
|  | 4.1 and 4.3 | We made clarifications as suggested by Reviewer #1. |
|  | 4.2 | We added the discussion of the impact of grain-size changes on 2BDA index. |
|  | 4.5 | We moved the estimation of algal population doubling time to the discussion section 5.3. |
|  | 4.6 | We moved the estimation of albedo reduction due to algal population doubling to the discussion section 5.3. |
|  | 5 Discussion | We moved the inferred algal population doubling time and albedo reduction due to algal population doubling to a new discussion section 5.3 (Relationship of 2BDA index with algal population), in response to Reviewer #2. |
| **Figures** | Figure 4 | We removed the grey background as suggested by Reviewer #1. |
|  | Figure 8 | We added the labelling information in the legend about the regression lines as suggested by Reviewer #1. |
|  | Figure 9 | We added graticules as suggested by Reviewer #1. |
|  | Figure 10 | We added graticules as suggested by Reviewer #1. We also updated the figure using the regression results between 2BDA index and surface albedo. |
|  | Figure 12 | We added graticules as suggested by Reviewer #1. We also updated the figure using the regression results between 2BDA index and albedo bias. |
| **Appendix** | Figure A1 | We added this supplementary figure to illustrate the impact of grain-size changes to the 2BDA index, in response to Review #2. |
|  | Other changes | We switched the order of Appendix B and Appendix C as suggested by Reviewer #1. |

[revised manuscript text omitted]
 also tested different values of snow  density (400 kg/m$^3$ vs. 900 kg/m$^3$), and found that the snow density value had a negligible effect on the simulation results. To evaluate the impact of snow grain size on the 2BDA index, we performed dust-free SNICAR simulations for different values of snow grain effect radius between 500 μm and 1500 μm (Fig. A1, Appendix A). We calculated the 2BDA index for each dust-free scenario, finding the lowest 2BDA value (0.959) for the 1500-μm spectra and

290 the highest 2BDA value (0.976) for the 500-µm spectra. We compared these two 2BDA values with the histogram distribution of the 2BDA values calculated for the MERIS 'clean' ($R_{620nm} \geq 0.65$) bare ice pixels (Fig. A1c). The sensitivity test suggests that the dust-free spectrum simulated  using the 1500-µm grain size is a good approximation to the MERIS bare ice spectrum. Nevertheless, to account for the potential influence of grain-size changes on the sensitivity of 2BDA index to dust presence, we also repeated the SNICAR simulations with varying dust sizes and concentrations with a

295 snow grain effective radius of 500 µm.

**4 Results**

**4.1 Comparison between different ratio indices**

[revised manuscript text omitted]
 analysed the  relationship between surface albedo reduction and algal growth using the time series data of MODIS broadband albedo and the MERIS 2BDA index. Figure 10 shows the results of  regression analysis with MODIS albedo as the dependent variable and the MERIS 2BDA index as the independent variable. The analysis indicates a statistically significant correlation between algal growth and albedo decrease at the DS area between the altitudes of 800 m and 1200 m, the middle ablation zone between the altitudes of 1200 m and 1400 m, and the 1000–1200 m area near the K-transect. Over these areas, the regression coefficient ranged from -4 to -2. Given the temporal rate of change of 2BDA index of 0.001 per day (Section 4.5), the surface albedo ~~decrease of 2~4% for eachpopulation doubling. This estimate is comparable to results from the field study byStibal et al. (2017) who estimated a net albedo reduction of 0.038±0.0035 for each algal population doubling based on-situ measurements of glacier algal abundance and coincident surface albedo. In general, glacier algal growth explains most of the temporal variability ofinbetween 1000 and 1400 m in elevation,areain elevation where~~where the correlation is less significant and other factors likely contribute to the observed albedo reduction.

[Figure]

**Figure 10. Relationship between**  **surface albedo and 2BDA index. (a) Regression coefficients. (b) Standard errors of the correlation coefficients. (c) P values. (d) $R^2$ values. (e) and (f) show**  **surface albedo versus 2BDA index at Site 1 and Site 2, respectively.**

**5 Discussion**

**5.1 Sensitivity to subpixel variability**

[revised manuscript text omitted]

**5.3 Relationship between 2BDA index and algal population**

In order to use remote sensing data to quantify the temporal change of algal population with time, it is necessary to establish
600 an empirical relationship between 2BDA index and algal abundance. However, there is no field data for glacial algal abundance coincident with the MERIS operational period. In ocean color studies, the relationship between 2BDA index and chlorophyll-a concentration is generally formulated as an exponential function or a linear function (Matthews, 2011; Gholizadeh et al., 2016). Wang et al. (2018) derived an exponential function relating the Sentinel-3 OLCI reflectance ratio $R_{709nm}/R_{673nm}$ and field data for glacier algal abundance (Stibal et al., 2015; Williamson et al., 2018) as:

605 $$y = 10^{-35} * e^{87.015*x} \tag{3}$$

where $x$ denotes the reflectance ratio and $y$ denotes the algal abundance (cells/ml). Following from equation (3), the empirical relationship between algal abundance and 2BDA index can be represented in a general form as:

$$y = a * b^{cx} \tag{4}$$

where $x$ denotes the 2BDA index, y is algal abundance, $b$ is the base number of the exponential function ($e$ in equation 3),
610 and $a$ and $c$ are the regression coefficients. The time for one algal population doubling (the number of algal cells has

doubled) can be calculated as the reciprocal of $\frac{d}{dt}(log_2 y)$, where $t$ represents time. Based on equation (4) and derivative rules, $\frac{d}{dt}(log_2 y)$ can be represented as:

$$\frac{d}{dt}(log_2 y) = c * log_2 b * \frac{dx}{dt} \tag{5}$$

where $\frac{dx}{dt}$ is the rate of change of 2BDA with time, i.e. the regression coefficient from the temporal trend analysis of 2BDA

615 index versus time (Section 4.5; Figure 8). Similarly, the relationship between surface albedo ($\alpha$) and algal population doubling level ($log_2 y$) can be calculated using:

$$\frac{d\alpha}{d(log_2 y)} = \frac{1}{c*log_2 b} * \frac{d\alpha}{dx} \tag{6}$$

where $\frac{d\alpha}{dx}$ is the regression coefficient of surface albedo $\alpha$ vs. 2BDA index (Section 4.6; Figure 10).

Given the similarity between the OLCI and MERIS band configurations and the negligible differences between the 673 nm

620 and 665 nm reflectance, we attempted to use equations (3), (5), and (6) to calculate the algal population doubling time corresponding to various values of the regression coefficients of 2BDA vs. time, as well as the albedo change rate due to each algal population doubling corresponding to different regression coefficients of albedo vs. 2BDA. The results of these calculations are listed in Table 3 and Table 4, respectively. According to Fig. 8c, the areas with significant algal growth trend ($R^2>0.5$) between 20 July and 20 August had a mean regression coefficient of 0.00076±0.0002, which corresponds to a mean

625 algal population doubling time of 11.2±2.6 days. The DS area had faster algal growth rate than other areas, which corresponds to a doubling time of 9.6±2.7 days. Figure 10a indicates that the regression coefficient of albedo vs. 2BDA over the algae-abundant areas ranges between -4 to -2, corresponding to a surface albedo decrease of 0.032~0.016 for each algal population doubling. Although these values were inferred using the Sentinel-3 derived relationship and there are uncertainties (e.g. spectral mixing) associated with algal abundance quantification, it is notable that our derived doubling

630 time and albedo impact estimates are comparable to previous field studies. Williamson et al. (2018) reported a doubling time of 7.18±1.04 days for algae-abundant ice (at the K-transect) based on field data collected during the summer of 2016. Stibal et al. (2017) estimated a net albedo reduction of 0.038±0.0035 for each algal population doubling based on in-situ measurements of glacier algal abundance and coincident surface albedo. Despite the apparent agreement, further research is required to build a robust relationship between 2BDA index and algal abundance and also quantify the uncertainties caused

635 by different factors.

**Table 3. Inferred algal population doubling time for given regression coefficients of 2BDA vs. time**

| Regression coefficient of 2BDA vs. time | Population doubling time (days) |
|---|---|
| 0.0004 | 19.91 |
| 0.0005 | 15.93 |
| 0.0006 | 13.28 |
| 0.0007 | 11.38 |

| | |
|---|---|
| 0.0008 | 9.96 |
| 0.0009 | 8.85 |
| 0.0010 | 7.97 |
| 0.0015 | 5.31 |
| 0.0020 | 3.98 |

**Table 4. Inferred surface albedo change rate due to algal doubling given regression coefficients of albedo vs. 2BDA index**

[revised manuscript text omitted]

**Figure A1.** (a) SNICAR dust-free simulations for different snow grain sizes (500–1500 µm). (b) Zoomed-in graph of (a) showing details of spectral albedo values at 665 nm and 709 nm. (c) Histogram of the 2BDA index for MERIS bare ice pixels with 620-nm reflectance greater than 0.65 (clean ice) and the corresponding 2BDA values for the SNICAR dust-free simulations with snow grain size of 500 and 1500 µm. (d) Impurity index vs. 2BDA index for MERIS bare ice pixels (density scatter plot with colours indicating relative frequency), and for SNICAR simulations (circles and dashed lines) with snow grain size of 500 µm for varying concentrations of dust (four different dust sizes). The circle size corresponds to the dust concentration, and dashed lines show the polynomial regression for each of the different dust sizes.

[revised manuscript text omitted]